# Learning in an Interactive Simulation Tool against Landslide Risks: The Role of Strength and Availability of Experiential Feedback

**Pratik Chaturvedi**[1, 2], **Akshit Arora**[1, 3], **and Varun Dutt**[1]

[1]Applied Cognitive Science Laboratory, Indian Institute of Technology, Mandi- 175005, India

[2]Defence Terrain Research Laboratory, Defence Research and Development Organization, Delhi -110054, India

[3]Computer Science and Engineering Department, Thapar University, Patiala - 147004, India

*Correspondence to*: Pratik Chaturvedi (prateek@dtrl.drdo.in)

**Abstract.** Feedback via simulation tools is likely to help people improve their decision-making against natural disasters. However, little is known on how differing strengths of experiential feedback and feedback's availability in simulation tools influences people's decisions against landslides. We tested the influence of differing strengths of experiential feedback and feedback's availability on people's decisions against landslides in Mandi, Himachal Pradesh, India. Experiential feedback (high or low) and feedback's availability (present or absent) were varied across four between-subject conditions in an interactive landslide simulation (ILS) tool: high-damage feedback-present, high-damage feedback-absent, low-damage feedback-present, and low-damage feedback-absent. In high-damage conditions, the probabilities of damages to life and property due to landslides were 10-times higher than those in the low-damage conditions. In feedback-present conditions, experiential feedback was provided in numeric, text, and graphical formats in ILS. In feedback-absent conditions, the probabilities of damages were described, however, there was no experiential feedback present. Investments were greater in conditions where experiential feedback was present and damages were high compared to conditions where experiential feedback was absent and damages were low. Furthermore, only high-damage feedback produced learning in ILS. Simulation tools like ILS seem appropriate for landslide risk communication and for performing what-if analyses.

## 1   Introduction

Landslides cause massive damages to life and property worldwide (Chaturvedi and Dutt, 2015; Margottini et al., 2011). Imparting knowledge about landslide causes-and-consequences as well

as spreading awareness about landslide disaster mitigation are likely to be effective ways of
managing landslide risks. The former approach supports structural protection measures that are
likely to help people take mitigation actions and reduce the probability of landslides (Becker et
al., 2013; Osuret et al., 2016; Webb and Ronan, 2014). In contrast, the latter approach likely
reduces people's and assets' perceived vulnerability to risk. However, it does not influence the
physical processes. One needs effective landslide risk communication systems (RCSs) to educate
people about cause-and-effect relationships concerning landslides (Glade et al., 2005). To be
effective, these RCSs should possess five main components (Rogers and Tsirkunov, 2011):
monitoring; analysing, risk communication, warning dissemination, and capacity building.
Among these components, prior research has focused on monitoring and analysing the
occurrence of landslide events (Dai et al., 2002; Montrasio et al., 2011). For example, there exist
various statistical and process-based models for predicting landslides (Dai et al., 2002; Montrasio
et al., 2011; Reder et al., 2018; Segoni et al., 2018; Vaz et al., 2018). Several satellite-based and
sensor-based landslide monitoring systems are being used in landslide RCSs (Hong et al., 2006;
Quanshah et al., 2010; Rogers et al., 2011; Frodella et al., 2017; Intrieri et al., 2017). To be
effective, however, landslide RCSs need not only be based upon sound scientific models, but,
they also need to consider human factors, i.e., the knowledge and understanding of people
residing in landslide-prone areas (Meissen and Voisard, 2008). Thus, there is an urgent need to
focus on the development, evaluation, and improvement of risk communication, warning
dissemination, and capacity building measures in RCSs.
Improvements in risk communication strategies are likely to help people understand the
cause-and-effect processes concerning landslides and help them improve their decision-making
against these natural disasters (Grasso and Singh, 2009). However, surveys conducted among
communities in landslide-prone areas (including those in northern India) have shown a lack of
awareness and understanding among people about landslide risks (Chaturvedi and Dutt, 2015;
Oven, 2009; Wanasolo, 2012). In a survey conducted in Mandi, India, Chaturvedi and Dutt
(2015) found that 60% of people surveyed were not able to answer questions on landslide
susceptibilities maps, which were prepared by experts. Also, Chaturvedi and Dutt (2015) found
that a sizeable population reported landslides to be "acts of God" (39%) and attributed activities
like "shifting of temple" as causing landslides (17%). These results are surprising as the literacy-
rate in Mandi and surrounding areas is quite high (81.5%) (Census, 2011) and these results show
numerous misconceptions about landslides among people in landslide-prone areas. Overall,
urgent measures need to be taken that improve public understanding and awareness about
landslides in affected areas.
Promising recent research has shown that experiential feedback in simulation tools likely
helps improve public understanding about dynamics of physical systems (Chaturvedi et al., 2017;
Dutt and Gonzalez, 2010; 2011; 2012; Fischer, 2008). Dutt and Gonzalez (2012) developed a
Dynamic Climate Change Simulator (DCCS) tool, which was based upon a more generic stock-
and-flow task (Gonzalez and Dutt, 2011a). The authors provided frequent feedback on cause-
and-effect relationships concerning Earth's climate in DCCS and this experiential feedback
helped people reduce their climate misconceptions compared to a no-DCCS intervention.
Although the prior literature has investigated the role of frequency of feedback about inputs and
outputs in physical systems, little is known on how differing strengths of experiential feedback
(i.e., differing probabilities of damages due to landslides) influences people's decisions over
time. Also, little is known on how experiential feedback's availability (presence or absence) in
simulation tools influences people's decisions.
The primary goal of this research is to evaluate how differing strengths of experiential
feedback and feedback's availability influences people's mitigation decisions against landslides.
A study of how the strength of experiential feedback influences people's decisions against
landslides is important because people's experience of landslide consequences due to differing
probabilities of landslide damages could range from no damages at all to large damages
involving several injuries, infrastructure damages, and deaths. Thus, due to differing
probabilities of landslide damages, some people may experience severe landslide damages and
consider landslides to be a serious problem requiring immediate actions; whereas, other people
may experience no damages and consider landslides to be a trivial problem requiring very little
attention.
In addition, the availability of feedback in simulation tools is also likely to influence
people's decisions against landslides. When feedback is absent, people are only likely to acquire
descriptive knowledge about the cause-and-effect relationships governing the landslide dynamics
(Dutt and Gonzalez, 2010). However, when feedback is present, people get to repeatedly
experience the positive or negative consequences of their decisions against landslide risks (Dutt
and Gonzalez, 2010; 2011). This repeated experience will likely help people understand the
cause-and-effect relationships governing the landslide dynamics.
Chaturvedi et al. (2017) proposed a computer-simulation tool, called the Interactive
Landslide Simulator (ILS). The ILS tool is based upon a landslide model that considers the
influence of both human factors and physical factors on landslide dynamics. Thus, in ILS, both
physical factors (e.g., spatial geology and rainfall) and human factors (e.g., monetary
contributions to mitigate landslides) influence the probability of catastrophic landslides. In a
preliminary investigation involving the ILS tool, Chaturvedi et al. (2017) varied the probability
of damages due to landslides at two levels: low probability and high probability. The high
probability was set about 10-times higher compared to the low probability. People were asked to
make monetary investment decisions, where people's monetary payments would be used for
mitigating landslides (e.g., by building a retaining wall, planned road construction, provision of
proper drainage or by planting crops with long roots in landslide-prone areas; please see Patra
and Devi (2015) for a review of such mitigation measures). People's investments were
significantly greater when the damage probability was high compared to when this probability
was low. However, Chaturvedi et al. (2017) did not fully evaluate the effectiveness of
experiential feedback of damages in ILS tool against control conditions where this experiential
feedback was not present. Also, Chaturvedi et al. (2017) did not investigate people's investment
decisions over time and certain strategies in ILS, where these decisions and strategies would be
indicative of learning of landslide dynamics in the tool.
Prior literature on learning from experiential feedback (Baumeister et al., 2007; Dutt and
Gonzalez, 2012; Finucane et al., 2000; Knutty, 2005; Reis and Judd, 2013; Wagner, 2007)
suggests that increasing the strength of damage feedback by increasing the probabilities of
landslide damages in simulation tools would likely increase people's mitigation decisions. That
is because a high probability of landslide damages will make people suffer monetary losses and
people would tend to minimize these losses by increasing their mitigation actions over time. It is
also expected that the presence of experiential feedback about damages in simulation tools is
likely to increase people's landslide-mitigation actions over time (Dutt and Gonzalez, 2010;
2011; 2012). That is because the experiential feedback about damages will likely enable people
to make decisions and see the consequences of their decisions, however, the absence of this
feedback will not allow people to observe the consequences of their decisions once these
decisions have been made (Dutt and Gonzalez, 2012). At first glance, these explanations may
seem to assume people to be economically rationale individuals while facing landslide disasters
(Bossaerts and Murawski, 2015; Neumann and Morgenstern, 1947), where one disregards
people's bounded rationality, risk perceptions, attitudes, and behaviours (De Martino, Kumaran,
Seymour, and Dolan; 2005; Gigerenzer and Selten, 2002; Kahneman and Tversky, 1979; Simon,
1959; Slovic, Peters, Finucane, and MacGregor, 2005; Thaler and Sunstein, 2008; Tversky and
Kahneman, 1992). However, in this paper, we consider people to be bounded rational agents
(Gigerenzer and Selten, 2002; Simon, 1959), who tend to minimize their losses against landslides
slowly over time via a trial-and-error learning process driven by personal experience in an
uncertain environment (Dutt and Gonzalez, 2010; Slovic et al., 2005).
In this paper, we evaluate the influence of differing strengths of experiential feedback
about landslide-related damages and the experiential feedback's availability in the ILS tool.
More specifically, we test whether people increase their mitigation actions in the presence of
experiential damage feedback compared to in the absence of this feedback. In addition, we
evaluate how different probabilities of damages influence people's mitigation actions in the ILS
tool. Furthermore, we also analyse people's mitigation actions over time across different
conditions.
In what follows, first, we detail the characteristics of the study area, and then a
computational model on landslide risks that considers the role of both human factors and
physical factors. Next, we detail the working of the ILS tool, i.e., based on the landslide model.
Furthermore, we use the ILS tool in an experiment to evaluate the influence of differing strengths
of experiential feedback and feedback's availability on people's decisions. Finally, we close this
paper by discussing our results and detailing the benefits of using tools like ILS for
communicating landslide risks in the real world.

## 2   Study area

In this paper, the study area was one involving the local communities living in the Mandi town
(31.58° N, 76.91° E), a township located in the state of Himachal Pradesh, India (see Figure 1).
The Mandi town has an average elevation of 850m above mean-sea level, 23 square km area, and
a population of 26,422 people (Census, 2011). Literacy rate in Mandi town is 81.5% and most of
the population are Hindus by religion. Mandi is a highly religious place with a huge number of
Hindu temples all around the town (Census, 2011). Geologically, Mandi town is located on the
folds of the lesser Himalayan mountains and it lies in the earthquake Zone IV and V, the highest
earthquake zones in the world (Hpsdma, 2017). Apart from inherent geological weaknesses that
may cause landslides in Mandi town, other anthropogenic activities such as road construction,
deforestation of hill slopes, building construction on slopes, and debris dumping may also trigger
landslides in the area surrounding the town (Hpsdma, 2017). As per Kahlon, Chandel, and Brar
(2014), around 90% of the Mandi town is prone to landslides, where 25% of this area falls under
the severe landslide hazard risk category. Landslide occurrences during the past 39 years (from
1971 to 2009) exhibit Mandi to account for 99 landslide events (11%) out of a total 919 landslide
events in Himachal Pradesh, forming the 4[th] highest ranked district in terms of number landslides
behind Shimla, Solan, and Kinnaur (Kahlon et al., 2014). The problem of landslides is
accelerated in the monsoon season (mid-June to mid-September) in the town. The per-capita
income of people in the Mandi town is close to INR 292 (~ USD 4.48 or EUR 3.63) per day
(Census, 2011). In addition, as per the tenancy laws of Himachal Pradesh, most people own land,
which cannot be sold to people from outside the state (Himachal, 2012). The average per-capita
property value in the state would be close to INR 20 million (Census, 2011). These values of per-
capita daily income and property wealth were used in the ILS tool and these values have been
detailed ahead in this paper. Furthermore, the prevailing rainfall pattern and the landslide hazard
zonation map of Mandi town, which were used in the ILS tool, have also been detailed ahead in
this paper.

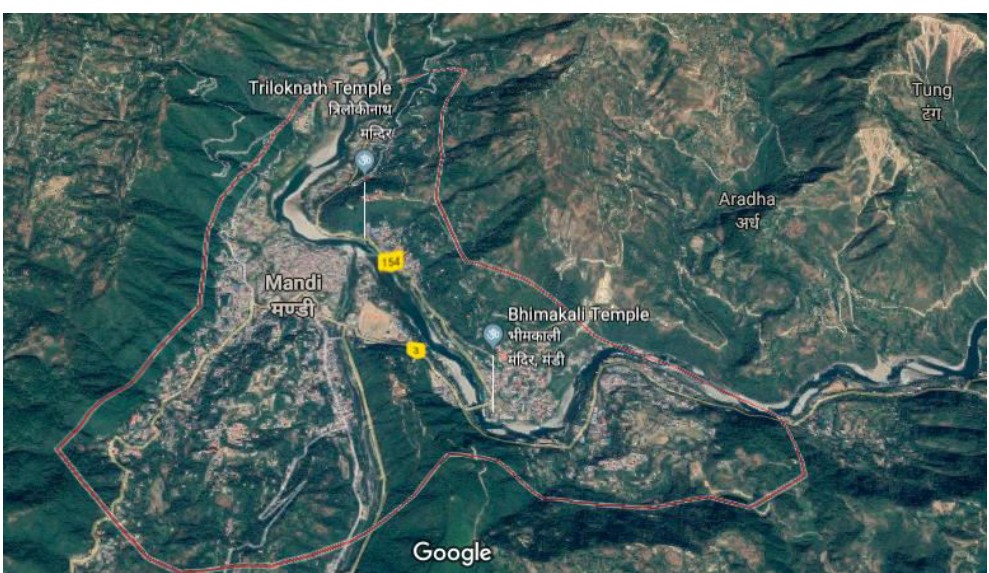

 Figure 1. The 3D satellite view of Mandi town and adjoining areas. The town is located in a valley around river Beas
with high mountains that are prone to landslides on both sides. Source: Google Maps.

## 3    Computational model of landslide risk

Chaturvedi et al. (2017) had proposed a computational model for simulating landslide risks that
was based upon the integration of human and physical factors (see Figure 2). Here, we briefly
detail this model and use it in the ILS tool for our experiment (reported ahead). As seen in Figure
2, the probability of landslides due to human factors in the ILS tool is adapted from a model
suggested by Hasson et al. (2010) (see box 1.1 in Figure 2). In Hasson et al. (2010)'s model, the
probability of a disaster (e.g., landslide) due to human factors (e.g., investment) was a function
of the cumulative monetary contributions made by participants to avert the disaster from the total
endowment available to participants. Thus, investing against the disaster in mitigation measures
reduces the probability of the disaster and not investing in mitigation measures increases the
probability of the disaster. However, by reducing the landslide risk, people also have lesser
ability in investing in other profitable investments due to loss in revenue. Although we assume
this model to incorporate human mitigation actions in the ILS tool, there may also be other
model assumptions possible where certain detrimental human actions (e.g., deforestation) may
increase the probability of landslides or the risk of landslides (where, risk = probability (hazard)
* consequence). We plan to consider such model assumptions as part of our future research. In
addition, there may be contributions made by the national, regional, and local governments for
providing protection measures against landslides in addition to the investments made by people
residing in the area (Hpsdma, 2017). Such investments may be made based upon the past
occurrences of landslides in the study area. Furthermore, people may also be able to buy
insurance that covers for the damages caused by landslides. However, in India, in the absence of
assistance from the government, mostly people tend to rely on their own wealth for adaptation to
landslide occurrence. Thus, purchasing insurance against disasters is less common and unpopular
as insurance companies mostly do not pay insured amounts in the event of natural disasters like
landslides (ICICI, 2018). In this paper, we restrict our analyses to only people's own investments
influencing landslides. We plan to consider the role of government contributions for mitigation
and adaptation (mostly after landslide events) and partial insurance payments as part of our
future research.
Furthermore, in the landslide model, the probability of landslides due to physical
(natural) factors (see box 1.2) is a function of the prevailing rainfall conditions and the nature of
geology in the area (Mathew et al., 2013). In this paper, we restrict our focus to considering only
weather (rainfall)-induced landslides. As shown in Figure 2, the ILS model focuses on
calculation of total probability of landslide (due to physical and human factors) (box 1.3). This
total probability of landslide is calculated as a weighted sum of probability of landslide due to
physical factors and probability of landslide due to human factors. Furthermore, the model
simulates different types of damages caused by landslides and their effects on people's earnings
(box 1.4).

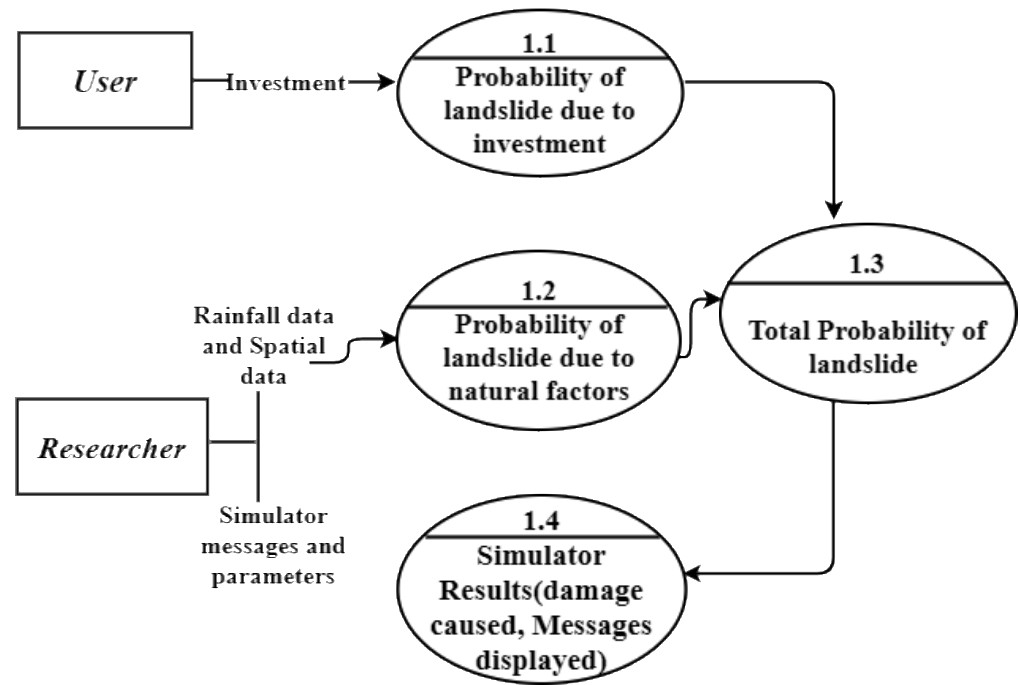


**Figure 2.** Probabilistic model of the Interactive Landslide Simulator tool. Figure adapted from Chaturvedi et al. (2017).

**3.1    Total probability of landslides**
As described by Chaturvedi et al. (2017), the total probability of landslides is a function of
landslide probabilities due to human factors and physical factors. This total probability of
landslides can be represented as the following:
$$P(T) = \big(W * P\,(I) + (1 - W) * P\,(E)\big) \quad (1)$$

Where W is a free weight parameter in [0, 1]. The total probability formula involves calculation
of two probabilities, probability of landslide due to human investments (*P(I)*) and probability of
landslide due to physical factors (*P(E)*). These probabilities have been defined below. According
to Equation 1, the total probability of landslides will change based upon both human decisions
and environmental factors over time. In the ILS model, we simulate the total probability of
landslides *P(T)*, where a landslide occurs when a uniformly distributed random number (~ *U(0,*
*1)*) is less than or equal to *P(T)* on a certain day. If a uniformly distributed random number in [0,
1] (*U (0, 1)*) is less than or equal to a point probability value, then it simulates this point
probability value. For example, if *U (0, 1)* ≤ 30%, then *U (0, 1)* will be less than or equal to the
30% value exactly 30% of the total number of times it is simulated; and, thus this random
process will simulate a 30% probability value.

**3.1.1    Probability of landslide due to human investments (*P(I)*)**
As suggested by Chaturvedi et al. (2017), the probability *P(I)* is calculated using the probability
model suggested by Hasson et al. (2010). In this model, *P(I)* is directly proportional to the
amount of money invested by participants for landslide mitigation. The probability of landslide
due to human investments is:
$$P(I) = 1 - \frac{M * \sum_{i=1}^{n} x_i}{n * B} \qquad (2)$$

Where,
*B* = Budget available towards addressing landslides for a day (if a person earns an income or
salary, then B is the same as this income or salary earned in a day).
*n* = Number of days.
$x_i$ = Investments made by a person for each day *i* to mitigate landslides; $x_i \leq$ B.
*M* = Return to Mitigation, which is a free parameter and captures the lower bound probability of
*P(I)*, i.e., *P (I) = 1- M* when a person puts her entire budget B into landslide mitigation ($\sum_{i=1}^{n} x_i$
$= n * B$); $0 \leq M \leq 1$.
People's monetary investments ($x_i$) are for mitigation measures like building retaining walls or
planting long root crops.

**3.1.2    Probability of landslide due to physical factors (*P(E)*)**
Some of the physical factors impacting landslides include rainfall, soil types, and slope profiles
(Chaturvedi et al., 2017; Dai et al., 2002). These factors can be categorized into two parts:

1. Probability of landslide due to rainfall (*P(R)*)

2. Probability of landslide due to soil types and slope profiles (spatial probability,

*P(S)*)

For the sake of simplicity, we have assumed that *P(S)* is independent of *P(R)*. Thus, given *P(R)*
and *P(S)*, the probability of landslide due to physical factors, *P(E)*, is defined as:
$$P(E) = P(R) * P(S) \qquad (3)$$
In the first step, *P(R)* is calculated based upon a logistic-regression model (Mathew et al., 2013)
as follows:
$$P(R) = \frac{1}{1+e^{-z}} \qquad (4a)$$
And,

$$z = -3.817 + (DR) * 0.077 + (3DCR) * 0.058 + (30DAR) * 0.009$$

$$z: (-\infty, +\infty) \qquad (4b)$$
Where, the $DR$, $3DCR$, and $30DAR$ is the daily rainfall, the 3-day cumulative rainfall, and the
30-day antecedent rainfall in the study area. This model in equations 4a and 4b was developed
for the study area by Mathew et al. (2013) and we have used the same model in this paper. The
rainfall parameters in the model were calculated from the daily rain data from the Indian
Metrological Department (IMD). We compared the shape of the *P(R)* distribution by averaging
rainfall data over the past five years with the shape of the *P(R)* distribution by averaging rainfall
data over the past 30-years. This comparison revealed that there were no statistical differences
between these two distributions. Thus, we used the daily rainfall data averaged over the past 5-
years (2010-14) to find the average rainfall values on each day out of the 365-days in a year.
Next, these averaged rainfall values were put into equations 4a and 4b to generate the landslide
probability due to rainfall (*P(R)*) over an entire year. Figure 3 shows the resulting shape of *P(R)*
distribution as a function of days in the year for the study area. Due to the monsoon period in
India during mid-June – mid-September, there is a peak in the *P(R)* distribution curve during
these months. Depending upon the start date in the ILS tool, one could read *P(R)* values from
Figure 3 as the probability of landslides due to rainfall on a certain day in the year. This *P(R)*
function was assumed to possess the same shape across all participants in the ILS tool.

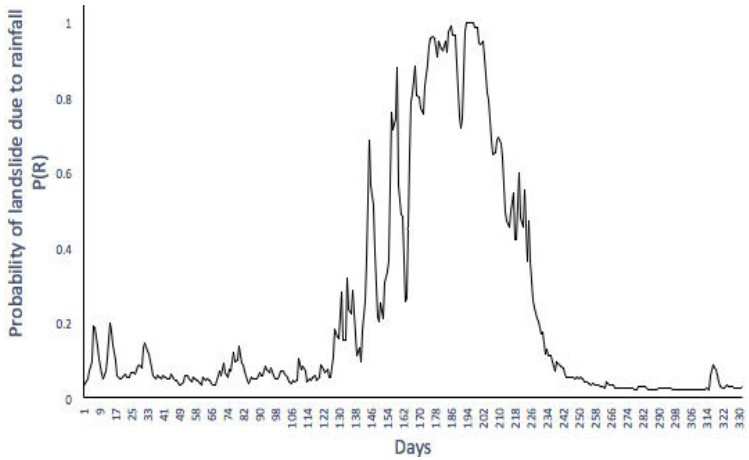


**Figure 3.** Probability of landslide due to rainfall over days for the study area. The probability was generated by using equations 4a and 4b.

The second step is to evaluate the spatial probability of landslides, *P(S)*. The determination of *P(S)* is done from the landslide hazard zonation (LHZ) map of the study area (see Figure 4A; Anbalagan, 1992; Chaturvedi et al., 2017; Clerici et al., 2002), which provides the landslide susceptibility of the area and it is based on various landslide causative factors in the study area (e.g., geology, geometry, and geomorphology). As shown in Figure 4A, we computed the spatial probability of landslides in the study area based upon the Total Estimated Hazard (THED) rating of different locations on a LHZ map (see legend) and their surface area of coverage (the maximum possible value of THED is 11.0 and its minimum possible value is 0.0). Table 1 provides the THED scale to report the susceptibility of an area to landslides (Anbalagan, 1992).

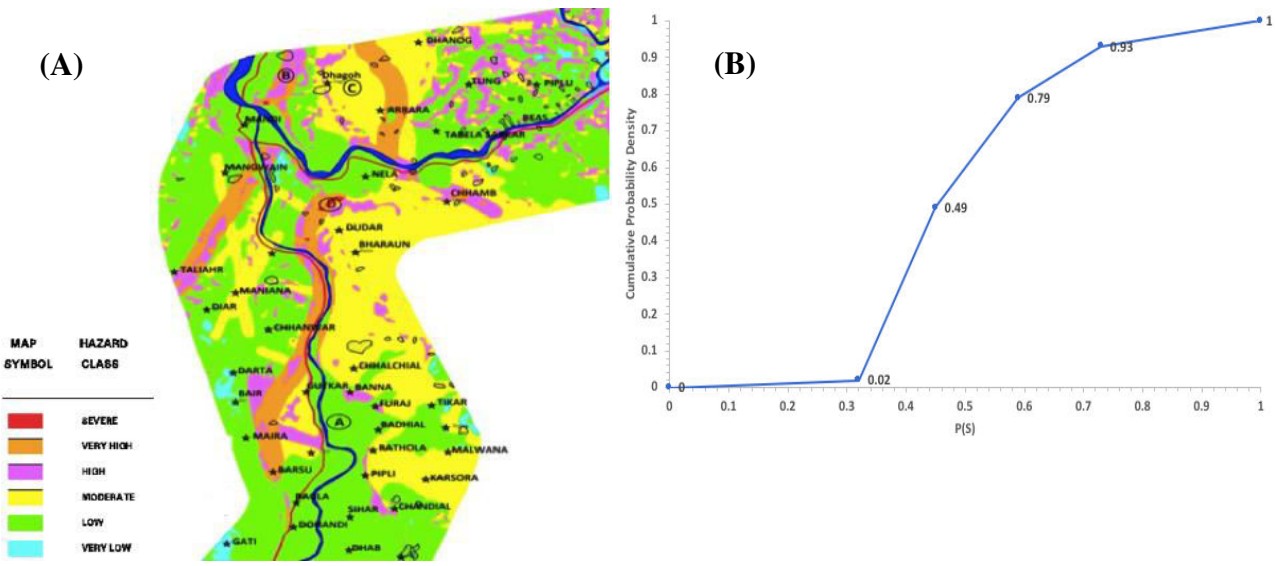

 **Figure 4 (A):** Landslide hazard map of study area. **(B):** The cumulative density function of the spatial probability of
landslides (*P(S)*). The P(S) is shaped by geological and other causative factors in the study area.
*Table 1. Total Estimated Hazard (THED) scale for evaluating the susceptibility of an area to*
*landslides across to different hazard classes*

| Hazard Zone | Range of corrected THED | Hazard class |
|---|---|---|
| I | THED < 3.5 | Very low hazard (VLH) zone |
| II | $3.5 \leq$ THED < 5.0 | Low hazard (LH) zone |
| III | $5.0 \leq$ THED $\leq 6.5$ | Moderate hazard (MH) zone |
| IV | $6.5 <$ THED $\leq 8.0$ | High Hazard (HH) zone |
| V | THED > 8.0 | Very high hazard (VHH) zone |


First, from Table 1, the critical THED values (e.g., 3.5, 5.0, 6.5, and 8.0) were converted into a
probability value by dividing with the highest THED value (= 11.0). Next, we used the LHZ map
of the study area (Figure 4A) to find the surface area that was under a hazard class (very low,
low, moderate, high, and very high) and used this area to determine the cumulative probability
density function for *P(S)*. For example, if a THED of 3.5 (low hazard class) has a 20% coverage
area on LSZ (Figure 4A), then the spatial probability is less than equal to 0.32 (=3.5/11.0) with a
20% chance. Similarly, if a THED of 5.0 (moderate hazard class) has a 30% coverage area on
LSZ, then the then the spatial probability is less than equal to 0.45 (=5.0/11.0) with a 50%
chance (30% + 20%). Such calculations enabled us to develop a cumulative density function for
*P(S)* (see Figure 4B). As shown in Figure 4B (the cumulative density function of *P(S)*), 1.94%
area belonged to the very low hazard class (*P(S)* from 0/11 to 3.5/11), 46.61% area belonged to
the low hazard class (*P(S)* from 3.5/11 to 5.0/11), 30.28% area belonged to the moderate hazard
class (*P(S)* from 5.0/11 to 6.5/11), and 13.71% area belonged to the high hazard class (*P(S)* from
6.5/11 to 8.0/11), and 7.43% area belonged to the very high hazard class (*P(S)* from 8.0/11 to

313 11/11).

In the ILS tool, using Figure 4B, we used a randomly determined point value of the *P(S)*
from its cumulative density function for each participant in the ILS tool (see Figure 4B). This
*P(S)* value stayed the same for participants across their performance in the ILS tool. Please note

that this exercise was not meant to accurately determine the spatial probability of landslide in the area of interest, where more accurate and advanced methods could be used. Rather, the primary objective of this exercise was to develop an approximate model that could account for the spatial probability in the ILS based upon the LHZ map and THED scale (the ILS tool was primarily meant to improve people's understanding about landslide risks and not for physical modeling of landslides).

### 3.1.3 Damages due to landslides

As suggested by Chaturvedi et al. (2017), the damages caused by landslides were classified into three independent categories: property loss, injury, and fatality. These categories have their own damage probabilities. When a landslide occurs, it could be harmless or catastrophic. A landslide becomes catastrophic with damage probability value of property loss, injury, and fatality. Thus, once a uniformly distributed random number is less or equal to the probability of the corresponding damage, then the corresponding damage is assumed to occur in ILS tool. Landslide damages have different effects on the player's wealth and income, where damage to property affects one's property wealth and damages concerning injury and fatality affect one's income level. When the landslide is harmless, then there is no injury, no fatality, and no damages to one's property. For calculation of the damage probabilities due to landslides, data of 371 landslide events in India over a period of about 300 years was used (Parkash, 2011). If we consider the entire 300-year period, then one could expect very different socio-economic conditions to prevail over this period. However, it is to be noted that, in this paper, we vary this probability in the experiment. Thus, the exact value of the probability from literature is not required in the simulation. The exact assumptions about damages are detailed ahead in this paper.

### 4 Interactive Landslide Simulator (ILS) tool

The ILS tool (Chaturvedi et al., 2017) is a web-based tool and it is based upon the ILS model described above. The ILS tool was coded in open-source programming languages PHP and MySQL and it is freely available for use at the following URL: www.pratik.acslab.org. The ILS tool allows participants to make repeated monetary investment decisions for landslide risk-mitigation, observe the consequences of their decisions via feedback, and try new investment

decisions. This way, ILS helps to improve people's understanding about the causes and
consequences of landslides. The ILS tool can run for different time periods, which could be from
days to months to years. This feature can be customized in the ILS tool. However, in this paper,
we have assumed a daily time-scale to make it match the daily probability of landslides
computed in equations 4a and 4b.

The goal in ILS tool is to maximize one's total wealth, where this wealth is influenced by

one's income, property wealth, and losses experienced due to landslides. Landslides and
corresponding losses are influenced by physical factors (spatial and temporal probabilities of
landslides) and human factors (i.e., the past contributions made by a participant for landslide
mitigation). The total wealth may decrease (by damages caused by landslides, like injury, death,
and property damage) or increase (due to daily income). While interacting with the tool, the
repeated feedback on the positive or negative consequences of their decisions on their income
and property wealth enables participants to revise their decisions and learn landslide risks and
dynamics over time.

Figure 5 represents graphical user interface of ILS tool's investment screen. On this

screen, participants are asked to make monetary mitigation decisions up to their daily income
upper bound (see Box A). The total wealth is a sum of income not invested for landslide
mitigation, property wealth, and total damages due to landslides (see Box B). As shown in Box
B, participants are also shown the different probabilities of landslide due to human and physical
factors as well as the probability weight used to combine these probabilities into the total
probability. Furthermore, as shown in Box C, participants are graphically shown the history of
total probability of landslide, total income not invested in landslides, and their remaining
property wealth across different days. As part of the instructions, the players were told that the
mitigation measures will be taken close to the places where they reside in the district in the ILS
tool.


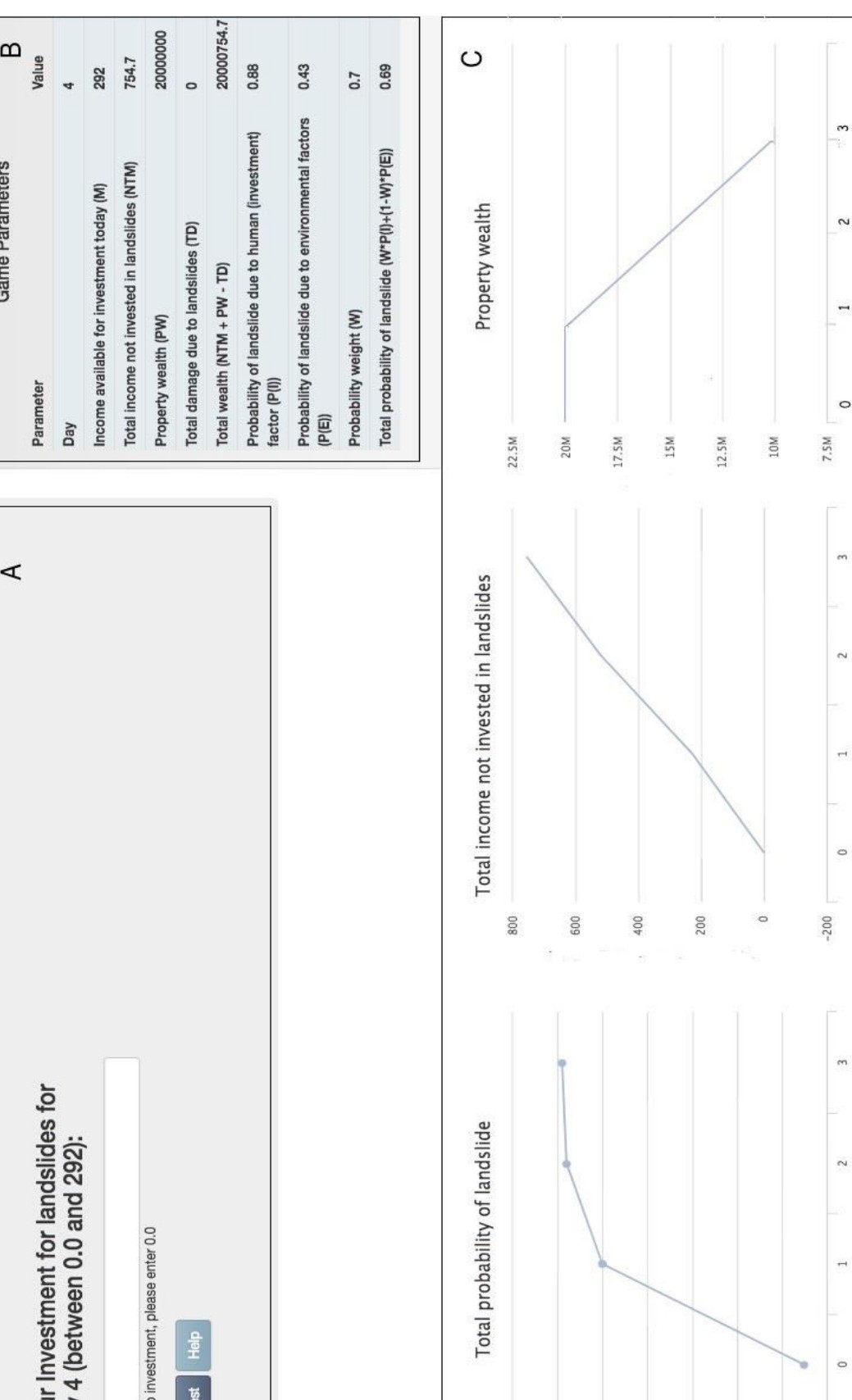

**Figure 5.** ILS tool's Investment Screen. Box (A): The text box where participants made investments against landslides. Box (B): The tool's different parameters and their values. Box (C): Line graphs showing the total probability of landslide, the total income not invested in landslides, and the property wealth over days. Horizontal axes in these graphs represents number of days of performance in the ILS tool. The goal was to maximize Total Wealth across a number of days. This figure is adapted from Chaturvedi et al. (2017).

As described above, participants, i.e., common people residing in the study area, could invest
between zero (minimum) and player's current daily income (maximum). Once the investment is
made, participants need to click the "Invest" button. Upon clicking the Invest button, participants
enter the experiential feedback screen where they can observe whether a landslide occurred or
not and whether there were changes in the daily income, property wealth, and damages due to the
landslide (see Figure 6). As discussed above, the landslide occurrence was determined by the
comparison of a uniformly distributed random number in [0, 1] with $P(T)$. If a uniformly
distributed random number in [0, 1] was less than or equal to $P(T)$, then a landslide occurred;
otherwise, the landslide did not occur. Furthermore, if the landslide occurred, then three
uniformly distributed random numbers in [0, 1] were compared with the probability of injury,
fatality, and property damage, respectively. If the values of any of these random numbers were
less than or equal to the corresponding injury, fatality, or property-damage probabilities, then the
landslide was catastrophic (i.e., causing injury, fatality, or property damage; all three events
could occur simultaneously). In contrast, if the random numbers were more than the
corresponding injury, fatality, and property-damage probabilities, then the landslide was
harmless (i.e., it did not cause injury, fatality, and property damage). As shown in Figure 6A,
feedback information is presented in three formats: monetary information about total wealth (box
I), messages about different losses (box I), and imagery corresponding to losses (box II). Injury
and fatality due to landslides causes a decrease in the daily income and damage to property
causes a loss of property wealth (the exact loss proportions are detailed ahead). If a landslide
does not occur in a certain trial, a positive feedback screen is shown to the decision maker (see
Figure 6B). The user can get back to investment decision screen by clicking on "Return to
Game" button on the feedback screen.








**(A)** Negative Feedback

⚠ **Landslide Occurred!**

**I**

You made **56** investment.

**Your friend invested: 161**

Fortunately, no one in your family died.

Thus, your daily income was not affected and stays at the same value.

Fortunately, no one in your family was injured.

Thus, your daily income was not affected and stays at the same value.

Sorry, your house was destroyed by the debris. Total damage occurred is **10000000**.

Thus, your property wealth is **10000000**.

Your total wealth is **10000631.4**.

Return To Game

**II**



**(B)** Positive Feedback

> ☺ Landslide did not Occur!
>
> You made **180** investment.
>
> **Your friend invested: 172**
>
> Thus, your income stays at **262.8**.
>
> Thus, your property wealth stays at **5000000**.
>
> Your total wealth is **5000777**.
>
> Return To Game

**Figure 6.** ILS tool's feedback screens. **(A)** Negative feedback when a landslide occurred. Box (I) contains the loss in terms of magnitude and messages and Box (II) contains associated imagery. **(B)** Positive feedback when a landslide did not occur.

## 5    Methods

To test the effectiveness of strength and availability of feedback, we performed a laboratory experiment involving human participants where we compared performance in the ILS tool in the presence or absence of experiential feedback about different damage probabilities. Based upon prior literature (Baumeister et al., 2007; Dutt and Gonzalez, 2012; Finucane et al., 2000; Knutty, 2005; Reis and Judd, 2013; Wagner, 2007), we expected the proportion of investments to be higher in the presence of experiential feedback compared to those in the absence of experiential feedback. Furthermore, we expected higher investments against landslides when feedback was more damaging in ILS compared to when it was less damaging (Chaturvedi et al., 2017; Dutt and Gonzalez, 2011; Gonzalez and Dutt, 2011a).

### 5.1    Experimental Design

Eighty-three participants were randomly assigned across four between-subjects conditions in the ILS tool, where the conditions differed in the strength of experiential feedback (high-damage (N= 40) or low-damage (N= 43)) and availability of feedback (feedback-present (N= 43) or feedback-absent (N= 40)) provided after every mitigation decision. An experiment involving the high-damage feed-present condition (N = 20) and the low-damage feedback-present condition (N = 23) in the ILS tool was reported by Chaturvedi et al. (2017). This data has been included in this paper with two more conditions, the high-damage feedback-absent (N = 20) and the low-damage

feedback-absent (N = 20). Data in all four conditions was collected simultaneously. They were
asked to invest repeatedly against landslides across 30-days. In feedback-present conditions,
participants made investment decisions on the investment screen and then they received feedback
about the occurrence of landslides or not on the feedback screen. Participants were also provided
graphical displays showing the total probability of landslides, the total income not invested in
landslides, and the property wealth over days. Figures 5 and 6 show the investment and feedback
screen that were shown to participants in the feedback-present conditions. In feedback-absent
conditions, participants were given a text description and they made an investment decision,
however, neither they were shown the feedback screen nor they were shown the graphical
displays on the investment screen. Thus, in the feedback-absent condition, although participants
were provided with the probability of damages due to landslides and the results of 0% and 100%
investments as a text description, however, they were not shown the feedback screen as well as
the graphical displays on the investment screen. The text description and investment screen
shown to participants in the feedback-absent conditions is given as Appendix 'A'. In high-
damage conditions, the probability of property damage, fatality and injury on any trial were set at
30%, 9%, and 90%, respectively, over 30-days. In low-damage conditions, the probability of
property damage, fatality and injury on any trial were set at 3%, 1%, and 10%, respectively, over
30-days (i.e., about $1/10^{th}$ of its values in the high-damage condition). Across all conditions,
participants made one investment decision per trial across 30-days (this end-point was unknown
to participants). Participants' goal was to maximize their total wealth over 30-days. Across all
conditions, only 1-landslide could occur on a particular day. The nature of functional forms used
for calculating different probabilities in ILS were unknown to participants.
The proportion of damage (in terms of daily income and property wealth) that occurred in an
event of fatality, injury, or property damage was kept constant across 30-days. The property
wealth decreased to half of its value every time property damage occurred in an event of a
landslide. The daily income was reduced by 10% of its latest value due to a landslide-induced
injury and 20% of its latest value due to a landslide-induced fatality. The initial property wealth
was fixed to 20 million EC, which is the expected property wealth in Mandi area. To avoid the
effects of currency units on people's decisions, we converted Indian National Rupees (INR) to a
fictitious currency called "Electronic Currency (EC)," where 1 EC = 1 INR. The initial per-trial
income was kept at 292 EC (taking into account the GDP and per-capita income of Himachal
state where Mandi is located). Overall, there was a large difference between the initial income
earned by a participant and the participant's initial property wealth. In this scenario, the optimal
strategy dictates participants to invest their entire income in landslide protection measures, since
participants' goal was to maximize total wealth. The weight (W) parameter in the equation 1 of
the ILS model was fixed at 0.7 across all conditions. This high value of the W parameter ensured
that participants' investment decisions played a dominant role in influencing the total landslide
probability as per the equation 1. To understand the effect of the W parameter on the total
probability of landslide in ILS, a Monte-Carlo simulation was performed in the ILS model for
different investment conditions over time (see Figure 7A and 7B). It can be seen from both
Figures 7A and 7B, in both the extreme investment conditions over 30-days (i.e., zero
investments and full investments from human players), the value of W determined the range of
possible values of the total probability of landslides, $P(T)$. For example, with a W = 1.0, zero
human investments over a 30-day period caused $P(T) = 1.0$ (a sure landslide) and full
investments caused $P(T) \sim 0.20$ (landslides to be 20% likely to occur). Thus, by keeping a higher
W value, we could ensure that there was a large possible change in the $P(T)$ due to human
actions, giving human participant salient feedback on how their decisions changed $P(T)$. The W
value was set to be 0.70 in the ILS tool and it was shown to participants through the investment
screen on the ILS tool's interface (see Figures 5). Furthermore, the return to mitigation free
parameter (M) was set at 0.8. Again the value of the M parameter ensured that probability of
landslides reduced to 20% (= 1 − M from equation 2) when participants invested their daily
income in full. Participants performed in the ILS for 30-days, starting in mid-July and ending in
mid-August. This period coincided with the period of heavy monsoon rainfall in Mandi area (see
the $P(R)$ peaks in Figure 3). Thus, participants performing in ILS experienced an increasing
probability of landslides due to environmental factors (due to an increasing amount of rainfall
over days). We used the investment ratio as a dependent variable for the purpose of data
analyses. The investment ratio was defined as the ratio of investment made in a trial to total
investment that could have been made up to the same trial. This investment ratio was averaged
across all participants in one case and averaged over all participants and days in another case.
We expected the average investment ratio to be higher in the feedback-present and high-damage
conditions compared to feedback-absent and low-damage conditions. We took an alpha-level
(the probability of rejecting the null hypothesis when it is true) to be 0.05 (or 5%).

(A) (B)

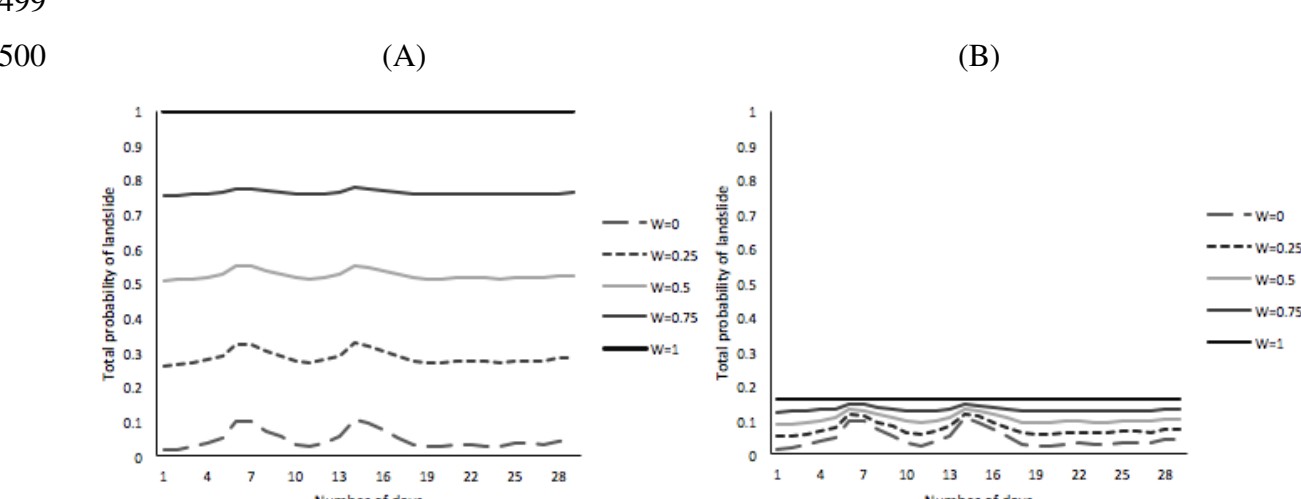

**Figure 7.** Simulation of total probability of landslides in ILS for different values of W in zero investment scenario
**(A)** and full investment scenario **(B)**.

## 5.2 Participants


Participants were recruited from Mandi town via an online advertisement. The research was
approved by the Ethics Committee at Indian Institute of Technology Mandi. Informed consent
was obtained from each participant and participation was completely voluntary. All participants
were from Science, Technology, Engineering, and Mathematics (STEM) backgrounds and their
ages ranged in between 21 and 28 years (Mean = 22 years; Standard Deviation = 2.19 years).
The following percentage of participants were pursuing or had completed different degrees:
6.0% high-school degrees; 54.3% undergraduate degrees; 33.7% Master's degrees; and, 6.0%
Ph.D. degrees. The Mandi area is prone to landslides and most participants self-reported to be
knowledgeable or possess basic understanding about landslides. The literacy rate in Mandi and
surrounding area is quite high (81.5%) (Census, 2011) and our sample was representative of the
population residing in this area. When asked about their previous knowledge about landslides,
2.4% claimed to be highly knowledgeable, 16.8% claimed to be knowledgeable, 57.8% claimed
to have basic understanding, 18.2% claimed to have little understanding, and 4.8% claimed to
have no idea. All participants received a base payment of INR 50 (~ USD 1). In addition, there
was a performance incentive based upon a lucky draw. Top-10 performing participants based
upon total wealth remaining at the end of the study were put in a lucky draw and one of the
participants was randomly selected and awarded a cash prize of INR 500. Participants were told
about this performance incentive before they started their experiment.

**5.3  Procedure**

Experimental sessions were about 30-minutes long per participant. Participants were given instructions on the computer screen and were encouraged to ask questions before starting their study (See Appendix "A" for text of instructions used). Once participants had finished their study, they were asked questions related to what information and decision strategy they used on the investment screen and the feedback screen to make their decisions. Once participants ended their study, they were thanked and paid for their participation.

**6  Results**

**6.1  Investment Ratio Across Conditions**

The data were subjected to a $2 \times 2$ repeated-measures analyses of variance. As shown in Figure 8A, there was a significant main effect of feedback's availability: the average investment ratio was higher in feedback-present conditions (0.53) compared to that in feedback-absent conditions (0.37) ($F(1, 79) = 8.86$, $p < 0.01$, $\eta^2 = 0.10$). We performed analysis of variance statistical tests for evaluating our expectations. The F-statistics is the ratio of between-group variance and the within-group variance. The numbers in brackets after the F-statistics are the degrees of freedom (K-1, N - K), where K are the total number of groups compared and N is the overall sample size. The $p$-value indicates the evidence in favor of the null-hypothesis when it is true. We reject the null-hypothesis when p-value is less than the alpha-level (0.05). The $\eta^2$ is the proportion of variance associated with one or more main effects. It is a number between 0 and 1 and a value of 0.02, 0.13, and 0.26 measures a small, medium, or large correlation between the dependent and independent variables given a population size. The bracket values are indicative of the F-value, its significance and effect size. This result is as per our expectation and shows that the presence of experiential feedback in ILS tool helped participants increase their investments against landslides compared to investments in the absence of this feedback.

As shown in Figure 8B, there was a significant main-effect of strength of feedback: the average investment ratio was significantly higher in high-damage conditions (0.51) compared to that in low-damage conditions (0.38) ($F(1, 79) = 5.46$, $p < 0.05$, $\eta^2 = 0.07$). Again, this result is as per our expectation and shows that high-damaging feedback helped participants increase their investments against landslides compared low-damaging feedback.

Furthermore, as shown in Figure 8C, the interaction between the strength of feedback and
feedback's availability was significant ($F$ (1, 79) = 8.98, $p < 0.01$, $\eta^2$ = 0.10). There was no
difference in the investment ratio between the high-damage condition (0.35) and low-damage
condition (0.38) when experiential feedback in ILS was absent, however, the investment ratio
was much higher in the high-damage condition (0.67) compared to the low-damage condition
(0.38) when experiential feedback in ILS was present (Chaturvedi et al., 2017). Thus, feedback
needed to be damaging in ILS to cause an increase in investments in mitigation measures against
landslides.

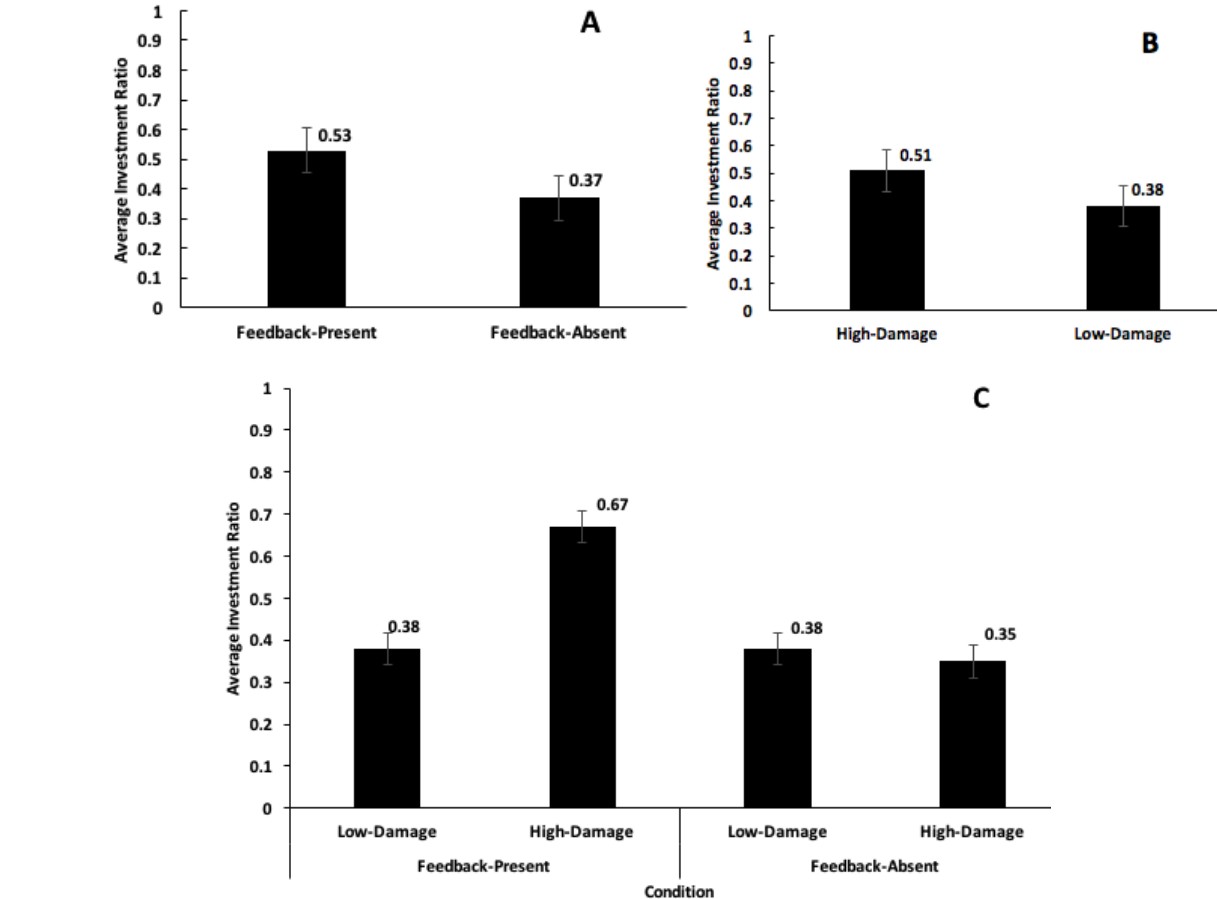


**Figure 8. (A)** Average investment ratio in Feedback-present and Feedback-absent conditions. **(B)** Average
investment ratio in low- and high-damage conditions. **(C)** Average investment ratio in low- and high-damage
conditions with Feedback-present and absent. The error bars show 95% Confidence Interval (CI) around the point
estimate.
**6.2   Investment Ratio Across Days**
The average investment ratio increased significantly over 30-days (see Figure 9A; $F$ (8.18,
646.1) = 8.35, $p < 0.001$, $\eta^2$ = 0.10). As shown in Figure 9B, the average investment ratio
increased rapidly over 30-days in feedback-present conditions, however, the increase was
marginal in feedback-absent conditions ($F$ (8.18, 646.1) = 3.98, $p < 0.001$, $\eta^2$ = 0.05).
Furthermore, in feedback-present conditions, the average investment ratio increased rapidly over
30-days in high-damage conditions, however, the increase was again marginal in the low-damage
conditions (see Figure 9C; $F$ (8.18, 646.1) = 6.56, $p < 0.001$, $\eta^2$ = 0.08). Lastly, as seen in Figure
9D, although there were differences in the increase in average investment ratio between low-
damage and high-damage conditions when experiential feedback was present, however, such
differences were non-existent between the two damage conditions when experiential feedback
was absent ($F$ (8.18, 646.1) = 4.16, $p < 0.001$, $\eta^2$ = 0.05). Overall, ILS performance helped
participants increase their investments for mitigating landslides when damage feedback was high
compared to low in ILS.

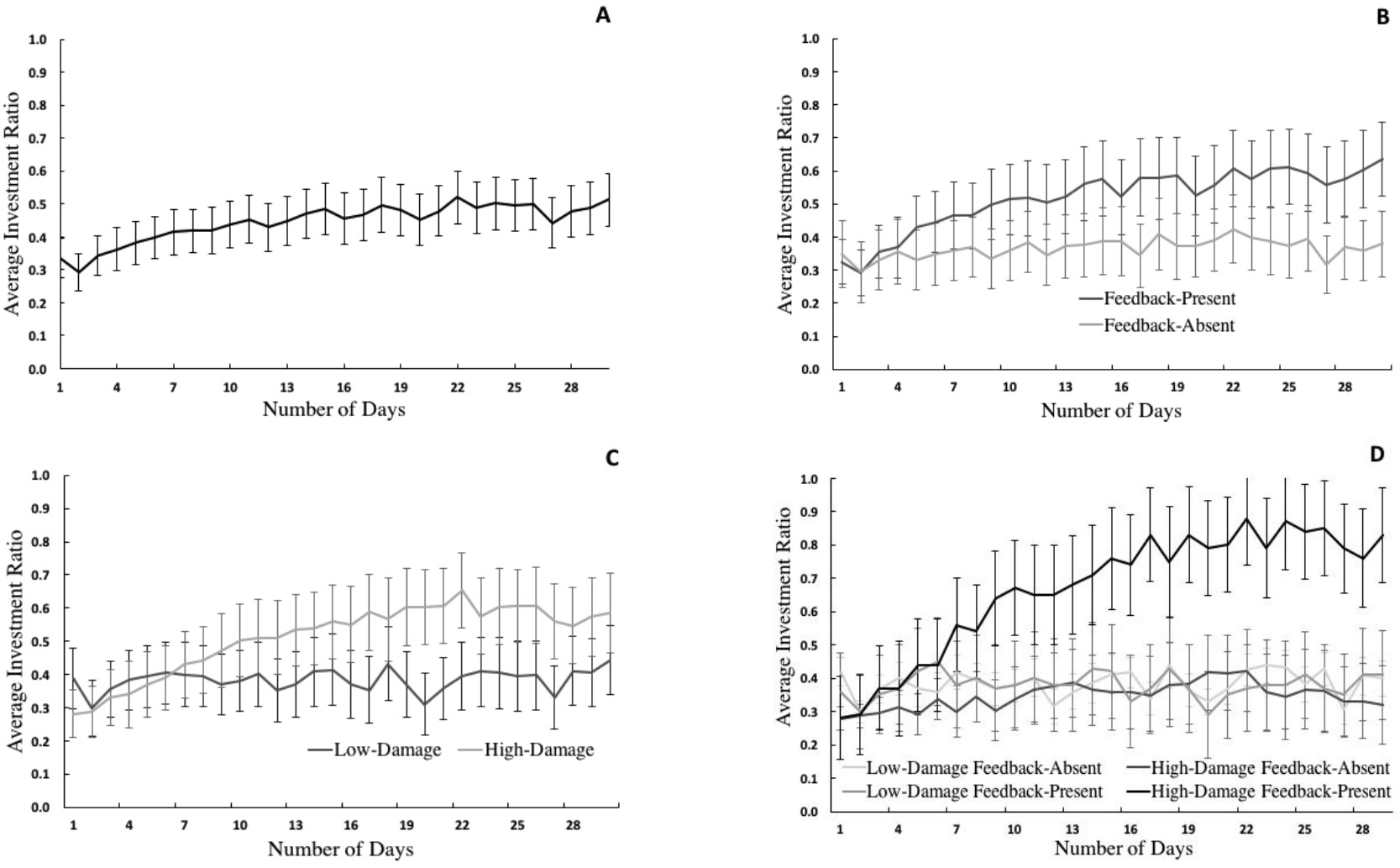

Figure 9. (A) Average investment ratio over days. (B) Average investment ratio over days in Feedback-present and Feedback-absent conditions. (C) Average investment ratio over days in low- and high-damage conditions. (D) Average investment ratio over days in low- and high- damage conditions with Feedback-present or absent. The error bars show 95% CI around the point estim


However, in feedback's absence in ILS, participants were unable to increase their investments for
mitigating landslides, even when damages were high compared to low.
**6.3    Participant Strategies**
We analyzed whether an "invest-all" strategy (i.e., investing the entire daily income in mitigating
landslides) was reported by participants across different conditions. As mentioned above, the invest-all
strategy was an optimal strategy and this strategy's use indicated learning in the ILS tool. Figure 10
shows the proportion of participants reporting the use of the invest-all strategy. Thus, many participants
learnt to follow the invest-all strategy in conditions where experiential feedback was present and it was
highly damaging compared to participants in the other conditions.

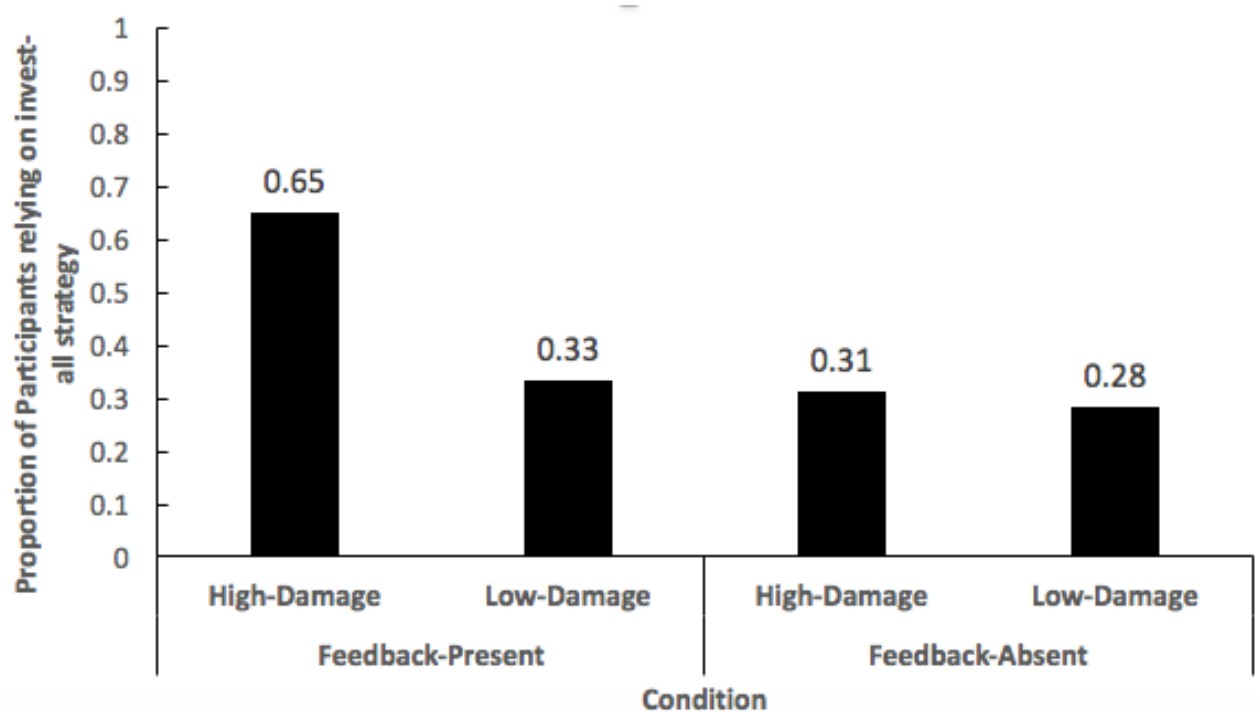


**Figure 10.** The proportion of reliance on the invest-all strategy across different conditions.

**8    Discussion**
In this paper, we used an existing ILS tool for evaluating the effectiveness of feedback in influencing
people's decisions against landslide risks. We used the ILS tool in an experiment involving human
participants and tested how the strength and availability of experiential feedback in ILS helped increase
people's investment decisions against landslides. Our results agree with our expectations: Experience
gained in ILS enabled improved understanding of processes governing landslides and helped
participants improve their investments against landslides.
First, the high-damaging feedback helped increase people's investments against landslides over
time compared to the low-damaging feedback. Furthermore, the feedback's presence helped participants
increase their investments against landslides over time compared to feedback's absence. These results
can be explained by the previous lab-based research on use of repeated feedback or experience
(Chaturvedi et al., 2017; Dutt and Gonzalez, 2010, 2011; Finucane et al., 2000; Gonzalez and Dutt,
2011a). Repeated experiential feedback likely enables learning by repeated trial-and-error procedures,
where bounded-rational individuals (Simon, 1959) try different investment values in ILS and observe
their effects on the occurrence of landslides and their associated consequences. The negative
consequences due to landslides are higher in conditions where the damages are more compared to
conditions where the damages are less. This difference in landslide consequences influences
participants' investments against landslides. According to Slovic et al. (2005), loss-averse individuals
tend to increase their contribution against a risk over time. In our case, similar to Slovic et al. (2005),
participants started contributing slowly against landslides and, with the experience of landslide losses
over time, they started contributing larger amounts to reduce landslide risks.
We also found that the reliance on invest-all strategy was higher in the high-damage and
feedback-present condition compared to the low-damage and feedback-absent condition. The invest-all
strategy was the optimal strategy in the ILS tool. This result shows that participants learned the
underlying system dynamics (i.e., how their actions influenced the probability of landslides) in ILS
better in the feedback-rich condition compared to the feedback-poor condition. As participants were not
provided with exact equations governing the ILS tool and they had to only learn from trial-and-error
feedback, the saliency of the feedback due to messages and images likely helped participants' learning
in the tool. In fact, we observed that the use of the optimal invest-all strategy was maximized when the
experiential feedback was highly damaging. One likely reason for this observation could be the high

educational levels of participants residing in the study area, where the literacy rate was more than 80%. Thus, it seems that participants' education levels helped them make the best use of damaging feedback.

We believe that the ILS tool can be integrated in teaching courses on landslide sustainable practices in schools from kindergarten to standard 12[th]. These courses could make use of the ILS tool and focus on educating students about causes, consequences, and risks of hazardous landslides. We believe that the use of ILS tool will make teaching more effective as ILS will help incorporate experiential feedback and other factors in teaching in interactive ways. The ILS tool's parameter settings could be customized to a certain geographical area over a certain time period of play. In addition, the ILS tool could be used to show participants the investment actions other participants (e.g., society or neighbours). The presence of investment decisions of opponents in addition to one's own decisions will likely enable social norms to influence people's investments and learning in the tool (Schultz et al., 2007). These features makes ILS tool very attractive for landslide education in communities in the future.

Furthermore, the ILS tool holds a great promise for policy-research against landslides. For example, in future, researchers may vary different system-response parameters in ILS (e.g. weight of one's decisions and return to mitigation actions) and feedback (e.g. numbers, text messages and images for damage) in order to study their effects on people's decisions against landslides. Here, researchers could evaluate differences in ILS's ability to increase public contributions in the face of other system-response parameters and feedback. In addition, researchers can use the ILS tool to do "what-if" analyses related to landslides for certain time periods and for certain geographical locations. The ILS tool has the ability to be customized to certain geographical area as well as certain time periods, where spatial parameters (e.g., soil type and geology) as well as temporal parameters (e.g., daily rainfall) can be defined for the study area. Once the environmental factors have been accounted for, the ILS tool enables researchers to account for assumptions on human factors (contribution against landslides) with real-world consequences (injury, fatality, and infrastructure damage). Such assumptions may help researchers model human decisions in computational cognitive models, which are based upon influential theories of how people make decisions from feedback (Dutt and Gonzalez, 2012; Gonzalez and Dutt, 2011b). In summary, these features make ILS tool apt for policy research, especially for areas

that are prone to landslides. This research will also help test the ILS tool and its applicability in different
real-world settings.

**9. Limitations**

Although the ILS tool causes the use of optimal invest-all strategies among people in conditions
where experiential feedback is highly damaging, more research is needed on investigating the nature of
learning that the tool imparts among people. As people's investments for mitigating landslides in ILS
directly influences the risk of landslides due to human and environmental factors, investments indeed
have the potential of educating people about landslide risks. Still, it is important to investigate how
investing money in the ILS tool truly educates people about landslides. We would like to investigate
this research question as part of our future research.

Currently, in the ILS model, we have assumed that damages from fatality and injury to influence
participants' daily-income levels. The reduced income levels do create adverse consequences, but one
could also argue that they would be much less of concern for most people compared to the injury and
fatality itself. Furthermore, people could also choose to migrate from an area when the landslide
mitigation costs are too high, and adaptation becomes impossible, especially due to the differences
between the landslide hazard and other hazards such as flood, drought, and general climate risks. As
part of our future research, we plan to investigate the influence of feedback that causes only injuries or
fatalities in ILS compared to the feedback that causes economic losses due to injuries and fatalities.
Also, as part of our future research in the ILS tool, we plan to investigate people's migration decisions
when the landslide mitigation costs are too high and adaptation to landslides is not possible.

In this paper, our primary objective was not to accurately predict rainfall or other landslide
parameters; rather, it was to educate people about landslide disasters. Thus, we have used approximate
models of real landslide phenomena in the ILS simulation tool. The use of approximate models is in line
with a large body of literature on using simulation tools for improving people's understanding about
natural processes like climate change and other natural disasters (Dutt and Gonzalez, 2010, 2011;
Finucane et al., 2000). As part of our abstraction, we may have missed certain aspects related to the
sensitivity of the different social classes to their economic and cultural resources. In future, we would
like to compare the proportion of investments in different experimental conditions to people's likely

socio-economic cost thresholds given that people may need to spend their wealth in other areas beyond landslide mitigation.

Furthermore, we used a linear model to compute the probability of landslides due to human factors in the ILS tool. Also, the probabilistic equations governing the physical factors in the ILS model were not disclosed to participants, who seemed to possess high education levels. One could argue that there are several other linear and non-linear models that could help compute the probability of landslides due to human factors. Some of these models may also influence the probability of landslides and the severity of consequences (damages) caused by landslides. Also, other more generic models could account for the physical factors in the ILS tool. We plan to try these possibilities as part of our future work in the ILS tool. Specifically, we plan to assume different models of investments in the ILS tool and we plan to test them with participants possessing different education levels.

In the current experiment, we assumed a large disparity between a participant's property wealth and her daily income. In addition, as part of the ILS model, we did not consider support from governments or insurance companies against landslide damages. In India, people mostly use their own finances to overcome the challenges put by natural disasters as insurance or other public methods have only shown limited success (ICICI, 2018). However, in certain cases, especially in developing countries, mitigation of landslide risks may often be financed by the government or international agencies. As part of our future work, we plan to extend the ILS model to include assumptions of contributions from government and other international agencies. Such assumptions will help us determine the willingness of common people to contribute against landslide disasters, which is important as the developing world becomes more developed over time.

To test our hypotheses, we presented participants with a high damage scenario and a low damage scenario, where the probabilities of property damage, injury, and fatality were high and low, respectively. However, such scenarios may not be realistic, where people may want to migrate from both low and damage areas in even the least developed countries. In future research with ILS, we plan to calibrate the probability of damages, injury, and fatality to realistic values and then test the effectiveness of ILS in improving decision making.

Furthermore, in our experiment, when landslide did not occur and experiential feedback was present, people were presented with a smiling face followed by a message. The message and emoticon were provided to connect the cause-and-effect relationships for participants in the ILS tool. However, it could also be that a landslide did not occur on a certain trial due to the stochasticity in the simulation rather than participants' investment actions. Although such situations are possible over shorter time-periods, over longer time-periods increased investments from people will only reduce the probability of landslides. Also, there is a possibility that the participant demographics in the experiment may not be representative of the study area. Thus, as part of future research, we plan to control the participant sample in different ways and test the effects that demographics produces on people's investments.

In this paper, the experiment used a daily investment setting in the ILS tool. However, the ILS tool can easily be customized to different time periods ranging from seconds, minutes, hours, days, months, and years. As part of our future research, we plan to extend the daily assumption by considering people making decisions on longer time-scales ranging from months to years. In addition, in the experiment, we assumed a value of 0.7 and 0.8 for the weight (W) and return to mitigation (M) parameters, respectively. These W and M values indicated that landslide risks could largely be mitigated by human actions. However, this assumption may not be the case always, especially for mitigation measures like tree plantations. For example, afforestation alone may not help in reducing deep-seated landslides in hilly areas (Forbes, 2013). Thus, it would be worthwhile to investigate as part of future research on how people's decision-making evolves in conditions where investments likely influence the landslide probability (higher values of W and M parameters) compared to conditions where investments unlikely influence the landslide probability (lower values of W and M parameters). Some of these ideas form the immediate next steps in our on-going research program on landslide risk communication.

## 10. Conclusions

It can be concluded from this preliminary research that simulation tools like ILS that provide feedback about the outcomes of landslide disasters influence people's investment decisions against

landslides. Given our results, we believe that ILS could potentially be used as a landslide-education tool for increasing public understanding about landslides among the adult population.

This work forms a good preliminary example for researchers involved in gamification and participative processes in case of landslide disasters. However, this research work is preliminary in nature and we plan to deepen it in the near future. To examine the full potential of ILS in influencing people's perceptions of landslide risk, lot of experiments manipulating system variables, feedback strengths, and severity of damages need to be conducted on a bigger population across several study areas. Another line of research could be to understand the people's behaviour or decision-making style in landslide scenarios by fitting computational cognitive models to the human data. The ILS tool can also be used by policymakers to do what-if analyses in different scenarios concerning landslides. However, the assumptions in the ILS tool should be evaluated in the study area before it is released for policy research.

*Data availability*. Data used in this article have not been deposited to respect the privacy of users. The data can be provided to readers upon request.

*Author contributions.* AA developed the ILS tool under guidance from PC and VD. AA and PC collected the data in the study. PC and VD analysed the data and prepared the manuscript. PC and VD revised the manuscript as per referee comments.

*Competing interests.* The authors declare that they have no conflict of interest.

*Acknowledgements.* This research was partially supported by the following grants to Varun Dutt: a grant from Himachal Pradesh State Council for Science, Technology and Environment (grant number: IITM/HPSCSTE/VD/130); a grant from National Disaster Management Authority (grant number: IITM/NDMA/VD/184); and, a grant from Defence Terrain Research Laboratory, Defence Research and Development Organization (grant number: IITM/DRDO-DTRL/VD/179). We thank Akanksha Jain and

Sushmita Negi, Centre for Converging Technologies, University of Rajasthan, India for providing
preliminary support for data collection in this project.

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

**Appendix A**
**Instructions of the Experiment**
Welcome!
You are a resident of Mandi district of Himachal Pradesh, India, a township in the lap of Himalayas.
You live in an area that is highly prone to landslides due to a number of environmental factors (e.g., the
prevailing geological conditions and rainfall). During the monsoon season, due to high intensity and
prolonged period of rainfall, a number of landslides may occur in the Mandi district. These landslides
may cause fatalities and injuries to you, your family, and to your friends, who reside in the same area. In
addition, landslides may also damage your property and cause loss to your property wealth.
This study consists of a task, where you will be making repetitive decisions to invest money in order to
mitigate landslides. Every trial, you'll earn certain money between 0 and 10 points. This money is
available to you to invest against landslides. You may invest certain amount from the money available
to you; however, if you do not wish to invest anything, you may invest 0.0 against landslides on a
particular trial. Based upon your investment against landslides, you'll get feedback on whether a
landslide occurred and whether there was an associated loss of life, injury, or property damage (all three
events are independent and they can occur at the same time).
**Your total wealth at any point in the game is the following: sum of the amounts you did not invest**
**against landslides across days + your property wealth - damages to you, your family, your friends,**
**and to your property due to landslides**. Your property wealth is assumed to be 100 points at the start
of the game. The amount of money **not invested against landslides** increases your total wealth. **Your**
**goal is to maximize your total wealth in the game**.
Whenever a landslide occurs, if it causes fatality, then your daily earnings will be reduced by 5% of its
present value at that time and if landslide causes injury to someone, then the daily earnings willbe
reduced by 2.5% of its present value at that time. Thus, the amount available to you to invest against
landslides will reduce with each fatality and injury due to landslides. Furthermore, if a landslide occurs
and it causes property damage, then your property wealth will be reduced by 80% of its present value at
that time; however, the money available to you to invest against landslides due to your daily earnings
will remain unaffected.

Generally, landslides are triggered by two main factors: environmental factors (e.g., rainfall; outside one's control) and investment factors (money invested against landslides; within one's own control). The total probability of landslide is a weighted average of probability of landslide due to environment factors and probability of landslide due to investment factors. The money you invest against landslides reduces the probability of landslide due to investment factors and also reduces the total probability of landslides. However, the money invested against landslides is lost and it cannot become a part of your total wealth.

At the end of the game, we'll convert your total wealth into INR and pay you for your effort. For this conversion, a ratio of 100 total wealth points = INR 1 will be followed. In addition, you will be paid INR 30 as base payment for your effort in the task. Please remember that your goal is to maximize your total wealth in the game.

Starting Game Parameters

Your wealth: **20 Million**

When a landslide occurs:

If a death occurs, your daily income will be reduced by **50**% of its current value.

If an injury takes place, your daily income will be reduced by **25**% of its current value.

If a property damage occurs, your wealth will be reduced by **50**% of your property wealth.

**Best of Luck!**