# Peer review of "Learning in an Interactive Simulation Tool against Landslide"

_Natural Hazards and Earth System Sciences, 2017_

## Referee Comment (RC1) · Anonymous Referee #1 · 9 Oct 2017

General comments:

The paper deals with a very relevant topic, the involvement of stakeholders in landslide risk management and the adoption of "gamification" type approaches to promote it. The ILS software results a promising tool for capturing the interest of attendees and it could be applied with reduced effort to other test cases. The sections 3 and 4 show in effective ways procedures and results. However, several elements would require further examination. First, the test case is not adequately introduced: geology, past and recent events, rainfall patterns recognized as main triggers. In this regard, also in ILS, dynamics inducing the events (physical or anthropic) are not adequately taken

into account. For example, it is not clear how the spatial distribution of landslide events is accounted for in ILS or if the information about occurrence probability are used in simulation. The role of "anthropic activities" on slopes could often be detrimental and the reduction in earnings due to reducing these one for preserving stability should be taken into account. Moreover, the main stakeholders for ILS are probably not citizens but policy makers and administrators and then financial management (daily income) should be revised accordingly. The timescales also for simulations does not appear adequate. Several decisions and protection measures need substantial longer times. Timing for measure implementation could be crucial for deciding the more effective strategies. Finally, the references in first part should be extended and updated. Under such constraints, a substantial revision (major revision) of the text should be performed in order to address the issues arisen above (and below) on specific items; on the other side, the text could be rearranged only to promote the general approach and followed procedures and main results stressing the role that it could cover for landslide risk management after proper characterizations of areas of interest.

Specific issues: Abstract:

-rephrase the first sentence; the verb appears missing

Introduction

L25-27: please give further details; in my view, "Knowledge about causes-and-consequences of landslides and awareness about landslide disaster mitigation" act in different ways; the first one supporting structural protection measurements could re-duce the occurrence/magnitude of landslides. The other one tends reducing people and assets vulnerability not varying the physical processes inducing them

L31-33: please add further details about Early Warning System tools; e.g. you could refer to reviews available in literature.

L71: "Chaturvedi et al. (2016)" reference is missing in the list

L82-83: please consider, I'm not sure that "increasing the amount of damage feedback" and " increasing the probabilities of landslide damages" could be assumed equivalent

2 Computational model of landslide risk

L106-108 (Figure 1): for landslides, the issue could be quite more challenging; indeed, you should consider "human interventions" detrimental for slope stability. For example, land use/cover changes (e.g. deforestation, conversion to agricultural practices). In this regard, rainfall required to induce the phenomena (e.g. duration, intensity) could be affected by "human interventions". Furthermore, researchers monitor data for landslide occurrence but not determine them as for "user" with investments. Finally, both influence not only the hazard ("total probability of landslide") but the risk.

L109: please specify if you consider weather(rainfall)-induced landslides

L128: the main part of investments for protection measurements as structural (e.g. drainages, retention walls) as soft (e.g. EWS) are funded by Administrations (National, Regional and Local); in which ways it is accounted for?

Section 2.1.2: further clarifications are needed. Firstly, brief information about the landslides in the area of interest are required; indeed, the relevance of antecedent precipitations is strictly linked to several geomorphological factors (e.g. soil depth, bottom boundary conditions, hydraulic and mechanical properties); without them, it is not possible to evaluate if considered durations (1d, 3d, 30d) are proper. Moreover, it is not clear the role of "Landslide Susceptibility Zonation"; indeed, "susceptibility" does not provide details about frequency of phenomena but attempts defining the area more "vulnerable" to the events while in this case it is intended providing also Hazard. Moreover, please add details about the rating (0-11). Finally, all the slopes in the area are recognized to be affected by the same rainfall patterns (similar properties, similar soil depths and so on)?

L170: what do you intend for landslide "benign"?

Section 2.1.3: please, what do you intend for "random numbers"? which ways are the three damage probabilities computed in?

Section 2.2: why do you consider a daily time step? Several decisions and protection measures need substantial longer times. Timing for measure implementation could be crucial for deciding the more effective strategies

L205: who is the reference stakeholder of interest? Citizens, administrators, policy makers.

L212-213: in ILS, how is it decided if, for a certain day, landslide could occur or not?

3 Experiment

L289-295: I am not sure that the sample composition is consistent with those of communities living in the area affected by landslides as in terms of background as in terms of age. It could deeply affect the findings and the generalization of the results also taking into account the very interesting issues arisen in L44-47

L302: It is quite equal to what reported in L287; in my view, it could be removed

L313: please, provide further details about the symbols reported in brackets

L374: what do you intend for "K-12"?

L457: Mathew et al. reference should be moved in proper alphabetical order

Appendix A

It reports information quite similar to those in Figure 4; for these reason, it could be removed

---

## Referee Comment (RC2) · Anonymous Referee #2 · 29 Oct 2017

The manuscript presents an interesting tool for testing the people's propensity to invest money for protecting goods and life from landslides. The tool has been applied for analyzing the effect of feedbacks availably in influencing the people's decision-making process when asked to invest resources for landslide protection. The topic of the manuscript fit into the scopes of the NHESS Journal since it deals with the design and implementation of mitigation and adaptation strategies to reduce the impact of hazardous natural events on human-made structures, infrastructure, and life.

General comments

The structure of the paper is fair and, even if I'm not a native English speaker, I found

the paper understandable. However, I think that some improvements can be made simplifying the sentences and re-phrasing some frequent constructs as "Although . . ... ; however . . . .", where the semicolon do not help to understand the sentence.

I suggest to promote the section "Interactive landslide Simulator (ILS) tool" from the level of a subsection to the level of a section. Currently it is, erroneously, inside the "Computational model of landslide risk" section. More in general I would also suggest to the author to use the common scientific structure which includes "Introduction", "Material and Methods", "Results", "Discussion" (currently discussion and results are in the same section).

I think that the ILS tool is very interesting but I see a major problem in the paper: there is not the possibility to test the ILS tool. My opinion is that, according to the open science, open data, open knowledge concepts, researchers should be put in condition of evaluating the ILS tool. From the paper it is not clear if the tool is a web application or a standalone program and there is not a description of the technology adopted for implementing it, nor of the intention of the authors of releasing the code and, if this is the case, adopting which license.

Another issue is about the significance of the results of the experiment. Evidences are that people using the ILS tool with feedbacks, rapidly understand that the best strategy to "win the game" is to invest the entire daily income in landslides mitigation measures. Even if this is interesting, the authors do not comment or discuss the fact that the population of the participants is made of people having high to very high educational levels. This can have a strong effect on their capacity to rapidly find the best strategy. This is particularly true where one considers that, as far as I know, the educational levels of people living in the Himalaya region is mostly low and very low. I think that representativeness of the participants to the experiment should be discussed more in detail.

Lastly: figures are enough rough and should be improved and better described in the

captions.

Specific comments

L138: I think you need to add that $0<M<1$

L162: It is not clear to me what the Total Estimated Hazard is. Please define it.

L162: Landslide Hazard Map: what is this? Not clear how this is related to the LSZ and to the THED. It is even not clear how the spatial probability is included in the tool. It is a single value or there is a map?

L172: Is "become less than" correct? I suppose should be "become greater than". If not please try to explain why must be "less".

L182: please change "their total wealth" with "the total wealth of the partecipants"

L207: "decision-maker". Are you meaning "partecipant"? If yes please change the text accordingly.

L241-243: the sentence is not clear. Please rephrase.

L243: "see Figure 2": please explain how the figure helps in understanding the text.

L262: "(W)": it is not immediate to understand that "W" is the parameter of the equation at page 4. Please number the equations and use those numbers in the text.

L263: "was fixed to 0.8": in figure 2, W is 0.7.

L302: the first sentence was already stated at the start of the 3.2 subsection.

L313: please describe the meaning of the statistical parameters (derived from statistic tests) inside the brackets.

L330: what "CI" means?

L385-386: unclear. Please rephrase.

---

## Referee Comment (RC3) · Anonymous Referee #3 · 8 Nov 2017

General Comments:

The study described in the paper addresses an important and very relevant issue in natural disaster risk management – to explore potential ways to improve risk awareness and knowledge. The authors reported how they used feedback in an Interactive Landslide Simulator to influence people's risk reduction investment behavior. The manuscript was written generally in good English that can be relatively easily understood, but the ILS model still needs to be better elaborated and explained. While the study represent a good initiative, it also suffers from a number of design problems.

Specific issues:

[Figure]

1. The ILS model and simulator structure Significant information about the ILS model was from the authors' published conference paper in 2016. The authors need clearly state this. Much of the information needs not to be repeated. Even so, the current description of the model is still not clear enough. More details are needed to help understand how the rather sophisticated landslide probability calculation relates to damage estimation. For example, the total P is an additive results of the two constituting components, P(I) and P(E), however P(E) is the multiplicative results of its two constituting components. The authors did not give full information to justify this choice. The authors mentioned "study area" only in 2.1.1, while very limited information was provided. The authors also did not give any explanation on how W is determined.

2. The assumptions of the ILS model

The ILS was designed with the assumption that people susceptible to landslide hazard aims to maximize their total wealth and the authors started that "a high probability of landslide damages will make people suffer monetary losses and people would tend to minimize these losses by increasing their mitigation actions". This assumption neglects much of the social science research on people's risk perception, attitude and behavior, that people do not behave as an economic rationale individual in the face of extreme events. The authors assumed that "damages concerning injury and fatality affect one's income levels". This is rather naïve. While reduced income level is going to be a consequence, but it would be much less a concern for most people than the injury and fatality itself. In reality people can also choose to migrate when mitigation cost is too high and adaptation becomes impossible. The nature of landslide hazard, including its notorious fame of being extremely hard, if not impossible, to predict, makes it quite different from other hazards such as flood and drought, and general climate risk. The authors' choice of P(I) formula from Hasson et al. 2010 does not seem to be appropriate.

It may seem to be obviously useful by applying specific parameters from the Mandi area in India as the participants seem to be mostly from the area (the authors did not clearly elaborate this), however, since the algorithms was not disclosed to the partic-

ipants and a random number generator was used in producing damages, using the seemingly sophisticated algorithms is in fact not much related to the authors' main objective, instead, a more generic algorithm would serve the same purpose and potential be more useful for testing with participants from other areas.

Day was used as the time unit for simulation and people make daily choice in landslide mitigation investment. This is not relevant for real world situation either. In most cases, especially in developing countries, households and communities themselves almost never have resources substantial enough to mitigate landslide risk, which is often financed by government and/or international donors. The huge disparity between the average asset (calculated as per capital GDP) and the salary (with the former being 2000 times of a person's annual income) also supported my above statement.

The authors chose a value of 0.8 for W, indicating that the landslide risk can largely be mitigated by human. This is in general not the case, especially for the type of mitigation measures mentioned by the authors – tree plantation. There has been studies showing that afforestation does not help with landslides in similar areas to Mandi in the Sivalik Hills.

3. The study design

The high damage scenario is simply not realistic at all. With such a high risk of mortality and 90% change of injury, no one would still choose to stay in the landslide area, even in least developed countries. The low damage scenario would already be a very high risk area in reality, in any countries.

In Fig. 3b, The authors give a smiling face followed by "Landslide did not Occur". This gives a false feeling that the fact that landslide did not occur because of mitigation investment, while in reality much of it should be due to stochastic in the nature of landslide.

4. The results

First, part of the results were already included in the 2016 paper (apparently including 43 of the 83 participants reported in this study) and this should be fully disclosed. Second, the part of the results on people's increasing investment in mitigation seems to be largely an artifact of the choice of M being 0.8. It'd be more interesting to study, with a much larger sample, how how changing M will affect people's behavior, given that the authors choose more realistic scenarios.

Some detailed comments on texts:

1. In Abstract, the first sentence is incomplete. 2. "Different amount of feedback" was used, but in fact the difference between the two different levels of feedback may better be described as "intensity" of "strength" of feedback. 3. Fig. 2 is similar to the Fig 2 in the authors' 2016 conference paper and needs to be disclosed. 4. Fig. 5b, it should be high/low damage instead of more/less damage. 5. Reference – Mathew et al. was published in 2014 and should be rearranged in alphabetic order.

While the study represents an interesting attempt, it suffers from seriously false model assumptions and weakness in study design in relation to reality. I personally even think that the simulator may falsely influence participants in terms of how they should make decisions in the face of landslide risk. But I strongly recommend the authors to continue developing the simulator with stronger social science understanding and better design.

---

## Author Comment (AC3) · 14 Dec 2017

General Comments:

The study described in the paper addresses an important and very relevant issue in natural disaster risk management – to explore potential ways to improve risk aware-ness and knowledge. The authors reported how they used feedback in an Interac-tive Landslide Simulator to influence people's risk reduction investment behavior. The manuscript was written generally in good English that can be relatively easily under-stood, but the ILS model still needs to be better elaborated and explained. While the study represent a good initiative, it also suffers from a number of design problems.

Authors: Thank you for summarizing our contribution and providing encouragement to our work. We have now made several improvements to the manuscript based upon review comments from you and other reviewers. In agreement with different reviewers, we have now also extended this paper in both the design choices as well as system constraints. Now the elaboration of the ILS model has been improved and we have also explained the experiment design in detail. In the revised manuscript, we have also addressed several design problems related to participant demographics and details concerning assumptions in the ILS tool.

Specific issues:

The ILS model and simulator structure Significant information about the ILS model was from the authors' published conference paper in 2016. The authors need clearly state this. Much of the information needs not to be repeated. Even so, the current description of the model is still not clear enough. More details are needed to help understand how the rather sophisticated landslide probability calculation relates to damage estimation. For example, the total P is an additive results of the two constituting components, P(I) and P(E), however P(E) is the multiplicative results of its two constituting components. The authors did not give full information to justify this choice. The authors mentioned "study area" only in 2.1.1, while very limited information was provided. The authors also did not give any explanation on how W is determined.

Authors: Thank you for your kind comments.

In the revised manuscript, we have now given proper citation to our 2016 conference paper at different places in the manuscript (actually the year of publication of this conference paper is 2017 and not 2016 and the year has been corrected in the manuscript). Furthermore, we have now clarified the contribution in the manuscript and how this work builds upon prior work (pg. 3). In addition, we have now extended the paper to include a better description of relevant theory (pg. 2-3) and a better description of the probability calculation for P(S), P(R), and P(E) (pg. 5-7). As part of our revision,

we have also suggested the rationale for different design and system choices made. In the revised version, we have explained study area by giving more details about its geographic location, climate, and demographic profile (pg. 3, 12-14).

The W is a free parameter. We have fixed the W parameter in this experiment such that human action play a significant role in the reduction of landslide risk (pg. 13). However, as a part of our future research work with ILS tool, we will also vary the M and W parameters to see the effect of this variation on participants' investment decisions against landslides (pg. 20-22).

The assumptions of the ILS model: The ILS was designed with the assumption that people susceptible to landslide hazard aims to maximize their total wealth and the authors started that "a high probability of landslide damages will make people suffer monetary losses and people would tend to minimize these losses by increasing their mitigation actions". This assumption neglects much of the social science research on people's risk perception, attitude and behavior, that people do not behave as an economic rationale individual in the face of extreme events.

Authors: Thank you for providing valuable comments that helped to further improve our research.

First, we have now revised our expectations to be over time (pg. 3). Second, at a first glance, the expectations may seem to assume people to be economically rationale individuals while facing landslide disasters (Bossaerts and Murawski, 2015; Neumann and Morgenstern, 1947), where one disregards people's bounded rationality, risk perceptions, attitudes, and behaviours (De Martino, Kumaran, Seymour, and Dolan; 2005; Gigerenzer and Selten, 2002; Kahneman and Tversky, 1979; Simon, 1959; Slovic, Peters, Finucane, and MacGregor, 2005; Thaler and Sunstein, 2008; Tversky and Kahneman, 1992). However, in this paper, we consider people to be bounded rational agents (Gigerenzer and Selten, 2002; Simon, 1959), who tend to minimize their losses against landslides slowly over time via a trial-and-error learning process driven by personal experience in an uncertain environment (Dutt and Gonzalez, 2010; Slovic et al., 2005). We have now added these explanations on pg. 3 of the manuscript.

Furthermore, we now also discuss how the repeated experiential feedback likely enables learning by repeated trial-and-error procedures, where bounded-rational individuals (Simon, 1959) try different investment values in ILS and observe their effects on occurrence of landslides and their associated consequences. Also, we now mention that according to Slovic et al. (2005), loss-averse individuals tend to increase their contribution against a risk over time. In our case, similar to Slovic et al. (2005), participants started contributing slowly against landslides and, with the experience of landslide losses over time, they started contributing larger amounts to reduce landslide risks. These explanations have been discussed on pg. 20 of the manuscript.

The authors assumed that "damages concerning injury and fatality affect one's income levels". This is rather naïve. While reduced income level is going to be a consequence, but it would be much less a concern for most people than the injury and fatality itself. In reality people can also choose to migrate when mitigation cost is too high and adaptation becomes impossible. The nature of landslide hazard, including its notorious fame of being extremely hard, if not impossible, to predict, makes it quite different from other hazards such as flood and drought, and general climate risk.

Authors: Thank you for sharing this thought provoking comment. In agreement with you, we have now stated as part of our discussion section that currently, in the ILS model, we have assumed that damages from fatality and injury influence participants' daily-income levels. The reduced income levels do create adverse consequences, but one could also argue that they would be much less of a concern for most people compared to the injury and fatality itself. Furthermore, people could also choose to migrate from an area when the landslide mitigation cost is too high and adaptation becomes impossible, especially due to the differences between the landslide hazard and other hazards such as flood, drought, and general climate risks. As part of our future research, we plan to investigate the influence of feedback that causes only injuries or

fatalities compared to feedback that causes economic losses due to injuries and fatalities. Also, as part of our future research in the ILS tool, we plan to investigate people's migration decisions when the landslide mitigation costs are too high and adaptation to landslides is not possible.

These explanations have been provided as part of the discussion section in the manuscript (see pg. 21).

The authors' choice of P(I) formula from Hasson et al. 2010 does not seem to be appropriate. It may seem to be obviously useful by applying specific parameters from the Mandi area in India as the participants seem to be mostly from the area (the authors did not clearly elaborate this), however, since the algorithms was not disclosed to the participants and a random number generator was used in producing damages, using the seemingly sophisticated algorithms is in fact not much related to the authors' main objective, instead, a more generic algorithm would serve the same purpose and potential be more useful for testing with participants from other areas.

Authors: Thank you for your kind comments. In agreement with your suggestions, we have now stated as part of our discussion section that in the ILS model, we used a linear model to compute the probability of landslides due to human factors (i.e., Hasson et al. 2010's model). Also, the probabilistic equations governing the physical factors in the ILS model were not disclosed to participants, who seemed to possess high education levels. One could argue that there are several other linear and non-linear models that could help compute the probability of landslides due to human factors. Some of these models could not only influence the probability of landslides, but also the severity of consequences (damages) caused by landslides. Also, other generic models could account for the physical factors in the ILS tool. We plan to try these possibilities as part of our future work in the ILS tool. Specifically, we plan to assume different models of investments in the ILS tool and we plan to test them against participants with different education levels.

These explanations have been added to the discussion section (pg. 21).

Also, we have now clearly elaborated in the revised manuscript that the sample used in the experiment was representative of the study area's population because the literacy rate in the town and surrounding areas is quite high (81.5%) (Pg. 15).

Day was used as the time unit for simulation and people make daily choice in landslide mitigation investment. This is not relevant for real world situation either.

Authors: Thank you for your observation.

We have now stated as part of our methods section that the ILS tool can run for different time periods, which could be from days to months to years. This feature can be customized in the ILS tool (pg. 8). However, to showcase the potential of using ILS, the experiment used the daily setting in the ILS tool. As part of our future research, we plan to extend this limitation by considering people to make decisions on a longer time scale ranging from months to years. Please see this discussion in the discussion section of the manuscript (pg. 20-22).

In most cases, especially in developing countries, households and communities themselves almost never have resources substantial enough to mitigate landslide risk, which is often financed by government and/or international donors. The huge disparity between the average asset (calculated as per capital GDP) and the salary (with the former being 2000 times of a person's annual income) also supported my above statement.

Authors: Thank you for your kind comments. In agreement with you, we have now added to our discussion section that we assumed a large disparity between a participant's property wealth and her daily income. In addition, as part of the ILS model, we did not consider any support from government or international agencies against damages from landslides. As suggested by you, in certain cases, especially in developing countries, mitigation of landslide risks may be often financed by government or international agencies. As part of our future work, we plan to extend the ILS model to

include assumptions of contributions from government or international agencies. Such assumptions will help us determine the willingness of common people to contribute against landslide disasters, which is important as the developing world becomes developed over time.

These comments have been reported on pg. 22.

The authors chose a value of 0.8 for W, indicating that the landslide risk can largely be mitigated by human. This is in general not the case, especially for the type of mitigation measures mentioned by the authors – tree plantation. There has been studies showing that afforestation does not help with landslides in similar areas to Mandi in the Sivalik Hills.

Authors: Thank you.

Now, as part of our discussion section (see pg. 22), we have mentioned that these W and M values indicated that landslide risks could largely be mitigated by human actions. However, in agreement with your suggestions, this assumption may not be the case always, especially for mitigation measures like tree plantations. For example, afforestation alone may not help in reducing deep-seated landslides in hilly areas (Forbes, 2011). Thus, it would be worthwhile investigating as part of future research on how people's decision-making evolves in conditions where investments likely influence the landslide probability (higher values of W and M parameters) compared to conditions where investments unlikely influence the landslide probability (lower values of W and M parameters).

3. The study design

The high damage scenario is simply not realistic at all. With such a high risk of mortality and 90% change of injury, no one would still choose to stay in the landslide area, even in least developed countries. The low damage scenario would already be a very high risk area in reality, in any countries.

Authors: Thank you.

In agreement with your suggestions, we have now mentioned as part of our discussion section (see pg. 22) that to test our hypotheses, we presented participants with a high damage scenario and a low damage scenario, where the probability of property damage, injury, and fatality were high and low, respectively. However, such scenarios may not be realistic, where people may want migrate from both low and damage areas in even the least developed countries. In future research with ILS, we plan to calibrate the probability of damages, injury, and fatality to realistic values and test the effectiveness of ILS in improving the participants' investment decision making.

In Fig. 3b, the authors give a smiling face followed by "Landslide did not Occur". This gives a false feeling that the fact that landslide did not occur because of mitigation investment, while in reality much of it should be due to stochastic in the nature of landslide.

Authors: Thank you for your kind comments. In our experiment, when landslide did not occur and experiential feedback was present, people were presented with a smiling face followed by a message. The message and emoticon were provided to connect the cause-and-effect relationships for participants in the ILS tool. However, it could also be that the landslide did not occur on a certain trial due to the stochasticity in the simulation rather than participants' investment actions. Although such situations are possible over shorter time-periods, however, over longer time-periods increased investments from people will only reduce the probability of landslides.

In agreement with your comments, we have now added these explanations as part of the discussion section (pg. 22).

4. The results

First, part of the results were already included in the 2016 paper (apparently including 43 of the 83 participants reported in this study) and this should be fully disclosed.

Authors: Thank you.

In the revised manuscript, we have now given proper citation to our 2016 conference paper (actually 2017 conference paper, where the year has been corrected). We have now clarified the contribution in the manuscript and how this work builds upon the prior 2017 work (pg. 3, 12).

Also, via a footnote on pg. 12, we have mentioned that data reported in Chaturvedi et al. (2017) has been included in this paper with two more conditions, the high-damage feedback-absent (N = 20) and the low-damage feedback-absent (N = 20). Data in all four conditions was collected simultaneously.

Second, the part of the results on people's increasing investment in mitigation seems to be largely an artifact of the choice of M being 0.8. It'd be more interesting to study, with a much larger sample, how how changing M will affect people's behavior, given that the authors choose more realistic scenarios.

Authors: Thank you for your kind comment, which helped us get new ideas for our research. In agreement with you, we have now mentioned that in the experiment, we assumed a value of 0.8 for the return to mitigation (M) parameter. This M value indicated that landslide risks could largely be mitigated by human actions. However, this assumption may not be the case always, especially for mitigation measures like tree plantations. For example, afforestation alone may not help in reducing deep-seated landslides in hilly areas (Forbes, 2011). Thus, it would be worthwhile investigating as part of future research on how people's decision-making evolves in conditions where investments likely influence the landslide probability (higher values of M parameter) compared to conditions where investments unlikely influence the landslide probability (lower values of M parameter).

This discussion appears on pg. 22 of the manuscript.

Some detailed comments on texts:

1. In Abstract, the first sentence is incomplete. ' Authors: Thank you.

In the revised manuscript, we have now improved the first line of abstract. All other formatting errors and references are corrected in the revised version. Now, the manuscript has also been proofed.

2. "Different amount of feedback" was used, but in fact the difference between the two different levels of feedback may better be described as "intensity" of "strength" of feedback.

Authors: Thank you.

In agreement with your kind suggestion, we have now changed the "amount of feedback" in the paper everywhere to the "strength of feedback."

3. Fig. 2 is similar to the Fig 2 in the authors' 2016 conference paper and needs to be disclosed.

Authors: Thank you.

In the revised manuscript, we have now given proper citation to our 2017 conference paper as part of this figure.

4. Fig. 5b, it should be high/low damage instead of more/less damage.

Authors: Thank you.

In the revised manuscript, we have now rectified this error.

5. Reference – Mathew et al. was published in 2014 and should be rearranged in alphabetic order.

Authors: Thank you.

In the revised manuscript, all the formatting errors and references are corrected.

While the study represents an interesting attempt, it suffers from seriously false model

assumptions and weakness in study design in relation to reality. I personally even think that the simulator may falsely influence participants in terms of how they should make decisions in the face of landslide risk. But I strongly recommend the authors to continue developing the simulator with stronger social science understanding and better design.

Authors: We are thankful for your kind comments as they helped us provide an improved exposition of our methods and results. These comments have also given a lot of new ideas which we will use in our future experimentation with ILS tool. We, hereby want to clarify that current experimental study with ILS was a preliminary but important work to test the effectiveness of simulation models on people's understanding of landslide risks. But, in future we will use several of the manipulations in the model parameters and probabilities to make the simulation exercise more realistic.

In agreement with you, we have now added several ideas suggested by you as part of our discussion section in the manuscript (pg. 20-22). The revised draft with the changes made is enclosed as a supplement.

Please also note the supplement to this comment:
https://www.nat-hazards-earth-syst-sci-discuss.net/nhess-2017-297/nhess-2017-297-AC3-supplement.pdf

**Supplement:**

[revised manuscript text omitted]

Dr. Stefano Luigi Gariano
Editor, *Journal of Natural Hazards and Earth System Sciences*

September 02nd , 2017

Dear Dr. Gariano,

I write to you concerning a manuscript, "Learning in an Interactive Simulation Tool against Landslide Risks: The Role of Amount and Availability of Experiential Feedback," that I co-authored with my Ph.D. advisor, Dr. Varun Dutt and Mr. Akshit Arora.

We have now modified our manuscript as per your kind suggestions. Please find attached a revised version of our manuscript with point-to-point replies against your comments and suggestions. Point-to-point responses are provided starting on the next page.

We look forward to hearing from you on the publication of this manuscript in the *Journal of Natural Hazards and Earth System Sciences*.

Sincerely,

Pratik Chaturvedi

Ph.D. Scholar, School of Computing and Electrical Engineering

Indian Institute of Technology Mandi

Kamand-175005, Himachal Pradesh, India

Phone: +91-931-313-1129

Email: prateek@dtrl.drdo.in

**Point-to-point responses**

*Thank you for having improved the quality of the figures. However, please check Figure 2C: the line in the third graph (property wealth) is now different from the previous one).*

**Authors:** Since the quality of figure that we submitted in the first version of the manuscript was poor, we had to re-run ILS to get a better-quality figure. However, the ILS tool is a stochastic simulation tool and thus the re-run gave us a figure that is like the one submitted in the first version of the manuscript; however, the new figure is not identical to the one submitted in the first version of the manuscript. Thus, in the revised version of the manuscript, we have now replaced the figures 2 (A), (B) and (C) with almost similar graphs as were submitted as part of the first submission. We hope that the changes made by us now to Figure 2 are acceptable with you.

*2) There is also another issue that should be addressed before your paper can be published in NHESS open discussion forum. Indeed, I noticed that you did not follow the "Manuscript preparation guidelines for authors" in particular for what concerns references, both in the text and in the list. Thus, I suggest you to please check the reference list following the guidelines. In particular, in the list, a "and" is always needed before the last (or second) author. Moreover, in the text, please replace "&" with "and", and use "et al." when citing papers with more than 2 authors. You can find several examples by looking at the "Copernicus Publications Reference Types" guidelines (see: https://www.natural-hazards-and-earth-system-sciences.net/for_authors/manuscript_preparation.html and https://www.natural-hazards-and-earth-system-sciences.net/Copernicus_Publications_Reference_Types.pdf ). Finally, please check the abbreviations of journal names, according to the ISI Journal Title Abbreviations Index (see: https://www.natural-hazards-and-earth-system-sciences.net/Copernicus_Publications_Reference_Types.pdf and http://library.caltech.edu/reference/abbreviations/).*

**Authors:** We have taken into consideration your advice and we have replaced everywhere "&" with "and" in the manuscript and used "et al." when citing papers with more than two authors. We have taken care of "Copernicus Publications Reference Types" guidelines in this version of our manuscript for "in-text" citations and references list. Thus, now, the formatting of the manuscript (especially the references) are as per the Journal guidelines.

| Page 7: [3] Formatted | \pratik | 25/11/17 10:48 AM |

Centered

| Page 7: [4] Formatted | Varun Dutt | 09/12/17 2:16 PM |

Font:Not Bold

| Page 7: [5] Formatted | Varun Dutt | 09/12/17 2:16 PM |

Font:Not Bold

| Page 7: [6] Formatted | Varun Dutt | 09/12/17 2:16 PM |

Font:Not Bold

| Page 7: [7] Formatted | Varun Dutt | 09/12/17 2:16 PM |

Font:Not Bold

| Page 7: [8] Deleted | \pratik | 25/11/17 10:49 AM |

Once these parameters are determined, equation 4a and 4b help determine the probability of landslide due to rainfall, $P(R)$.

| Page 7: [9] Moved to page 6 (Move #1) | Varun Dutt | 07/12/17 12:45 PM |

In the ILS tool reported ahead, $P(R)$ is shown as the probability of landslides due to rainfall in a certain trial.

| Page 7: [10] Deleted | Varun Dutt | 07/12/17 3:52 PM |

According to Anbalagan (1992), the spatial probability can be provided as seen in

| Page 7: [11] Deleted | Varun Dutt | 07/12/17 3:58 PM |

This table provides a Total Estimated Hazard (THED) based on the Landslide Hazard Map sectioning.

| Page 7: [12] Formatted | \pratik | 27/11/17 7:16 PM |

Font:Cambria Math

| Page 7: [13] Formatted | \pratik | 25/10/17 12:10 AM |

Centered

| Page 7: [14] Formatted | \pratik | 25/10/17 12:11 AM |

Font:10 pt

| Page 7: [15] Formatted | \pratik | 25/10/17 12:11 AM |

Font:10 pt

| Page 7: [16] Formatted | \pratik | 25/10/17 12:11 AM |

Font:10 pt

| Page 7: [17] Formatted | \pratik | 25/10/17 12:11 AM |

Font:10 pt

| Page 7: [18] Formatted | \pratik | 25/10/17 12:11 AM |

Font:10 pt

| Page 7: [19] Formatted | \pratik | 25/10/17 12:11 AM |

Font:10 pt

| Page 7: [20] Formatted | \pratik | 25/10/17 12:11 AM |
|---|---|---|

Font:10 pt

| Page 7: [21] Formatted | \pratik | 25/10/17 12:11 AM |
|---|---|---|

Font:10 pt

| Page 7: [22] Formatted | \pratik | 25/10/17 12:11 AM |
|---|---|---|

Font:10 pt

| Page 7: [23] Formatted | \pratik | 25/10/17 12:11 AM |
|---|---|---|

Font:10 pt

| Page 7: [24] Formatted | \pratik | 25/10/17 12:11 AM |
|---|---|---|

Font:10 pt

| Page 7: [25] Formatted | \pratik | 25/10/17 12:11 AM |
|---|---|---|

Font:10 pt

| Page 7: [26] Formatted Table | \pratik | 25/10/17 12:11 AM |
|---|---|---|

Formatted Table

| Page 7: [27] Formatted | \pratik | 25/10/17 12:11 AM |
|---|---|---|

Font:10 pt

| Page 7: [28] Formatted | \pratik | 25/10/17 12:11 AM |
|---|---|---|

Font:10 pt

| Page 7: [29] Formatted | \pratik | 25/10/17 12:11 AM |
|---|---|---|

Font:10 pt

| Page 7: [30] Formatted | \pratik | 25/10/17 12:11 AM |
|---|---|---|

Font:10 pt

| Page 7: [31] Formatted | \pratik | 25/10/17 12:11 AM |
|---|---|---|

Font:10 pt

| Page 7: [32] Formatted | \pratik | 25/10/17 12:11 AM |
|---|---|---|

Font:10 pt

| Page 7: [33] Formatted | Varun Dutt | 07/12/17 4:06 PM |
|---|---|---|

Font:Italic

| Page 7: [34] Deleted | Varun Dutt | 07/12/17 4:06 PM |
|---|---|---|

From this table, the spatial probability can be calculated by dividing the THED by the corrected Landslide Hazard Evaluation Factor (LHEF) which considers individual and net effect of landslide causal factors also used for Landslide Hazard Zonation (LHZ) mapping.

| Page 7: [35] Formatted | Varun Dutt | 07/12/17 4:08 PM |
|---|---|---|

Font:Italic

| Page 7: [36] Formatted | Varun Dutt | 07/12/17 4:10 PM |
|---|---|---|

Font:Italic

| Page 7: [37] Formatted | Varun Dutt | 07/12/17 4:10 PM |
|---|---|---|

Font:Italic

| Page 7: [38] Deleted | Varun Dutt | 07/12/17 4:13 PM |
|---|---|---|

A landslide occurs on a certain day when a independent random number ($\sim U(0, 1)$) become less than or equal to the corresponding net probability of occurrence of landslide which is a weighted sum of landslide probability due to environment (spatial and triggering factors) and human factors. Once the random number is less than the probability of the corresponding landslide occurrence probability, the landslide occurs.

| Page 8: [39] Deleted | \pratik | 27/11/17 7:17 PM |
|---|---|---|

| Page 14: [40] Deleted | \pratik | 13/12/17 10:55 PM |
|---|---|---|

| Page 14: [41] Deleted | \pratik | 13/12/17 11:35 PM |
|---|---|---|

The ILS tool in the feedback-absent condition. Participants were tasked to enter across 30-days how much out of 292 EC they were willing to contribute against landslides. The task was similar in the high-damage feedback-absent condition; however, the damage percentages in the last paragraph were 30%, 9%, and 90%, respectively.

| Page 17: [42] Deleted | \pratik | 27/11/17 7:20 PM |
|---|---|---|

However, in feedback's absence in ILS, participants were unable to increase their investments for mitigating landslides, even when damages were high compared to low.

**4.3 Participant Strategies**

We analyzed whether an "invest-all" strategy (i.e., investing the entire daily income in mitigating landslides) was reported by participants across different conditions. As mentioned above, the invest-all strategy was an optimal strategy and this strategy's use indicated learning in the ILS tool. Figure 7 shows the proportion of participants reporting the use of the invest-all strategy. Thus, many participants learnt to follow the invest-all strategy in conditions where experiential feedback was present and it was highly damaging compared to participants in the other conditions.

**Discussions and Conclusion**

In this paper, we used an existing Interactive Landslide Simulator (ILS) tool for evaluating the effectiveness of feedback in influencing people's decisions against landslide risks. We used the ILS tool in an experiment involving human participants and tested how the amount and availability of experiential feedback in ILS, including the use of ILS tool itself, helped increase people's investment decisions against landslides. Our results agree with our expectations: Experience gained in ILS enabled improved understanding of processes governing landslides and helped participants improve their investments against landslides. Given our results, we believe

that ILS could potentially be used as a landslide-education tool for increasing public understanding and awareness about landslides. The ILS tool can also be used by policymakers to do what-if analyses in different scenarios concerning landslides.

First, high-damaging feedback in ILS tool helped increase people's investment against landslides over time compared to low-damaging feedback in the tool. Furthermore, the experiential feedback helped participants increase their investments against landslides compared to conditions where this feedback was absent. These result can be explained by previous lab-based research on use of repeated feedback or experience (Chaturvedi et al., 2016; Dutt & Gonzalez, 2011; Fischoff, 2001; Finucane et al., 2000). Repeated experiential feedback likely enables learning by repeated trial-and-error procedures, where participants try different investment values in ILS and observe their effects on occurrence of landslides. This feedback is higher in the condition when damages are more compared to when damages are less and this difference in feedback influences participant investments against landslides. In fact, we observed that the use of the optimal invest-all strategy was maximized when the experiential feedback was highly damaging.

We also believe that the ILS tool can be integrated in teaching courses on landslide sustainable practices in K-12 schools. This course could make use of the ILS tool and focus on educating students about causes, consequences, and risks of hazardous landslides. We believe that the use of ILS tool will make teaching more effective as ILS will help incorporate experiential feedback and social norms in teaching in interactive ways.

| Page 20: [43] Deleted | Varun Dutt | 09/12/17 9:04 PM |
|---|---|---|

in ILS tool

| Page 20: [43] Deleted | Varun Dutt | 09/12/17 9:04 PM |

in ILS tool

| Page 20: [44] Deleted | Varun Dutt | 09/12/17 9:16 PM |

people who are loss averse,

people who are loss averse,

| **Page 20: [45] Deleted** | **Varun Dutt** | **03/12/17 3:10 PM** |

| **Page 20: [45] Deleted** | **Varun Dutt** | **03/12/17 3:10 PM** |

| **Page 20: [45] Deleted** | **Varun Dutt** | **03/12/17 3:10 PM** |

| **Page 20: [45] Deleted** | **Varun Dutt** | **03/12/17 3:10 PM** |

| **Page 20: [46] Deleted** | **Varun Dutt** | **03/12/17 3:13 PM** |

.

| **Page 20: [46] Deleted** | **Varun Dutt** | **03/12/17 3:13 PM** |

.

| **Page 20: [46] Deleted** | **Varun Dutt** | **03/12/17 3:13 PM** |

.

| **Page 20: [47] Deleted** | **Varun Dutt** | **09/12/17 9:09 PM** |

is

| **Page 20: [47] Deleted** | **Varun Dutt** | **09/12/17 9:09 PM** |

is

| **Page 20: [47] Deleted** | **Varun Dutt** | **09/12/17 9:09 PM** |

is

| **Page 20: [47] Deleted** | **Varun Dutt** | **09/12/17 9:09 PM** |

is

| **Page 20: [47] Deleted** | **Varun Dutt** | **09/12/17 9:09 PM** |

is

| Page 20: [47] Deleted | Varun Dutt | 09/12/17 9:09 PM |

is

| Page 20: [47] Deleted | Varun Dutt | 09/12/17 9:09 PM |

is

| Page 20: [47] Deleted | Varun Dutt | 09/12/17 9:09 PM |

is

| Page 20: [47] Deleted | Varun Dutt | 09/12/17 9:09 PM |

is

| Page 21: [48] Deleted | \pratik | 27/11/17 7:13 PM |

The ILS tool's parameter settings could be customized to a certain geographical area over a certain time period of play. In addition, the ILS tool could be used to present investment actions of other decision-makers (e.g., society or neighbours) compared to one's own investment actions. The presence of investment of other decision-makers in addition to one's own decisions will likely enable the use of social norms towards learning (Schultz et al., 2007). These features makes ILS tool very attractive for landslide education in communities in the future. Furthermore, the ILS tool holds a great promise for policy-research against landslides. For example, in future, researchers may vary different system-response parameters in ILS (e.g. weight of one's decisions and return to mitigation actions) and feedback (e.g. numbers, text messages and images for damage) in order to study their effects on people's decisions against landslides. Here, researchers could evaluate differences in ILS's ability to increase public contributions in the face of other system-response parameters and feedback. In addition, researchers can use the ILS tool to do "what-if" analyses related to landslides for certain time periods and for certain geographical locations. The ILS tool has the ability to be customized to certain geographical area as well as certain time periods, where spatial parameters (e.g., soil type and geology) as well as temporal parameters (e.g., daily rainfall) can be defined for the area of interest. Once the environmental factors have been accounted for, the ILS tool enables researchers to account for assumptions on human factors (contribution against landslides) with real-world consequences (injury, fatality, and infrastructure damage). Such assumptions may help researchers model human decisions in computational cognitive models, which are based upon influential theories of how people make decisions from feedback (Dutt & Gonzalez, 2012; Gonzalez & Dutt, 2011). In summary, these features make ILS tool apt for policy research, especially for areas that are prone to landslides. This research will also help test the ILS tool and its applicability in different real-world settings.

| Page 21: [49] Deleted | Varun Dutt | 09/12/17 9:15 PM |

Although we could investigate that the ILS tool causes the use of optimal invest-all strategies among people in conditions where experiential feedback is highly damaging; however, future research should focus on investigating more deeply about the nature of learning that the tool imparts among people. As people's investments for mitigating landslides in ILS directly influences the risk of landslides due to human and environmental factors, investments indeed have the potential of educating people about landslide risks. Still, it is important to investigate how investing money in the ILS tool truly educates people about landslides. Our current research was a preliminary work and the assumptions made by us in ILS model may not be realistic, but in future, we will manipulate the probabilities related to landslide and damages caused to see effects of different

settings of ILS on participants' risk perception, attitude and behaviour. However, up to certain extent, we were able to capture the people's behavior.

| Page 21: [50] Moved to page 20 (Move #2) | Varun Dutt | 09/12/17 9:15 PM |

According to Slovic et al. (2005), people who are loss averse, tend to increase their contribution in case of a risk over time. In our case also, the participants' in all the experimental conditions, did not started contributing good amount upfront, but with time as they experienced some losses due to their poor investments, they have started contributing large amount of money to reduce the risk.

| Page 21: [51] Deleted | Varun Dutt | 09/12/17 4:07 PM |

we will try to find without causing reduction in income, only due to fatality and injury what effect it have on participants' investment

| Page 21: [52] Deleted | Varun Dutt | 09/12/17 4:09 PM |

Another idea is to test whether people would continue to invest large money or choose to migrate.

| Page 21: [53] Deleted | Varun Dutt | 09/12/17 4:10 PM |

This idea is very interesting to study because The nature of landslide hazard, including its notorious fame of being extremely hard, if not impossible, to predict, makes it quite different from other hazards such as flood and drought, and general climate risk.

| Page 21: [54] Deleted | Varun Dutt | 09/12/17 4:49 PM |

to calculate P(I) to showcase the potential of using ILS in the real-world

| Page 24: [55] Deleted | \pratik | 03/09/17 10:36 PM |

*Acknowledgements*

This research was partially supported by a grant from Himachal Pradesh State Council for Science, Technology & Environment to Varun Dutt (grant number: IITM / HPSCSTE / VD / 130). We thank Akanksha Jain and Sushmita Negi, Centre for Converging Technologies, University of Rajasthan, India for providing preliminary support for data collection in this project.

| Page 24: [56] Deleted | \pratik | 29/08/17 10:52 PM |

Becker, J. S., Paton, D., Johnston, D. M., & Ronan, K. R. (2013). Salient beliefs about earthquake hazards and household preparedness. Risk analysis, 33(9), 1710-1727.

| Page 24: [57] Deleted | Varun Dutt | 01/09/17 5:20 PM |

Available at: (http://iret.co.in/Docs/IJETEE/Volume%2010/Issue10/25.%20Remote%20Sensing%20Based%20Regional%20 Landslide%20Risk%20Assessment.pdf) (Accessed 29 August 2017)

| Page 24: [58] Deleted | \pratik | 29/08/17 11:07 PM |

Chaturvedi P., Dutt V., Jaiswal B., Tyagi N., Sharma S., Mishra S. P., Dhar S., & Joglekar P. N. (2014). Remote Sensing Based Regional Landslide Risk Assessment. International Journal of Emerging Trends in Electrical and Electronics (IJETEE –ISSN: 2320-9569) Vol. 10, Issue. 10, 135-140.

| Page 24: [59] Deleted | \pratik | 29/08/17 11:21 PM |

Chaturvedi, P., & Dutt, V. (2015). Evaluating the Public Perceptions of Landslide Risks in the Himalayan Mandi Town. In Proceedings of the Human Factors and Ergonomics Society Annual Meeting (Vol. 59, No. 1, pp. 1491-1495). SAGE Publications.

| Page 24: [60] Deleted | \pratik | 29/08/17 11:25 PM |
|---|---|---|

Clerici, A., Perego, S., Tellini, C., & Vescovi, P. (2002). A procedure for landslide susceptibility zonation by the conditional analysis method. Geomorphology, 48(4), 349-364.

| Page 24: [61] Deleted | \pratik | 29/08/17 11:26 PM |
|---|---|---|

Dai, F. C., Lee, C. F., & Ngai, Y. Y. (2002). Landslide risk assessment and management: an overview. Engineering geology, 64(1), 65-87.

| Page 24: [62] Deleted | \pratik | 29/08/17 11:27 PM |
|---|---|---|

Dutt, V., & Gonzalez, C. (2012a). Why do we want to delay actions on climate change?   Effects of probability and timing of climate consequences. Journal of Behavioral Decision Making, 25(2), 154-164.

Dutt, V., & Gonzalez, C. (2012b). Human control of climate change. Climatic change, 111(3-4), 497-518.

| Page 24: [63] Deleted | \pratik | 29/08/17 11:31 PM |
|---|---|---|

Dutt, V., & Gonzalez, C. (2012c). Decisions from experience reduce misconceptions about climate change. Journal of Environment Psychology, 32(1), 19-29. doi: 10.1016/j.jenvp.2011.10.003

| Page 24: [64] Deleted | \pratik | 29/08/17 11:33 PM |
|---|---|---|

Dutt, V., & Gonzalez, C. (2011). A generic dynamic control task for behavioral research and education. Retrieved from CMU website: http://repository.cmu.edu/sds/118/

Fischer, C. (2008). Feedback on household electricity consumption: a tool for saving energy? Energy Efficiency, 1(1), 79-104. doi: 10.1016/j.jenvp.2011.10.003

| Page 25: [65] Formatted | \pratik | 13/12/17 10:33 PM |
|---|---|---|

Indent: First line:  0 cm

| Page 25: [66] Formatted | \pratik | 13/12/17 10:34 PM |
|---|---|---|

Font:Not Italic

| Page 25: [67] Formatted | \pratik | 11/12/17 11:12 PM |
|---|---|---|

Font:Not Italic

| Page 25: [68] Deleted | \pratik | 29/08/17 11:35 PM |
|---|---|---|

Finucane, M. L., Alhakami, A., Slovic, P., & Johnson, S. M. (2000). The affect heuristic in judgments of risks and benefits. Journal of behavioral decision making, 13(1), 1-17.

Geosciences group. (2015). Experimental Landslide Early Warning System for Rainfall Triggered Landslides. Retrieved from http://bhuvan-noeda.nrsc.gov.in/disaster/disaster/tools/landslide/doc/landslide_warning.pdf

| Page 25: [69] Formatted | \pratik | 03/09/17 8:41 PM |
|---|---|---|

Indent: First line:  1.27 cm

| Page 25: [70] Deleted | \pratik | 29/08/17 11:43 PM |
|---|---|---|

Glade, T., Anderson, M. G., & Crozier, M. J. (Eds.). (2006). Landslide hazard and risk. John Wiley & Sons.

| Page 25: [71] Formatted | \pratik | 03/09/17 7:40 PM |
|---|---|---|

Indent: First line:  1.27 cm

| Page 25: [72] Deleted | \pratik | 29/08/17 11:37 PM |

Gonzalez, C., & Dutt, V. (2011). Instance-based learning: Integrating sampling and repeated decisions from experience. Psychological review, 118(4), 523.

| Page 25: [73] Formatted | \pratik | 03/09/17 7:40 PM |

Indent: First line: 1.27 cm

| Page 25: [74] Deleted | \pratik | 30/08/17 11:27 PM |

Grasso, V. F., & Singh, A. (2011). Early warning systems: State-of-art analysis and future directions. Draft report, UNEP.

| Page 25: [75] Formatted | \pratik | 03/09/17 7:40 PM |

Indent: First line: 1.27 cm

| Page 25: [76] Formatted | \pratik | 01/12/17 10:40 PM |

Indent: First line: 0 cm

| Page 25: [77] Formatted | \pratik | 01/12/17 10:42 PM |

Font:Not Italic

| Page 25: [78] Formatted | \pratik | 11/12/17 11:12 PM |

Font:Not Italic

| Page 25: [79] Deleted | \pratik | 29/08/17 11:38 PM |

Hasson, R., Löfgren, Å., & Visser, M. (2010). Climate change in a public goods game: investment decision in mitigation versus adaptation. Ecological Economics, 70(2), 331-338.

ISDR 2004 Terminology: basic terms of disaster risk reduction. http://www.unisdr.org/eng/library/lib-terminology-eng%20home.htm, International Strategy for Disaster Reduction secretariat, Geneva.

| Page 25: [80] Formatted | \pratik | 01/09/17 11:28 PM |

Subscript

| Page 25: [81] Formatted | \pratik | 03/09/17 7:40 PM |

Indent: First line: 1.27 cm

| Page 25: [82] Deleted | \pratik | 29/08/17 11:40 PM |

Knutti, R., Joos, F., Müller, S. A., Plattner, G. K., & Stocker, T. F. (2005). Probabilistic climate change projections for CO2 stabilization profiles. Geophysical Research Letters, 32(20).

| Page 25: [83] Formatted | \pratik | 03/09/17 7:40 PM |

Indent: First line: 1.27 cm

| Page 25: [84] Formatted | \pratik | 24/10/17 2:29 PM |

Indent: First line: 1.27 cm

| Page 25: [85] Formatted | \pratik | 03/09/17 7:40 PM |

Indent: First line: 1.27 cm

| Page 25: [86] Formatted | \pratik | 03/09/17 7:40 PM |

Indent: Left: 1.27 cm

**Page 25: [87] Deleted** \pratik 31/08/17 12:01 AM

Meissen, U., &Voisard, A. (2008, May). Increasing the effectiveness of early warning via context-aware alerting. In Proceedings of the 5th International Conference, on Information Systems for Crisis Response and Management (ISCRAM) (pp. 431-440).

**Page 25: [88] Formatted** \pratik 30/08/17 11:16 PM

Font:Not Bold

**Page 25: [89] Formatted** \pratik 30/08/17 11:16 PM

Font:Not Bold

**Page 25: [90] Formatted** \pratik 30/08/17 11:16 PM

Font:Not Bold

**Page 25: [91] Formatted** \pratik 03/09/17 7:40 PM

Indent: First line: 1.27 cm

**Page 25: [92] Formatted** \pratik 30/08/17 11:16 PM

Font:Not Bold

**Page 25: [93] Formatted** \pratik 30/08/17 11:16 PM

Font:Not Bold

**Page 25: [94] Deleted** \pratik 30/08/17 11:16 PM

Montrasio, L., Valentino, R., &Losi, G. L. (2011). Towards a real-time susceptibility  assessment of rainfall-induced shallow landslides on a regional scale. Natural Hazards and Earth System Science, 11(7), 1927-1947.

**Page 25: [95] Formatted** \pratik 03/09/17 7:40 PM

Indent: First line: 1.27 cm

**Page 25: [96] Formatted** \pratik 03/09/17 7:40 PM

Indent: Left: 1.27 cm

**Page 25: [97] Deleted** \pratik 30/08/17 12:07 AM

Osuret, J., Atuyambe, L. M., Mayega, R. W., Ssentongo, J., Tumuhamye, N., Bua, G. M., Tuhebwe, D., & Bazeyo, W. (2016). Coping strategies for landslide and flood disasters: a qualitative study of Mt. Elgon Region, Uganda. PLoS currents, 8.

**Page 25: [98] Formatted** \pratik 03/09/17 7:41 PM

Indent: First line: 1.27 cm

**Page 25: [99] Formatted** \pratik 01/12/17 10:34 PM

Font:Not Italic

**Page 25: [100] Formatted** \pratik 13/12/17 10:22 PM

Font:Not Italic

**Page 26: [101] Deleted** \pratik 29/08/17 11:50 PM

Quigley, K. S., Lindquist, K. A., & Barrett, L. F. (2013). Inducing and measuring emotion: Tips, tricks, and secrets. In H. T. Reis and C. M. Judd (Eds.) Handbook of Research Methods in Social and Personality Psychology (p. 220-250). New York: Cambridge University Press.

| Page 26: [102] Deleted | Varun Dutt | 13/12/17 1:19 PM |
| --- | --- | --- |

Slovic, P., Peters, E., Finucane, M.L. and MacGregor, D.G.: Affect, risk, and decision making. *Health psychology*, *24*(4S), p.S35, 2005.

| Page 26: [103] Deleted | \pratik | 30/08/17 10:51 PM |
| --- | --- | --- |

Rogers, D., &Tsirkunov, V. (2011). Implementing hazard early warning systems. Report, Global Facility for Disaster Reduction and Recovery.

Schultz, P. W., Nolan, J. M., Cialdini, R. B., Goldstein, N. J., & Griskevicius, V. (2007). The constructive, destructive, and reconstructive power of social norms. Psychological science, 18(5), 429-434.

| Page 26: [104] Deleted | \pratik | 30/08/17 10:44 PM |
| --- | --- | --- |

Sterman, J. D. (2000). Business dynamics: Systems thinking and modeling for a complex world. Cambridge, MA: McGraw Hill.

[revised manuscript text omitted]

**Best of Luck!**

---

## Author Response (AR1)

December 25th, 2017

Dr. Stefano Luigi Gariano

Handling Editor, Natural Hazards and Earth System Sciences (NHESS)

Email: gariano@irpi.cnr.it

Dear Dr. Gariano:

I write to you concerning a manuscript, "Learning in an Interactive Simulation Tool against Landslide Risks: The Role of Strength and Availability of Experiential Feedback," that I co-authored with my Ph.D. advisor, Dr. Varun Dutt. Please note that the manuscript's title has changed from the last submission to account for one of the reviewer's comments.

We want to thank you and the reviewers for considering our work to NHESS. As per your suggestions and those of reviewers', we have now modified and improved the exposition of our research in the manuscript. We are now re-submitting an improved version of our manuscript to NHESS. We are also submitting point-to-point answers against different comments and suggestions given by you and reviewers. We hope that you now find our revised paper fit for publication in NHESS and we look forward to hearing from you on this draft.

Sincerely,

Pratik Chaturvedi

Ph.D. Scholar, School of Computing and Electrical Engineering

Indian Institute of Technology Mandi

Kamand-175005, Himachal Pradesh, India

Phone: +91-931-313-1129

Email: prateek@dtrl.drdo.in

**General comments:**

*The paper deals with a very relevant topic, the involvement of stakeholders in landslide risk management and the adoption of "gamification" type approaches to promote it. The ILS software results a promising tool for capturing the interest of attendees and it could be applied with reduced effort to other test cases. The sections 3 and 4 show in effective ways procedures and results.*

**Authors:** Thank you for appreciating our research. We agree with you that the ILS tool is a promising tool for capturing the decisions of participants against landslide risks and it could be applied with reduced effort to other natural disasters involving human decisions.

We have now added these suggestions as part of our discussion section in the manuscript (pg. 20-22).

*However, several elements would require further examination. First, the test case is not adequately introduced: geology, past and recent events, rainfall patterns recognized as main triggers. In this regard, also in ILS, dynamics inducing the events (physical or anthropic) are not adequately taken into account. For example, it is not clear how the spatial distribution of landslide events is accounted for in ILS or if the information about occurrence probability are used in simulation.*

**Authors:** Thank you for your kind comments.

We have now extended our methodological exposition by showing how spatial probabilities (susceptibility of an area to landslides) along with environmental probabilities (triggers due to rainfall patterns) influence the total landslide risk excluding the human factor (pgs. 5-7). Specifically, we have now shown how we used the spatial area and the total estimated hazard (THED) scale of study area in ILS to compute the spatial probability distribution (P(S); pg. 7). In addition, we have now explained how a value of spatial probability was sampled from the P(S) distribution for each participant in ILS (pg. 10). Next, we have also shown how the environmental probability distribution was calculated in ILS from the seasonal rainfall in the area (pg. 5-6). Finally, we have now also shown how the human decisions cause a change in the anthropic probability of landslides and how the anthropic probability interacts with the spatial and environmental probabilities (pg. 5-6).

*The role of "anthropic activities" on slopes could often be detrimental and the reduction in earnings due to reducing these one for preserving stability should be taken into account. Moreover, the main stakeholders for ILS are probably not citizens but policy makers and administrators and then financial management (daily income) should be revised accordingly.*

**Authors:** Thank you for your kind comments.

In agreement with you, we have now explained how the anthropic activities may be detrimental to landslide risks (footnote 1 on pg. 4). Also, we have discussed both these ideas as part of our manuscript's discussion section. Specifically, we now discuss both the positive and negative (detrimental) effects of human actions in influencing the anthropic probability of landslides (e.g., afforestation may not help deep-seated landslides). Also, we have now discussed that the

use of the ILS tool goes beyond school education and it applies to administrative and policy research as well (pg. 20-22). Here, we have mentioned that for pursuing this research in future, the financial components would have to be revised in ILS to include the population at the risk (rather than a single individual's savings) (pg. 20-22).

*The timescales also for simulations does not appear adequate. Several decisions and protection measures need substantial longer times. Timing for measure implementation could be crucial for deciding the more effective strategies.*

**Authors:** Thank you for your comment. We have now stated as part of our methods section that the ILS tool can run for different time periods, which could be from days to months to years.

This feature can be customized in the ILS tool (pg. 8). However, to showcase the potential of using ILS in the real-world, the experiment used the daily setting in the ILS tool. By using the daily setting, we were also able to use the logistic-regression equation to derive the daily probability of landslides due to rainfall (pg. 7). However, as part of our future research, we plan to extend this daily assumption by considering people to make decisions on longer time-scales ranging from months to years. We have added this discussion on pgs. 20-22.

*Finally, the references in first part should be extended and updated. Under such constraints, a substantial revision (major revision) of the text should be performed in order to address the issues arisen above (and below) on specific items; on the other side, the text could be rearranged only to promote the general approach and followed procedures and main results stressing the role that it could cover for landslide risk management after proper characterizations of areas of interest.*

**Authors:** Thank you for your comment.

We have now cited latest research concerning landslide risk in the paper, including more research about Early Warning Systems (EWSs) for landslide risk reduction (pg. 1-3). In addition, we have now broadened the discussion section of the paper by including the points suggested by you and other referees (pg. 20-22). Furthermore, we have now also clarified the exposition of different probabilities concerning the anthropic, spatial, and environmental factors in influencing landslide susceptibility in the manuscript (pg. 5-7). In agreement with your kind suggestion, this exposition allowed us to promote the general capabilities of the ILS tool and the procedures we followed for generating outcomes and probabilities.

**Specific issues:**

**Abstract:**

*rephrase the first sentence; the verb appears missing*

**Authors:** Thank you for the comment.

We have now improved the first sentence of the abstract (Pg. 1).

**Introduction**

*L25-27: please give further details; in my view, "Knowledge about causes-and consequences of landslides and awareness about landslide disaster mitigation" act in different ways; the first one supporting structural protection measurements could reduce the occurrence/magnitude of landslides. The other one tends reducing people and assets vulnerability not varying the physical processes inducing them.*

**Authors:** Thank you for your kind comments.

In agreement with you, we have now clarified on lines 25-27 that imparting knowledge about causes-and-consequences as well as spreading awareness about landslide disaster mitigation are two different ways of managing landslide risks. The former supports structural protection measures that reduce the probability of landslides. In contrast, the latter likely reduces people's and assets' perceived vulnerability and it does not influence the physical processes. We believe that the ILS tool engages people in both ways (pg. 1).

*L31-33: please add further details about Early Warning System tools; e.g. you could refer to reviews available in literature.*

**Authors:** Thank you for your kind suggestions. We have now cited more research about Risk Communication Systems.

Specifically, we have now added on pg. 2 of the revised manuscript that Several satellite-based and sensor-based landslide monitoring systems are being used in landslide RCSs (Hong et al., 2006; Quanshah et al., 2010; Rogers et al., 2011). To be effective, however, landslide RCSs need not only be based upon sound scientific models, but, they also need to consider human factors, i.e., the knowledge and understanding of people residing in landslide-prone areas (Meissen and Voisard, 2008).

*L71: "Chaturvedi et al. (2016)" reference is missing in the list*

**Authors:** The reference's year should have been 2017 and not 2016. We have fixed this typo everywhere in the revised manuscript.

We have also rectified the referencing problems in the reference section in the revised manuscript (Pg. 24-26).

*L82-83: please consider, I'm not sure that "increasing the amount of damage feedback" and "increasing the probabilities of landslide damages" could be assumed equivalent*

**Authors:** Thank you for the comment. In agreement with you, we have now revised the wording as the following:

"…increasing the strength of damage feedback by increasing the probabilities of landslide damages in simulation tools." (pg. 3).

**2 Computational model of landslide risk**

*L106-108 (Figure 1): for landslides, the issue could be quite more challenging; indeed, you should consider "human interventions" detrimental for slope stability. For example, land*

*use/cover changes (e.g. deforestation, conversion to agricultural practices). In this regard, rainfall required to induce the phenomena (e.g. duration, intensity) could be affected by "human interventions". Furthermore, researchers monitor data for landslide occurrence but not determine them as for "user" with investments. Finally, both influence not only the hazard ("total probability of landslide") but the risk.*

**Authors:** Thank you for your kind comments. We agree with your observations.

Now, as part of our revised manuscript, we have mentioned that although our model assumes human mitigation actions in the ILS tool, there may also be other model assumptions possible where certain human detrimental actions (footnote 1 on pg. 4). For example, deforestation may increase the probability of landslides or the risk (probability * consequence) of landslides. We plan to consider these model assumptions as improvements to our model as part of our future research (pg. 20-22).

Furthermore, in this manuscript, we restricted our analyses to only people's investments influencing landslides. However, we agree with you that there may be contributions made by the national, regional, and local governments for providing protection measures against landslides in addition to the investments made by people residing in the area We plan to consider the role of governments as part of our future research (pg. 4). We have also discussed these issues in the discussion section of our revised manuscript and we will take them up as a part of the future work to make the ILS model more realistic (pg. 20-22).

*L109: please specify if you consider weather(rainfall)-induced landslides*

**Authors:** Thank you. In the current work, we are only dealing with weather (rainfall)-induced landslides.

We have now mentioned this point as footnote 2 on pg. 4.

*L128: the main part of investments for protection measurements as structural (e.g. drainages, retention walls) as soft (e.g. EWS) are funded by Administrations (National, Regional and Local); in which ways it is accounted for?*

**Authors:** Thank you. The theme of our research in the manuscript was focused upon common people's contribution for mitigating landslide risks and the effectiveness of the ILS tool in improving people's understanding about landslide processes.

We agree with your comments and as part of our revision we have now added this point on page 4 as a foot note as well as in the discussion section (pg. 20-22).

**Section 2.1.2:**

*further clarifications are needed. Firstly, brief information about the landslides in the area of interest are required; indeed, the relevance of antecedent precipitations is strictly linked to several geomorphological factors (e.g. soil depth, bottom boundary conditions, hydraulic and mechanical properties); without them, it is not possible to evaluate if considered durations (1d, 3d, 30d) are proper. Moreover, it is not clear the role of "Landslide Susceptibility Zonation";*

*indeed, "susceptibility" does not provide details about frequency of phenomena but attempts defining the area more "vulnerable" to the events while in this case it is intended providing also Hazard. Moreover, please add details about the rating (0-11). Finally, all the slopes in the area are recognized to be affected by the same rainfall patterns (similar properties, similar soil depths and so on)?*

**Authors:** Thank you for your kind comments.

We have now extended our methodological exposition by showing how spatial probabilities (susceptibility of an area to landslides) along with environmental probabilities (triggers due to rainfall patterns) influence the total landslide risk (test case) excluding the human factor (pg. 5-7). Specifically, we have now shown how we used the spatial area and the total estimated hazard (THED) scale of the study area in ILS to compute the spatial probability (P(S)) distribution (pg. 7). In addition, we have explained how a value of spatial probability was sampled from the P(S) distribution for each participant.

Next, we have now also shown the environmental probability distribution and how it was calculated in ILS from the seasonal rainfall in the area (pg. 5-7). Finally, we have now shown how the human decisions causes a change in the anthropic probability of landslides and these decisions interact with the spatial and environmental probabilities (pg. 5-7).

*L170: what do you intend for landslide "benign"?*
**Authors:** When the landslide is benign, then there is no injury, fatality, or damage to property.

We have now added this definition to the manuscript (pg. 8, 10).

**Section 2.1.3:**

*please, what do you intend for "random numbers"? which ways are the three damage probabilities computed in?*

**Authors:** Thank you for your kind comments. If a uniformly distributed random number in [0, 1] (U (0, 1)) is less than a probability value, then it simulates this probability value. For example, if U (0, 1) < 30%, then U(0, 1) will be less than the 30% value exactly 30% of the total number of times it is simulated. Thus, this process will simulate a 30% probability value. A landslide occurs on a certain day when a independent random number (~ U(0, 1)) become less than or equal to the corresponding net probability of occurrence of landslide. Similarly, we have used three independent random numbers (uniformly distributed, values ranging from 0 to 1) for each of the three damage probabilities. Whenever, the random number corresponding to the probability value, become less than the probability, then that kind of damage will occur.

We have now included these details on pgs. 5-7 of the revised manuscript.

**Section 2.2:**

*why do you consider a daily time step? Several decisions and protection measures need substantial longer times. Timing for measure implementation could be crucial for deciding the more effective strategies.*

**Authors:** Thank you for your kind comments.

We have now stated as part of our methods section that the ILS tool can run for different time periods, which could be from days to months to years (pg. 7). Furthermore, the length of the time-period in the ILS can also be customized (pg. 7). For this manuscript, we have used the daily setting in the ILS tool to showcase the potential of using this tool for improving understanding of landslide risks among people. As part of our future research, however, we plan to extend our findings by considering people to make decisions on a longer time scales ranging from months to years. Please see this discussion in the discussion section of the revised manuscript (pg. 20-22).

*L205: who is the reference stakeholder of interest? Citizens, administrators, policy makers.*

**Authors:** Thank you for your kind comment.

We have now clarified that "decision maker" refers to participants, i.e., common people residing in the study area (pg. 10).

*L212-213: in ILS, how is it decided if, for a certain day, landslide could occur or not?*

**Authors:** A landslide occurs on a certain day when an independent random number ($\sim U(0, 1)$) become less than or equal to the corresponding net probability of occurrence of landslide, which is a weighted sum of landslide probability due to environment (spatial and triggering factors) and human factors.

We have now mentioned this point on line 145 (pg. 5).

**3 Experiment**

*L289-295: I am not sure that the sample composition is consistent with those of communities living in the area affected by landslides as in terms of background as in terms of age. It could deeply affect the findings and the generalization of the results also taking into account the very interesting issues arisen in L44-47*

**Authors:** Thank you for your kind comments. The sample was representative of the study area's population because, like in our sample, the literacy rate is quite high (81.5%) in the study area. In addition, before the experiment, participants were also asked about their self-rated knowledge level for landslide risks.

We have now mentioned these points in the revised manuscript on pg. 15. Furthermore, we have also observed that the use of the optimal invest-all strategy was maximized when the experiential feedback was highly damaging in the ILS tool. One likely reason for this observation could be the high educational levels of participants residing in the study area, where the literacy rate was more than 80%. Thus, it seems that participants' education levels helped them make the best use of damaging feedback. We have discussed these points on pg. 20 in the revised manuscript.

*L302: It is quite equal to what reported in L287; in my view, it could be removed*

**Authors:** Thank you for your comment.

As per your kind suggestions, we have now removed this repeated line from the paper.

*L313: please, provide further details about the symbols reported in brackets*

**Authors:** We performed analysis of variance statistical tests for evaluating our expectations. The F-statistics is the ratio of between-group variance and the within-group variance. The numbers in brackets after the F-statistics are the degrees of freedom (K-1, N - K), where K are the total number of groups compared and N is the overall sample size. The p-value indicates the evidence in favour of the null-hypothesis when it is true. We reject the null-hypothesis when p-value is less than the alpha-level (0.05). The $\eta^2$ is the proportion of variance associated with one or more main effects. It is a number between 0 and 1 and a value of 0.02, 0.13, and 0.26 measures a small, medium, or large correlation between the dependent and independent variables given a population size.

We have now mentioned these points as a footnote on pg. 15.

*L374: what do you intend for "K-12"?*

**Authors:** By K-12 we meant kindergarten to standard $12^{th}$.

We have now clarified this definition on pg. 20.

*L457: Mathew et al. reference should be moved in proper alphabetical order*

**Authors:** Thank you for the comment.

As per your kind suggestions, we have now moved the reference Mathew et al. to the proper place as per alphabetic order in the manuscript.

**Appendix A**

*It reports information quite similar to those in Figure 4; for these reason, it could be removed*

**Authors:** We agree with your kind assessment.

As per your kind suggestions, we are now removing Appendix A from the paper.

*The manuscript presents an interesting tool for testing the people's propensity to invest money for protecting goods and life from landslides. The tool has been applied for analyzing the effect of feedbacks availably in influencing the people's decision-making process when asked to invest resources for landslide protection. The topic of the manuscript fit into the scopes of the NHESS Journal since it deals with the design and implementation of mitigation and adaptation strategies to reduce the impact of hazardous natural events on human-made structures, infrastructure, and life.*

**Authors:** Thank you for appreciating our research. We agree with you that the ILS tool is a promising tool for capturing the decisions of participants against landslide risks and it could be applied with reduced effort to other natural disasters involving human decisions.

We have added these points as part of our discussion section in the manuscript (pgs. 20-22).

**General comments:**
*The structure of the paper is fair and, even if I'm not a native English speaker, I found the paper understandable. However, I think that some improvements can be made simplifying the sentences and re-phrasing some frequent constructs as "Although . . . . . ; however . . . . .", where the semicolon do not help to understand the sentence.*

**Authors:** Thank you for appreciating our research.

We have now modified the language of the manuscript according to your suggestions and removed the use of semicolons.

*I suggest to promote the section "Interactive landslide Simulator (ILS) tool" from the level of a subsection to the level of a section. Currently it is, erroneously, inside the "Computational model of landslide risk" section. More in general I would also suggest to the author to use the common scientific structure which includes "Introduction", "Material and Methods", "Results", "Discussion" (currently discussion and results are in the same section).*

**Authors:** Thank you for the comment.

We have now made ILS tool as a separate section (pg. 8). Also, we have modified the headings in the manuscript as per your kind suggestions.

*I think that the ILS tool is very interesting but I see a major problem in the paper: there is not the possibility to test the ILS tool. My opinion is that, according to the open science, open data, open knowledge concepts, researchers should be put in condition of evaluating the ILS tool. From the paper it is not clear if the tool is a web application or a standalone program and there is not a description of the technology adopted for implementing it, nor of the intention of the authors of releasing the code and, if this is the case, adopting which license.*

**Authors:** Thank you for your kind comments. ILS is a web-based tool that one can access on the following URL: www.pratik.acslab.org.

We have now provided a link to the tool on pg. 8 and can provide the tool's code upon request.

*Another issue is about the significance of the results of the experiment. Evidences are that people using the ILS tool with feedbacks, rapidly understand that the best strategy to "win the game" is to invest the entire daily income in landslides mitigation measures. Even if this is interesting, the authors do not comment or discuss the fact that the population of the participants is made of people having high to very high educational levels. This can have a strong effect on their capacity to rapidly find the best strategy. This is particularly true where one considers that, as far as I know, the educational levels of people living in the Himalaya region is mostly low and very low. I think that representativeness of the participants to the experiment should be discussed more in detail.*

**Authors:** Thank you for your kind comments. The sample was representative of the study area's population because, like in our sample, the literacy rate is quite high (81.5%) in the study area. In addition, before the experiment, participants were also asked about their self-rated knowledge level for landslide risks.

We have now mentioned these points in the revised manuscript on pg. 15. Furthermore, we have also observed that the use of the optimal invest-all strategy was maximized when the experiential feedback was highly damaging in the ILS tool. One likely reason for this observation could be the high educational levels of participants residing in the study area, where the literacy rate was more than 80%. Thus, it seems that participants' education levels helped them make the best use of damaging feedback. We have discussed these points on pg. 20 in the revised manuscript.

*Lastly: figures are enough rough and should be improved and better described in the captions.*

**Authors:** Thank you for your kind suggestions.

We have now improved the quality of figures and their captions in the modified manuscript. All other formatting errors and references are corrected in the revised version. Now, the manuscript has also been proofed for English grammar.

**Specific comments:**

*L138: I think you need to add that 0<M<1*

**Authors:** Thank you for this comment.

We have now added $0 \leq M \leq 1$ as per your kind suggestion in the revised manuscript on pg. 6.

*L162: It is not clear to me what the Total Estimated Hazard is. Please define it.*
*L162: Landslide Hazard Map: what is this? Not clear how this is related to the LSZ and to the THED. It is even not clear how the spatial probability is included in the tool. It is a single value or there is a map?*

**Authors:** Thank you for your kind comments.

We have now defined the Total Estimated Hazard (THED) as a rating of different locations on a Landslide Hazard Map and their surface area of coverage on pg. 7 of the revised manuscript.

Also, as part of our revision, we have now provided the THED scale in Table 1 (see pg. 7). From this table, the critical THED values (e.g., 3.5, 5.0, 6.5, and 8.0) were converted into a probability value by dividing with the highest THED value (= 11.0). Next, we used the LSZ map of the study area to find the surface area that was under a specific THED value and used this area to determine the cumulative probability density function for P(S). For example, if a THED of 3.5 has a 20% coverage area on LSZ, then the spatial probability is less than equal to 0.32 (=3.5/11.0) with a 20% chance. Similarly, if a THED of 5.0 has a 30% coverage area on LSZ, then the then the spatial probability is less than equal to 0.45 (=5.0/11.0) with a 50% chance (30% + 20%). Such calculations enabled us to develop a cumulative density function for P(S). In the ILS tool, a participant was assumed to belong to a location in the study area and this study area determined the P(S) value. This P(S) value stayed the same for this participant across her performance in the ILS tool (see pg. 7).

*L172: Is "become less than" correct? I suppose should be "become greater than". If not please try to explain why must be "less".*

**Authors:** Thank you for your kind comments.

Yes, becomes less than is correct and we have clarified this reasoning as a footnote on pg. 5 in the manuscript.

*L182: please change "their total wealth" with "the total wealth of the participants"*

**Authors:** Thank you for the comment.

To keep the grammar consistent, we have changed the sentence to the following, "The goal in ILS tool is to maximize one's total wealth, where this wealth is influenced by one's income, property wealth, and losses experienced due to landslides." (pg. 8)

*L207: "decision-maker". Are you meaning "participant"? If yes please change the text accordingly.*

**Authors:** Thank you.

We have replaced the term decision-maker with participant everywhere in the revised manuscript as per your suggestion.

*L241-243: the sentence is not clear. Please rephrase.*

**Authors:** Thank you for the comment.

We have rephrased the sentence to make it clearer in its meaning.

*L243: "see Figure 2": please explain how the figure helps in understanding the text.*

**Authors:** Thank you for the comment. Figure 2 (now Figure 3) in the revised manuscript shows the investment screen that were shown to participants in the feedback-present conditions.

We have now mentioned this explanation on pg. 12 in the manuscript.

*L262: "(W)": it is not immediate to understand that "W" is the parameter of the equation at page 4. Please number the equations and use those numbers in the text.*

**Authors:** Thank you for the comment.

We have addressed this comment in the revised manuscript (pg. 13) by stating the line with the equation number. Now, we state that, "the weight (W) parameter in the equation 1 of the ILS model was fixed at 0.7 across all conditions."

*L263: "was fixed to 0.8": in figure 2, W is 0.7.*

**Authors:** Thank you. We made a typo in the manuscript.

We have now fixed this typo and made W equal 0.7 in the manuscript (pg. 13).

*L302: the first sentence was already stated at the start of the 3.2 subsection.*

**Authors:** Thank you.

We have now removed this sentence to avoid repeated use.

*L313: please describe the meaning of the statistical parameters (derived from statistic tests) inside the brackets.*

**Authors:** Thank you for your kind comment. We performed analysis of variance statistical tests for evaluating our expectations. The F-statistics is the ratio of between-group variance and the within-group variance. The numbers in brackets after the F-statistics are the degrees of freedom (K-1, N - K), where K are the total number of groups compared and N is the overall sample size. The p-value indicates the evidence in favour of the null-hypothesis when it is true. We reject the null-hypothesis when p-value is less than the alpha-level (0.05). The $\eta^2$ is the proportion of variance associated with one or more main effects. It is a number between 0 and 1 and a value of 0.02, 0.13, and 0.26 measures a small, medium, or large correlation between the dependent and independent variables given a population size.

We have now mentioned these details as a footnote on pg. 15 in the manuscript.

*L330: what "CI" means?*

**Authors:** Thank you for the comment. CI stands for confidence interval value.

We have now added this full form on pg. 16 in the manuscript.

*L385-386: unclear. Please rephrase.*

**Authors:** Thank you for the comment.

We have rephrased these sentences to make their meaning clearer in the manuscript (pg. 21).

**General Comments:**

*The study described in the paper addresses an important and very relevant issue in natural disaster risk management – to explore potential ways to improve risk awareness and knowledge. The authors reported how they used feedback in an Interactive Landslide Simulator to influence people's risk reduction investment behavior. The manuscript was written generally in good English that can be relatively easily understood, but the ILS model still needs to be better elaborated and explained. While the study represent a good initiative, it also suffers from a number of design problems.*

**Authors:** Thank you for summarizing our contribution and providing encouragement to our work. We have now made several improvements to the manuscript based upon review comments from you and other reviewers. In agreement with different reviewers, we have now also extended this paper in both the design choices as well as system constraints. Now the elaboration of the ILS model has been improved and we have also explained the experiment design in detail. In the revised manuscript, we have also addressed several design problems related to participant demographics and details concerning assumptions in the ILS tool.

**Specific issues:**

*The ILS model and simulator structure Significant information about the ILS model was from the authors' published conference paper in 2016. The authors need clearly state this. Much of the information needs not to be repeated. Even so, the current description of the model is still not clear enough. More details are needed to help understand how the rather sophisticated landslide probability calculation relates to damage estimation. For example, the total P is an additive results of the two constituting components, P(I) and P(E), however P(E) is the multiplicative results of its two constituting components. The authors did not give full information to justify this choice. The authors mentioned "study area" only in 2.1.1, while very limited information was provided. The authors also did not give any explanation on how W is determined.*

**Authors:** Thank you for your kind comments.

In the revised manuscript, we have now given proper citation to our 2016 conference paper at different places in the manuscript (actually the year of publication of this conference paper is 2017 and not 2016 and the year has been corrected in the manuscript). Furthermore, we have now clarified the contribution in the manuscript and how this work builds upon prior work (pg. 3). In addition, we have now extended the paper to include a better description of relevant theory (pg. 2-3) and a better description of the probability calculation for P(S), P(R), and P(E) (pg. 5-7). As part of our revision, we have also suggested the rationale for different design and system choices made. In the revised version, we have explained study area by giving more details about its geographic location, climate, and demographic profile (pg. 3, 12-14).

The W is a free parameter. We have fixed the W parameter in this experiment such that human action play a significant role in the reduction of landslide risk (pg. 13). However, as a

part of our future research work with ILS tool, we will also vary the M and W parameters to see the effect of this variation on participants' investment decisions against landslides (pg. 20-22).

*The assumptions of the ILS model: The ILS was designed with the assumption that people susceptible to landslide hazard aims to maximize their total wealth and the authors started that "a high probability of landslide damages will make people suffer monetary losses and people would tend to minimize these losses by increasing their mitigation actions". This assumption neglects much of the social science research on people's risk perception, attitude and behavior, that people do not behave as an economic rationale individual in the face of extreme events.*

**Authors:** Thank you for providing valuable comments that helped to further improve our research.

First, we have now revised our expectations to be over time (pg. 3). Second, at a first glance, the expectations may seem to assume people to be economically rationale individuals while facing landslide disasters (Bossaerts and Murawski, 2015; Neumann and Morgenstern, 1947), where one disregards people's bounded rationality, risk perceptions, attitudes, and behaviours (De Martino, Kumaran, Seymour, and Dolan; 2005; Gigerenzer and Selten, 2002; Kahneman and Tversky, 1979; Simon, 1959; Slovic, Peters, Finucane, and MacGregor, 2005; Thaler and Sunstein, 2008; Tversky and Kahneman, 1992). However, in this paper, we consider people to be bounded rational agents (Gigerenzer and Selten, 2002; Simon, 1959), who tend to minimize their losses against landslides slowly over time via a trial-and-error learning process driven by personal experience in an uncertain environment (Dutt and Gonzalez, 2010; Slovic et al., 2005). We have now added these explanations on pg. 3 of the manuscript.

Furthermore, we now also discuss how the repeated experiential feedback likely enables learning by repeated trial-and-error procedures, where bounded-rational individuals (Simon, 1959) try different investment values in ILS and observe their effects on occurrence of landslides and their associated consequences. Also, we now mention that according to Slovic et al. (2005), loss-averse individuals tend to increase their contribution against a risk over time. In our case, similar to Slovic et al. (2005), participants started contributing slowly against landslides and, with the experience of landslide losses over time, they started contributing larger amounts to reduce landslide risks. These explanations have been discussed on pg. 20 of the manuscript.

*The authors assumed that "damages concerning injury and fatality affect one's income levels". This is rather naïve. While reduced income level is going to be a consequence, but it would be much less a concern for most people than the injury and fatality itself. In reality people can also choose to migrate when mitigation cost is too high and adaptation becomes impossible. The nature of landslide hazard, including its notorious fame of being extremely hard, if not impossible, to predict, makes it quite different from other hazards such as flood and drought, and general climate risk.*

**Authors:** Thank you for sharing this thought provoking comment. In agreement with you, we have now stated as part of our discussion section that currently, in the ILS model, we have assumed that damages from fatality and injury influence participants' daily-income levels. The reduced income levels do create adverse consequences, but one could also argue that

they would be much less of a concern for most people compared to the injury and fatality itself. Furthermore, people could also choose to migrate from an area when the landslide mitigation cost is too high and adaptation becomes impossible, especially due to the differences between the landslide hazard and other hazards such as flood, drought, and general climate risks. As part of our future research, we plan to investigate the influence of feedback that causes only injuries or fatalities compared to feedback that causes economic losses due to injuries and fatalities. Also, as part of our future research in the ILS tool, we plan to investigate people's migration decisions when the landslide mitigation costs are too high and adaptation to landslides is not possible.

These explanations have been provided as part of the discussion section in the manuscript (see pg. 21).

*The authors' choice of P(I) formula from Hasson et al. 2010 does not seem to be appropriate. It may seem to be obviously useful by applying specific parameters from the Mandi area in India as the participants seem to be mostly from the area (the authors did not clearly elaborate this), however, since the algorithms was not disclosed to the participants and a random number generator was used in producing damages, using the seemingly sophisticated algorithms is in fact not much related to the authors' main objective, instead, a more generic algorithm would serve the same purpose and potential be more useful for testing with participants from other areas.*

**Authors:** Thank you for your kind comments. In agreement with your suggestions, we have now stated as part of our discussion section that in the ILS model, we used a linear model to compute the probability of landslides due to human factors (i.e., Hasson et al. 2010's model). Also, the probabilistic equations governing the physical factors in the ILS model were not disclosed to participants, who seemed to possess high education levels. One could argue that there are several other linear and non-linear models that could help compute the probability of landslides due to human factors. Some of these models could not only influence the probability of landslides, but also the severity of consequences (damages) caused by landslides. Also, other generic models could account for the physical factors in the ILS tool. We plan to try these possibilities as part of our future work in the ILS tool. Specifically, we plan to assume different models of investments in the ILS tool and we plan to test them against participants with different education levels.

These explanations have been added to the discussion section (pg. 21).

Also, we have now clearly elaborated in the revised manuscript that the sample used in the experiment was representative of the study area's population because the literacy rate in the town and surrounding areas is quite high (81.5%) (Pg. 15).

*Day was used as the time unit for simulation and people make daily choice in landslide mitigation investment. This is not relevant for real world situation either.*

**Authors:** Thank you for your observation.

We have now stated as part of our methods section that the ILS tool can run for different time periods, which could be from days to months to years. This feature can be customized in the ILS tool (pg. 8). However, to showcase the potential of using ILS, the experiment used the

daily setting in the ILS tool. As part of our future research, we plan to extend this limitation by considering people to make decisions on a longer time scale ranging from months to years. Please see this discussion in the discussion section of the manuscript (pg. 20-22).

*In most cases, especially in developing countries, households and communities themselves almost never have resources substantial enough to mitigate landslide risk, which is often financed by government and/or international donors. The huge disparity between the average asset (calculated as per capital GDP) and the salary (with the former being 2000 times of a person's annual income) also supported my above statement.*

**Authors:** Thank you for your kind comments. In agreement with you, we have now added to our discussion section that we assumed a large disparity between a participant's property wealth and her daily income. In addition, as part of the ILS model, we did not consider any support from government or international agencies against damages from landslides. As suggested by you, in certain cases, especially in developing countries, mitigation of landslide risks may be often financed by government or international agencies. As part of our future work, we plan to extend the ILS model to include assumptions of contributions from government or international agencies. Such assumptions will help us determine the willingness of common people to contribute against landslide disasters, which is important as the developing world becomes developed over time.

These comments have been reported on pg. 22.

*The authors chose a value of 0.8 for W, indicating that the landslide risk can largely be mitigated by human. This is in general not the case, especially for the type of mitigation measures mentioned by the authors – tree plantation. There has been studies showing that afforestation does not help with landslides in similar areas to Mandi in the Sivalik Hills.*

**Authors:** Thank you.

Now, as part of our discussion section (see pg. 22), we have mentioned that these W and M values indicated that landslide risks could largely be mitigated by human actions. However, in agreement with your suggestions, this assumption may not be the case always, especially for mitigation measures like tree plantations. For example, afforestation alone may not help in reducing deep-seated landslides in hilly areas (Forbes, 2011). Thus, it would be worthwhile investigating as part of future research on how people's decision-making evolves in conditions where investments likely influence the landslide probability (higher values of W and M parameters) compared to conditions where investments unlikely influence the landslide probability (lower values of W and M parameters).

**3. The study design**

*The high damage scenario is simply not realistic at all. With such a high risk of mortality and 90% change of injury, no one would still choose to stay in the landslide area, even in least developed countries. The low damage scenario would already be a very high risk area in reality, in any countries.*

**Authors:** Thank you.

In agreement with your suggestions, we have now mentioned as part of our discussion section (see pg. 22) that to test our hypotheses, we presented participants with a high damage scenario and a low damage scenario, where the probability of property damage, injury, and fatality were high and low, respectively. However, such scenarios may not be realistic, where people may want migrate from both low and damage areas in even the least developed countries. In future research with ILS, we plan to calibrate the probability of damages, injury, and fatality to realistic values and test the effectiveness of ILS in improving the participants' investment decision making.

*In Fig. 3b, the authors give a smiling face followed by "Landslide did not Occur". This gives a false feeling that the fact that landslide did not occur because of mitigation investment, while in reality much of it should be due to stochastic in the nature of landslide.*

**Authors:** Thank you for your kind comments. In our experiment, when landslide did not occur and experiential feedback was present, people were presented with a smiling face followed by a message. The message and emoticon were provided to connect the cause-and-effect relationships for participants in the ILS tool. However, it could also be that the landslide did not occur on a certain trial due to the stochasticity in the simulation rather than participants' investment actions. Although such situations are possible over shorter time-periods, however, over longer time-periods increased investments from people will only reduce the probability of landslides.

In agreement with your comments, we have now added these explanations as part of the discussion section (pg. 22).

**4. The results**

*First, part of the results were already included in the 2016 paper (apparently including 43 of the 83 participants reported in this study) and this should be fully disclosed.*

**Authors:** Thank you.

In the revised manuscript, we have now given proper citation to our 2016 conference paper (actually 2017 conference paper, where the year has been corrected). We have now clarified the contribution in the manuscript and how this work builds upon the prior 2017 work (pg. 3, 12).

Also, via a footnote on pg. 12, we have mentioned that data reported in Chaturvedi et al. (2017) has been included in this paper with two more conditions, the high-damage feedback-absent (N = 20) and the low-damage feedback-absent (N = 20). Data in all four conditions was collected simultaneously.

*Second, the part of the results on people's increasing investment in mitigation seems to be largely an artifact of the choice of M being 0.8. It'd be more interesting to study, with a much larger sample, how how changing M will affect people's behavior, given that the authors choose more realistic scenarios.*

**Authors:** Thank you for your kind comment, which helped us get new ideas for our research. In agreement with you, we have now mentioned that in the experiment, we assumed a value of

0.8 for the return to mitigation (M) parameter. This M value indicated that landslide risks could largely be mitigated by human actions. However, this assumption may not be the case always, especially for mitigation measures like tree plantations. For example, afforestation alone may not help in reducing deep-seated landslides in hilly areas (Forbes, 2011). Thus, it would be worthwhile investigating as part of future research on how people's decision-making evolves in conditions where investments likely influence the landslide probability (higher values of M parameter) compared to conditions where investments unlikely influence the landslide probability (lower values of M parameter).

This discussion appears on pg. 22 of the manuscript.

**Some detailed comments on texts:**

1. *In Abstract, the first sentence is incomplete.*

`

**Authors:** Thank you.

In the revised manuscript, we have now improved the first line of abstract. All other formatting errors and references are corrected in the revised version. Now, the manuscript has also been proofed.

2.*"Different amount of feedback" was used, but in fact the difference between the two different levels of feedback may better be described as "intensity" of "strength" of feedback.*

**Authors:** Thank you.

In agreement with your kind suggestion, we have now changed the "amount of feedback" in the paper everywhere to the "strength of feedback."

3. *Fig. 2 is similar to the Fig 2 in the authors' 2016 conference paper and needs to be disclosed.*

**Authors:** Thank you.

In the revised manuscript, we have now given proper citation to our 2017 conference paper as part of this figure.

4. *Fig. 5b, it should be high/low damage instead of more/less damage.*

**Authors:** Thank you.

In the revised manuscript, we have now rectified this error.

5. *Reference – Mathew et al. was published in 2014 and should be rearranged in alphabetic order.*

**Authors:** Thank you.

In the revised manuscript, all the formatting errors and references are corrected.

*While the study represents an interesting attempt, it suffers from seriously false model assumptions and weakness in study design in relation to reality. I personally even think that the simulator may falsely influence participants in terms of how they should make decisions in the face of landslide risk. But I strongly recommend the authors to continue developing the simulator with stronger social science understanding and better design.*

**Authors:** We are thankful for your kind comments as they helped us provide an improved exposition of our methods and results. These comments have also given a lot of new ideas which we will use in our future experimentation with ILS tool. We, hereby want to clarify that current experimental study with ILS was a preliminary but important work to test the effectiveness of simulation models on people's understanding of landslide risks. But, in future we will use several of the manipulations in the model parameters and probabilities to make the simulation exercise more realistic.

In agreement with you, we have now added several ideas suggested by you as part of our discussion section in the manuscript (pg. 20-22).

**Learning in an Interactive Simulation Tool against Landslide Risks: The Role of Strength and Availability of Experiential Feedback**

**Pratik Chaturvedi**[1, 2], **Akshit Arora**[1, 3], **and** **Varun Dutt**[1]

[1]Applied Cognitive Science Laboratory, Indian Institute of Technology, Mandi- 175005, India

[2]Defence Terrain Research Laboratory, Defence Research and Development Organization, Delhi -110054, India

[3]Computer Science and Engineering Department, Thapar University, Patiala - 147004, India

*Correspondence to*: Pratik Chaturvedi (prateek@dtrl.drdo.in)

**Abstract.** Feedback via simulation tools is likely to help people improve their decision-making against natural disasters; however, currently little is known on how differing strengths of experiential feedback and feedback's availability in simulation tools influences people's decisions against landslides. In an experiment involving participants, we tested the influence of differing strengths of experiential feedback and feedback's availability on people's decisions against landslide risks in an Interactive Landslide Simulation (ILS) tool. Experiential feedback (high or low) and feedback's availability (present or absent) were varied across four between-subject conditions: high-damage feedback-present, high-damage feedback-absent, low-damage feedback-present, and low-damage feedback-absent. In high-damage conditions, the probabilities of damages to life and property due to landslides were 10-times higher than those in the low-damage conditions. In feedback-present conditions, experiential feedback was provided in numeric, text, and graphical formats in ILS. In feedback-absent conditions, the probabilities of damages were described; however, there was no experiential feedback present. Investments were greater in conditions where experiential feedback was present and damages were high compared to conditions where experiential feedback was absent and damages were low. Furthermore, only high-damage feedback produced learning in ILS. Simulation tools like ILS seem appropriate for landslide risk communication and for performing what-if analyses.

**1  Introduction**

Landslides cause massive damages to life and property worldwide (Chaturvedi and Dutt, 2015; Margottini et al., 2011). Imparting knowledge about landslide causes-and-consequences as well as spreading awareness about landslide disaster mitigation are likely to be effective ways of managing landslide risks. The former approach supports structural protection measures that are likely to help people take mitigation actions and reduce the probability of landslides (Becker et al., 2013; Osuret et al., 2016; Webb and Ronan, 2014). In contrast, the latter approach likely reduces people's and assets' perceived vulnerability to risk. However, it does not influence the physical processes. One needs effective landslide risk communication systems (RCSs) to educate people about cause-and-effect relationships concerning landslides (Glade et al., 2005). To be effective, these RCSs should possess five main components (Rogers and Tsirkunov, 2011): monitoring; analysing; risk communication; warning dissemination; and capacity building.

Deleted: This research worko investigates how differing amounts of experiential feedback and feedback's availability in an interactive simulation tool influences people's decision-making against landslide risks.

Deleted: Knowledge about causes-and-consequences of landslides and awareness about landslide disaster mitigation are likely to help people take good mitigation actions that prevent landslides from occurring (Becker , Paton, Johnston, & Ronanet al., 2013; Osuret et al., 2016; Webb &and Ronan, 2014). Imparting knowledge about causes-and-consequences as well as spreading awareness about landslide disaster mitigation are two different ways of managing landslide risks. The former supports structural protection measures that reduce the probability of landslides. In contrast, the latter 
[revised manuscript text omitted]
. Our current research was a preliminary work and the assumptions made by us in ILS model may not be realistic, but in future, we will manipulate the probabilities related to landslide and damages caused to see effects of different settings of ILS on participants' risk perception, attitude and behaviour. However, up to certain extent, we were able to capture the people's behavior.

Moved up [2]: According to

Deleted: The ILS tool's parameter settings could be customized to a certain geographical area over a certain time period of play. In addition, the ILS tool could be used to present investment actions of other decision-makers (e.g., society or neighbours) compared to one's own investment actions. The presence of investment of other … [6]

In the current experiment, we assumed a large disparity between a participant's property wealth and her daily income. In addition, as part of the ILS model, we did not consider any support from government or international agencies against damages from landslides. In certain cases, especially in developing countries, mitigation of landslide risks may often be financed by government or international agencies. As part of our future work, we plan to extend the ILS model to include assumptions of contributions from government or international agencies. Such assumptions will help us determine the willingness of common people to contribute against landslide disasters, which is important as the developing world becomes developed over time.

To test our hypotheses, we presented participants with a high damage scenario and a low damage scenario, where the probabilities of property damage, injury, and fatality were high and low, respectively. However, such scenarios may not be realistic, where people may want to migrate from both low and damage areas in even the least developed countries. In future research with ILS, we plan to calibrate the probability of damages, injury, and fatality to realistic values and test the effectiveness of ILS in improving the participants' investment decision making.

Furthermore, in our experiment, when landslide did not occur and experiential feedback was present, people were presented with a smiling face followed by a message. The message and emoticon were provided to connect the cause-and-effect relationships for participants in the ILS tool. However, it could also be that the landslide did not occur on a certain trial due to the stochasticity in the simulation rather than participants' investment actions. Although such situations are possible over shorter time-periods, however, over longer time-periods increased investments from people will only reduce the probability of landslides.

In this paper, the experiment used a daily investment setting in the ILS tool. However, the ILS tool can easily be customized to different time periods ranging from seconds, minutes, hours, days, months, and years. As part of our future research, we plan to extend the daily assumption by considering people making decisions on longer time-scales ranging from months to years. In addition, in the experiment, we assumed a value of 0.7 and 0.8 for the weight (W) and return to mitigation (M) parameters. These W and M values indicated that landslide risks could largely be mitigated by human actions. However, this assumption may not be the case always, especially for mitigation measures like tree plantations. For example, afforestation alone may not help in reducing deep-seated landslides in hilly areas (Forbes, 2013). Thus, it would be worthwhile investigating as part of future research on how people's decision-making evolves in conditions where investments likely influence the landslide probability (higher values of W and M parameters) compared to conditions where investments unlikely influence the landslide probability (lower values of W and M parameters). Some of these ideas form the immediate next steps in our ongoing research program on landslide risk communication.

[revised manuscript text omitted]

in ILS tool

| **Page 20: [2] Deleted** | **Varun Dutt** | **09/12/17 9:04 PM** |
|---|---|---|

in ILS tool

| **Page 20: [2] Deleted** | **Varun Dutt** | **09/12/17 9:04 PM** |
|---|---|---|

in ILS tool

| **Page 20: [2] Deleted** | **Varun Dutt** | **09/12/17 9:04 PM** |
|---|---|---|

in ILS tool

| **Page 20: [2] Deleted** | **Varun Dutt** | **09/12/17 9:04 PM** |
|---|---|---|

in ILS tool

| **Page 20: [2] Deleted** | **Varun Dutt** | **09/12/17 9:04 PM** |
|---|---|---|

in ILS tool

| **Page 20: [2] Deleted** | **Varun Dutt** | **09/12/17 9:04 PM** |
|---|---|---|

in ILS tool

| **Page 20: [2] Deleted** | **Varun Dutt** | **09/12/17 9:04 PM** |
|---|---|---|

in ILS tool

| **Page 20: [2] Deleted** | **Varun Dutt** | **09/12/17 9:04 PM** |
|---|---|---|

in ILS tool

| **Page 20: [2] Deleted** | **Varun Dutt** | **09/12/17 9:04 PM** |
|---|---|---|

in ILS tool

| **Page 20: [2] Deleted** | **Varun Dutt** | **09/12/17 9:04 PM** |
|---|---|---|

in ILS tool

| **Page 20: [2] Deleted** | **Varun Dutt** | **09/12/17 9:04 PM** |
|---|---|---|

in ILS tool

| **Page 20: [2] Deleted** | **Varun Dutt** | **09/12/17 9:04 PM** |
|---|---|---|

in ILS tool

| **Page 20: [2] Deleted** | **Varun Dutt** | **09/12/17 9:04 PM** |
|---|---|---|

in ILS tool

| **Page 20: [2] Deleted** | **Varun Dutt** | **09/12/17 9:04 PM** |
|---|---|---|

in ILS tool

| **Page 20: [3] Deleted** | **Varun Dutt** | **09/12/17 9:16 PM** |
|---|---|---|

people who are loss averse,

| **Page 20: [3] Deleted** | **Varun Dutt** | **09/12/17 9:16 PM** |
|---|---|---|

people who are loss averse,

| **Page 20: [3] Deleted** | **Varun Dutt** | **09/12/17 9:16 PM** |
|---|---|---|

people who are loss averse,

| **Page 20: [3] Deleted** | **Varun Dutt** | **09/12/17 9:16 PM** |
|---|---|---|

people who are loss averse,

| Page 20: [3] Deleted | Varun Dutt | 09/12/17 9:16 PM |

people who are loss averse,

| Page 20: [3] Deleted | Varun Dutt | 09/12/17 9:16 PM |

people who are loss averse,

| Page 20: [3] Deleted | Varun Dutt | 09/12/17 9:16 PM |

people who are loss averse,

| Page 20: [3] Deleted | Varun Dutt | 09/12/17 9:16 PM |

people who are loss averse,

| Page 20: [3] Deleted | Varun Dutt | 09/12/17 9:16 PM |

people who are loss averse,

| Page 20: [3] Deleted | Varun Dutt | 09/12/17 9:16 PM |

people who are loss averse,

| Page 20: [3] Deleted | Varun Dutt | 09/12/17 9:16 PM |

people who are loss averse,

| Page 20: [3] Deleted | Varun Dutt | 09/12/17 9:16 PM |

people who are loss averse,

| Page 20: [3] Deleted | Varun Dutt | 09/12/17 9:16 PM |

people who are loss averse,

| Page 20: [3] Deleted | Varun Dutt | 09/12/17 9:16 PM |

people who are loss averse,

| Page 20: [4] Deleted | Varun Dutt | 03/12/17 3:10 PM |

| Page 20: [4] Deleted | Varun Dutt | 03/12/17 3:10 PM |

| Page 20: [4] Deleted | Varun Dutt | 03/12/17 3:10 PM |

| Page 20: [4] Deleted | Varun Dutt | 03/12/17 3:10 PM |

| Page 20: [4] Deleted | Varun Dutt | 03/12/17 3:10 PM |

| Page 20: [5] Deleted | Varun Dutt | 03/12/17 3:13 PM |

.

| Page 20: [5] Deleted | Varun Dutt | 03/12/17 3:13 PM |

.

| Page 20: [5] Deleted | Varun Dutt | 03/12/17 3:13 PM |

.

| Page 20: [5] Deleted | Varun Dutt | 03/12/17 3:13 PM |

.

| Page 20: [5] Deleted | Varun Dutt | 03/12/17 3:13 PM |

.

| Page 20: [5] Deleted | Varun Dutt | 03/12/17 3:13 PM |

.

| Page 20: [5] Deleted | Varun Dutt | 03/12/17 3:13 PM |

.

| Page 20: [5] Deleted | Varun Dutt | 03/12/17 3:13 PM |

.

| Page 20: [5] Deleted | Varun Dutt | 03/12/17 3:13 PM |

.

| Page 20: [5] Deleted | Varun Dutt | 03/12/17 3:13 PM |

.

| Page 20: [5] Deleted | Varun Dutt | 03/12/17 3:13 PM |

.

| Page 20: [5] Deleted | Varun Dutt | 03/12/17 3:13 PM |

.

| Page 21: [6] Deleted | \pratik | 27/11/17 7:13 PM |

The ILS tool's parameter settings could be customized to a certain geographical area over a certain time period of play. In addition, the ILS tool could be used to present investment actions of other decision-makers (e.g.,

society or neighbours) compared to one's own investment actions. The presence of investment of other decision-makers in addition to one's own decisions will likely enable the use of social norms towards learning (Schultz et al., 2007). These features makes ILS tool very attractive for landslide education in communities in the future. Furthermore, the ILS tool holds a great promise for policy-research against landslides. For example, in future, researchers may vary different system-response parameters in ILS (e.g. weight of one's decisions and return to mitigation actions) and feedback (e.g. numbers, text messages and images for damage) in order to study their effects on people's decisions against landslides. Here, researchers could evaluate differences in ILS's ability to increase public contributions in the face of other system-response parameters and feedback. In addition, researchers can use the ILS tool to do "what-if" analyses related to landslides for certain time periods and for certain geographical locations. The ILS tool has the ability to be customized to certain geographical area as well as certain time periods, where spatial parameters (e.g., soil type and geology) as well as temporal parameters (e.g., daily rainfall) can be defined for the area of interest. Once the environmental factors have been accounted for, the ILS tool enables researchers to account for assumptions on human factors (contribution against landslides) with real-world consequences (injury, fatality, and infrastructure damage). Such assumptions may help researchers model human decisions in computational cognitive models, which are based upon influential theories of how people make decisions from feedback (Dutt & Gonzalez, 2012; Gonzalez & Dutt, 2011). In summary, these features make ILS tool apt for policy research, especially for areas that are prone to landslides. This research will also help test the ILS tool and its applicability in different real-world settings.

| Page 21: [7] Deleted | Varun Dutt | 09/12/17 4:07 PM |
|---|---|---|

we will try to find without causing reduction in income, only due to fatality and injury what effect it have on participants' investment

| Page 21: [8] Deleted | Varun Dutt | 09/12/17 4:09 PM |
|---|---|---|

Another idea is to test whether people would continue to invest large money or choose to migrate.

| Page 21: [9] Deleted | Varun Dutt | 09/12/17 4:10 PM |
|---|---|---|

This idea is very interesting to study because The nature of landslide hazard, including its notorious fame of being extremely hard, if not impossible, to predict, makes it quite different from other hazards such as flood and drought, and general climate risk.

| Page 21: [10] Deleted | Varun Dutt | 09/12/17 4:49 PM |
|---|---|---|

to calculate P(I) to showcase the potential of using ILS in the real-world

| Page 26: [11] Deleted | \pratik | 25/12/17 1:30 PM |
|---|---|---|

Chaturvedi, P., Arora, A., Dutt, V.Interactive Landslide Simulator: A Tool for Landslide Risk Assessment andCommunication n Advances in Applied Digital Human Modeling and Simulation

Slovic, P., Peters, E., Finucane, M. L., & MacGregor, D. G. (2005). Affect, Risk, and Decision Making. Health Psych, 24(4), pp.S35-S40. H. and Sunstein, C. R.Improving Decisions About Health, Wealth, and Happiness. Yale University PressNew Haven, USA,  Webb, M., & Ronan, K. R. (2014). Interactive Hazards Education Program for Youth in a Low SES Community: A Quasi-Experimental Pilot Study. Risk analysis, 34(10), 1882-1893.

**Appendix A**

**Instructions of the Experiment**

Welcome!

You are a resident of Mandi district of Himachal Pradesh, India, a township in the lap of Himalayas. You live in an area that is highly prone to landslides due to a number of environmental factors (e.g., the prevailing geological conditions and rainfall). During the monsoon season, due to high intensity and prolonged period of rainfall, a number of landslides may occur in the Mandi district. These landslides may cause fatalities and injuries to you, your family, and to your friends, who reside in the same area. In addition, landslides may also damage your property and cause loss to your property wealth.

This study consists of a task, where you will be making repetitive decisions to invest money in order to mitigate landslides. Every trial, you'll earn certain money between 0 and 10 points. This money is available to you to invest against landslides. You may invest certain amount from the money available to you; however, if you do not wish to invest anything, you may invest 0.0 against landslides on a particular trial. Based upon your investment against landslides, you'll get feedback on whether a landslide occurred and whether there was an associated loss of life, injury, or property damage (all three events are independent and they can occur at the same time).

**Your total wealth at any point in the game is the following: sum of the amounts you did not invest against landslides across days + your property wealth - damages to you, your family, your friends, and to your property due to landslides**. Your property wealth is assumed to be 100 points at the start of the game. The amount of money **not invested against landslides** increases your total wealth. **Your goal is to maximize your total wealth in the game**.

Whenever a landslide occurs, if it causes fatality, then your daily earnings will be reduced by 5% of its present value at that time and if landslide causes injury to someone, then the daily earnings willbe reduced by 2.5% of its present value at that time. Thus, the amount available to you to invest against landslides will reduce with each fatality and injury due to landslides. Furthermore, if a landslide occurs and it causes property damage, then your property wealth will be reduced by 80% of its present value at that time; however, the money available to you to invest against landslides due to your daily earnings will remain unaffected.

Generally, landslides are triggered by two main factors: environmental factors (e.g., rainfall; outside one's control) and investment factors (money invested against landslides; within one's own control). The total probability of landslide is a weighted average of probability of landslide due to environment factors and probability of landslide due to investment factors. The money you invest against landslides reduces the probability of landslide due to investment factors and also reduces the total probability of landslides. However, the money invested against landslides is lost and it cannot become a part of your total wealth.

At the end of the game, we'll convert your total wealth into INR and pay you for your effort. For this conversion, a ratio of 100 total wealth points = INR 1 will be followed. In addition, you will be paid INR 30 as base payment for your effort in the task. Please remember that your goal is to maximize your total wealth in the game.

Starting Game Parameters

Your wealth: **20 Million**

When a landslide occurs:

If a death occurs, your daily income will be reduced by **50%** of its current value.

If an injury takes place, your daily income will be reduced by **25%** of its current value.

If a property damage occurs, your wealth will be reduced by **50%** of your property wealth.

**Best of Luck!**

---

## Author Response (AR2)

March 18$^{th}$, 2018

Dr. Stefano Luigi Gariano
Editor, *Journal of Natural Hazards and Earth System Sciences*

Dear Dr. Gariano:

I write to you concerning a manuscript, "Learning in an Interactive Simulation Tool against Landslide Risks: The Role of Strength and Availability of Experiential Feedback," that I co-authored with my Ph.D. advisor, Dr. Varun Dutt and Mr. Akshit Arora.

We want to thank you for considering our work to the special issue on "Landslide early warning systems: monitoring systems, rainfall thresholds, warning models, performance evaluation and risk perception" of NHESS journal.

As per your kind suggestions, we have now extensively modified our manuscript as per your and referee comments. We are now submitting a revised version of the manuscript with point-to-point replies against the comments and suggestions given by you and referees. These point-to-point replies are attached with this covering letter as an annexure "A".

We would like to mention that our focus in the manuscript was not to develop a very precise model of landslide occurrence, which considers all possible technical-social-economic parameters. Rather, our focus was on developing approximate models, which could be used in improving people's understanding about landslide disasters. We have mentioned this part both in response to referee comments as well as in the manuscript draft.

We look forward to hearing from you on our revision.

Sincerely,

Pratik Chaturvedi

Ph.D. Scholar, School of Computing and Electrical Engineering

Indian Institute of Technology Mandi

Kamand-175005, Himachal Pradesh, India

Phone: +91-931-313-1129

Email: prateek@dtrl.drdo.in

**Comments to the Author:**

Dear Authors,

your paper passed through a second round of review and received again constructive comments.

I still believe that your paper has potential but need a huge improvement before being considered publishable. Some drawbacks remains in your paper, both in the contents and in the presentation.

I suggest to reconsider all the comments made by reviewers in the first round of revision and to address all of them. In some case, you did not provide precise answers to the referees' comments.

Moreover, I suggest to follow the suggestions proposed by the referees in the second round of revision. In particular, please:

**i) include a section related to previous events in the area;**

**ii) justify the values of weights adopted to calculate hazard probability;**

**iii) clarify the link between hazard and risk;**

**iv) improve the discussion of the methods used for assessing landslide probability and susceptibility;**

**v) discuss other economic issues (such as the sensitivity of the different social classes in relation to their economic and cultural resources; the public/private intervention in the investments; and a cost-threshold).**

Finally, I suggest an amelioration of the manuscript presentation, in particular regarding the figures, which are still improvable.

My decision is that your manuscript need again major revisions before being reconsidered for publication. The reviewed manuscript will be checked again by referees and editor before proceeding for next steps.

**Authors:** Thank you for summarizing our contribution and providing encouragement to our work. As per the referees' comments, we have now clarified

several improvements to the manuscript based upon review comments in the second round. In agreement with reviewers, we have now also added a section about the study area (previous landslide events, their features, and associated damages) in this paper (pgs. 5-6). We have also added a simulation result showing effect of weight parameter on landslide probability calculation and how having high values of W parameter gave salient feedback on actions to our participants (pgs. 19-20). We have now clarified the link between hazard and risk in the revised manuscript (pg. 7). As per our paper, risk = hazard = probability * consequence. Furthermore, the damage modelling and spatial probability calculation from hazard maps in ILS tool have been better elaborated and explained (pgs. 11-13). As per your suggestion, we have also explained the economic issues related to model used in ILS. To express the limitations of the current ILS model, we have now added a separate section titled "Limitations" before the "Conclusions" section (pgs. 29-31). Finally, we have now also improved the quality of figures in the revised manuscript.

**Point by point replies to referee 1**

**General Comments**

I confirm my positive feelings about the significance of investigated topic and the type of approach used in this perspective. In some parts, on my view, the paper results highly improved after the revision but on the other sections, some weaknesses persist. For example, a section about previous events, suffered damages and features of landslides in the area is again lacking. Moreover, several weights adopted to calculate hazard probability are not properly justified and the results could be highly dependent by such choices (for example, for W parameter). Furthermore, the link between hazard and risk should be clarified (see section "Damages due to landslides"). In the revised version, all the assumptions result clearly reported in Conclusions. Nevertheless, a section devoted to them could be added prior to the "Conclusions".

**Authors:** Thank you for summarizing our contribution and providing encouragement to our work. As per your review comments, we have now clarified several improvements to the manuscript based upon review comments. In agreement with you, we have now added a section about the study area listing previous landslide events, their features, and associated damages (pgs. 5-6). Also, we have now added a simulation result showing the effects of weight parameter on landslide probability calculation (pgs. 19-20). Specifically, we have now shown that a high W value allowed us to give human participants salient feedback on how their decisions changed total probability of landslides (pgs. 19-20). Furthermore, we have now explained how we computed the damages, rainfall probability of landslides, and spatial probability of landslides in the study area (pgs. 9-12). As per your kind suggestion, we have now explained all the assumptions in the current study in detail in a limitation separate section prior to conclusions (pgs. 29-31).

**Specific comments:**

**Abstract:** again, the text should be improved. Too few details were given about the approaches, the context, and the scopes of the work. On the other side, some findings

are reported also if they could be not clearly understood by the readers at this stage. I suggest you reorganizing the text to improve its appealing.

**Authors:** Thank you for your kind suggestions. We have now improved the language of the abstract section to improve its appeal and understandability.

**L10:** please divide the two parts of the sentence.

**Authors:** Thank you for your kind suggestion. We have now divided this sentence into two parts as suggested by you.

**L64-65:** it practically replies the previous sentence; in my view, it could be removed

**Authors:** Thank you for your kind suggestion. We have now removed the extra sentence as per the suggestion in the revised manuscript.

**L79:** probably, other anthropic interventions could be cited. The word "likely" appears overused in the text; please check and, when possible, use synonyms.

**Authors:** Thank you for your kind suggestions. In agreement with you, we have now added the following anthropogenic interventions: a retaining wall, planned road construction, provision of proper drainage, and planting of crops with long roots. Furthermore, we have now cut the repetitive use of the word "likely" in the text in the revised manuscript.

**Figure 1:** please, attempt improving it in quality; moreover, check capital letters and punctuation.

**Authors:** We have now improved the figure quality, corrected the use of capital letters and other typos in the revised manuscript.

**L127:** I suggest reporting the footnote directly in the text; moreover, reducing landslide risk could entail also avoiding profitable investments (then loss of revenue); it should be cited in the text.

**Authors:** Thank you for your kind suggestions. We have now removed the footnotes everywhere and made them part of the main text. In agreement with you, we have now mentioned that by reducing the landslide risk, people also have lesser ability in investing in other profitable investments (pg. 7).

**L157-158:** the sentence should be clarified; at the moment, it is not really clear the meaning of M.

**Authors:** The meaning of the return to mitigation parameter (M) has been clarified by an example in the main text (pg. 9).

**L180-182:** the approach does not result consistent; five years represent a too short interval to climatologically characterize an area. According WMO indications, 30 years are required. Moreover, linking to a single day could be not a proper way as rainfall patterns could be characterized at seasonal scale but not for single rainfall days.

**Authors:** We plotted the distribution of rainfall averaged over a 30-year period and this distribution was similar to the one that was computed over the past 5-year period.

There was a peak during the monsoon months (mid-June to mid-September) in the average rainfall distribution.

As our primary objective was not to accurately predict rainfall or other landslide parameters; rather, to educate people about landslide disasters, we have used an approximate model of the complex landslide phenomena in the ILS simulation tool. There is a large body of literature on using approximate simulation tools for improving people's understanding about natural processes like climate change and other natural disasters. The development and use of the ILS tool follows the same line of research and tradition. We have now added these arguments to a "Limitations" section in the paper before the conclusions section (pgs. 29-31).

**L190-198:** please avoid confusing between susceptibility and hazard in the text; if you consider susceptibility, occurrence probability is not taken into account; please display a map with landslide susceptibility map.

**Authors:** Thank you for your comments. We used the landslide hazard zonation map (now shown as Figure 4 (A) in the manuscript) for computing the spatial probability of landslide occurrence. We have now clarified our method of computing the spatial probability using the area and the colors on the map in the manuscript (pgs. 11-13). We have also shown the cumulative density function for the landslide spatial probability that we used in the ILS tool (Figure 4 (B)). As explained in the manuscript, we have used a point value of the spatial probability from this cumulative distribution for each participant in the ILS tool. Again, please note that this exercise was not meant to accurately determine the spatial probability of landslide in the area of interest. Rather, the primary objective of this exercise was to develop an approximate model that accounts for the spatial probability in the in the ILS, where the tool was meant to improve people's understanding about landslide risks (pgs. 13 and 29-31).

**L200-208:** do the areas account for invaded zones?

**Authors:** We have now added the details of the study area in a separate section to the manuscript to improve the depiction of the invaded zones (pg. 5-6).

**L211-220:** the dynamics regulating "Damages due to landslides" are not clearly reported at the moment; please attempt improving them; in the specific, the role of random U and in which way it is related to total probability.

**Authors:** Thank you for your comments. We have now improved the exposition of the dynamic aspects in the ILS tool in the manuscript (pg. 8-13). Specifically, we have now explained that the use of a uniformly distributed random variable was to simulate the total probability of landslides as well as the probability of damages due to landslides (pg. 8).

**L225:** I'm not native English speaker but I suppose that "to help" requires –ing form after.

**Authors:** Thank you for your kind suggestions. We have now corrected the grammatical mistakes and other typos in the revised manuscript.

**L252-256:** Is it taken into account where the mitigation measurements are designed to be performed? (e.g. where the player decides the investments).

**Authors:** Thank you for your kind comments. As part of the instructions, the players were told that the mitigation measures will be taken close to the places where they reside in the district in the ILS tool. We have now added this detail in the methods section of the manuscript (pg. 14).

**L256-265:** to take into account damages, could you refer to previous landslide events monitored in the area?

**Authors:** As suggested by you, we have used Prakash (2011) to estimate the historical damages in the study area. However, please note that, in this manuscript, we vary this probability in the experiment. Thus, the exact value of the probability from literature is not required in the simulation (pg. 13). However, we have tried to keep the probability values to those found by Prakash (2011) for this study area.

**L322-324:** you should carefully justify this choice

**Authors:** As suggested by you, we have now carefully justified our choice of values on these lines (pgs. 19-20).

**Figure 5:** the "Instructions" could be reported as Appendix to make them clearer.

**Authors:** Thank you for your kind suggestions. We have now removed instructions in figure 5 and report them as a part of the appendix section as per your kind suggestions.

**L363**: do you confirm that these percentages reflect those of communities affected by landslides in the area?

**Authors:** Yes, as suggested by our citations, we feel that people residing in the study area are well educated. Thus, these percentages more or less reflect the general population of the area (pg. 21).

**Footnote7:** on my view, you should avoid the use of footnotes not commonly used in NHESS research papers

**Authors:** Thank you for your kind suggestions. We have now removed the footnotes everywhere and made them part of the main text to match the format of NHESS.

**L436:** please check the use of Capital letters.

**Authors:** Thank you for your kind suggestions. We have now corrected the use of capital letters and other typos.

**Point by point replies to referee 2**

**The authors present a simulator (ILS model) that aims to evaluate the interactions between economic investments of the population and reduction of the risk related to landslides. Despite the bibliographic citations, the model is presented inadequately and it is not clear whether the results obtained can be subjected to a validation phase.**

**Authors:** Thank you for summarizing our contribution and providing encouragement to our work. As per your review comments, we have now clarified several improvements to the manuscript based upon your comments. We have improved the description about the model behind the ILS tool in the revised manuscript (pgs. 9-13) so that it can be subjected to a validation phase. Furthermore, now the damage modelling in ILS tool have been better elaborated and explained (pg. 13). In addition, to express the limitations of the current ILS model, we have now added a separate section titled "Limitations" before the "Conclusions" section in the manuscript (pgs. 29-31).

**Moreover, many aspects relating to the sensitivity of the different social classes in relation to their economic and cultural resources are neglected. The problem of the amount of the necessary economic investments, that imply an indispensable public intervention instead of the private one, is not addressed. It would therefore be appropriate to indicate a cost-threshold for interventions which, when exceeded, causes the model stop.**

**Authors:** Thank you for your comments. We have now discussed the sensitivity of different social classes to their economic, geographical, and cultural resources, where, in the absence of insurance payments, people mostly use their own finances to overcome the challenges put by natural disasters like landslides (ICICI, 2018) (pg. 7 and pg. 30). Furthermore, as part of "Limitation" section, we have now compared large public investments against landslides with respect to people's other investments and the reasons for the same (pg. 29-31). Also, we have now discussed future research where we plan to compare public investments against landslides to

their likely socio-economic cost thresholds given that people may need to spend their wealth in other areas (pgs. 29-31).

**The methods for assessing the probabilities and the susceptibility to landslides are presented inadequately.**

**Authors:** The exposition of methods for assessing the probabilities and the susceptibility to landslides have been improved now. We used the landslide hazard zonation map (now shown as Figure 4 (A) in the manuscript) for computing the spatial probability of landslide occurrence. We have now clarified our method of computing the spatial probability using the area and the colors on the map in the manuscript (pgs. 11-13). We have also shown the probability density function for the landslide spatial probability that we used in the ILS tool (Figure 4 (B)). Please note that this modeling exercise was not meant to very accurately determine the spatial probability of landslides in the study area. Rather, the primary objective of this exercise was to develop an approximate model that accounted for the spatial probability in ILS, where the tool was meant to improve people's understanding about landslide risks (pgs. 13 and 30-32).

**Finally, even the rain data used for the application seems insufficient: it is not realistic that after a few days we pass from probabilities higher than 0.8 to values below 0.3 (see Fig. 2). These irregularities indicate a poor representation of the sample used compared to an average trend.**

**Authors:** Thank you for the comment. The total probability of landslides is influenced by spatial probability of landslides, rainfall probability of landslides, and the human probability of landslides. Any of rapid changes in total probability of landslides as observed by you in the Figure could be a result of changes in any of these constituting probabilities (pg. 11-13). To improve the exposition of our methods, we have now plotted the distribution of rainfall that was averaged over a 5-year period and this distribution was found to be similar to the one over a 30-year period. Also, we have now improved the explanation of calculation of the spatial probability of landslides in the study area (pg. 11-13).

Please note that our primary objective in the manuscript is not to accurately predict rainfall or other landslide parameters; rather, our objective is to understand how people improve their understanding about landslide disasters in a simulation that provides an approximate representation of the complex landslide phenomena. There is a large body of literature on using approximate simulation tools for improving people's understanding about natural processes like climate change and other natural disasters. Again, these simulation tools only use an approximate models of the real phenomena. The development and use of the ILS tool follows the same line of research and tradition. We have added these points to a "limitations" section in the manuscript (pgs. 29-31).

[revised manuscript text omitted]

We plotted the distribution of rainfall averaged over a 30-year period and this distribution was similar to the one that was computed over the past 5-year period. There was a peak during the monsoon months (mid-June to mid-September) in the average rainfall distribution. In addition, in this paper, we are using the 5-year rainfall data in a simulation (ILS) that is meant to improve their socio-economic understanding of landslide disasters. As our primary objective is not to accurately predict rainfall or other landslide parameters; rather, to educate people about landslide disasters, we have used an approximate model of the real landslide phenomena in the ILS simulation tool. There is a large body of literature on using simulation tools for improving people's understanding about natural processes like climate change and other natural disasters. Again, these simulation tools only use an approximate models of the real phenomena. The development and use of the ILS tool follows the same line of research and tradition.

[revised manuscript text omitted]
 in the ILS tool.Specifically, wplan to assumein the ILS tool we plan to test them against with different education levels

In the current experiment, we assumed a large disparity between a participant's property wealth and her daily income. In addition, as part of the ILS model, we did not consider any support from government or international agencies against damages from landslides. In certain cases, especially in developing countries, mitigation of landslide risks may be often financed by government or international agencies. As part of our future work, we plan to extend the ILS model to include assumptions of contributions from government or international agencies. Such assumptions will help us determine the willingness of common people to contribute against landslide disasters, which is important as the developing world becomes developed over time.

To test our hypotheses, scenario and a low damage , where the probability of property damage, injury, and fatality were high and low, respectively. However, such scenarios may not be , where people may want migrate from both low and damage areas in even the least developed countriesplan to calibrate theprobability of damages, , and fatality

Furthermore, in our experiment, when landslide did not occur and experiential feedback was present, people were presented with a smiling face followed by a message. The message

and emoticon were provided to connect the cause-and-effect relationships for participants in the ILS tool. However, it could also be that the landslide did not occur on a certain trial due to the stochasticity in the simulation rather than participants' investment actions. Although such situations are possible over shorter time-periods, however, over longer time-periods increased investments from people will only reduce the probability of landslides.

In this paper,ainvestment However, the ILS tool can easily be customized to different time periods ranging from seconds, minutes, hours, days, months, and years. the daily assumption ing-s In addition, in the experiment, we assumed a value of 0.7 and 0.8 for the weight (W) and return to mitigation (M) parameters. These W and M values indicated that landslide risks could largely be mitigated by human actions. However, this assumption may not be the case always, especially for mitigation measures like tree plantations. For example, afforestation may not help in reducing deep-seated landslides in hilly areas (Forbes, 2011). it Thus, it would be worthwhile investigating as part of future research on how people's decision-making evolves in conditions where investments likely influence the landslide probability parameters and incompared to conditions where investments do notunlikely influence the landslide probability much parameters. Some of these ideas form the immediate next steps in our ongoing research program on landslide risk communication.

| Page 45: [7] Deleted | \pratik | 17/03/18 8:50 PM |

---

## Author Response (AR3)

April 22nd, 2018

Dr. Stefano Luigi Gariano
Editor, *Journal of Natural Hazards and Earth System Sciences*

Dear Dr. Gariano:

I write to you concerning a manuscript, "Learning in an Interactive Simulation Tool against Landslide Risks: The Role of Strength and Availability of Experiential Feedback," that I co-authored with my Ph.D. advisor, Dr. Varun Dutt and Mr. Akshit Arora.

We want to thank you for considering our work to the special issue on "Landslide early warning systems: monitoring systems, rainfall thresholds, warning models, performance evaluation and risk perception" of NHESS journal.

As per your kind suggestions, we have addressed all the minor comments. We are now submitting a revised version of the manuscript with point-to-point replies against the comments and suggestions given by you and the anonymous referee. These point-to-point replies are attached with this covering letter as an annexure "A".

We look forward to hearing from you on our revision.

Sincerely,

Pratik Chaturvedi

Ph.D. Scholar, School of Computing and Electrical Engineering

Indian Institute of Technology Mandi

Kamand-175005, Himachal Pradesh, India

Phone: +91-931-313-1129

Email: prateek@dtrl.drdo.in

**Comments to the Author:**
Dear Authors,
Anonymous Referee #1 has found several improvements in your manuscript, achieved during the review phase. Thus, he/she has evaluated your paper as publishable after minor revisions. I suggest you to follow his/her suggestions and to re-submit a final version of your paper. Moreover, since the reviewer noted that your work can be improved and deepened in next years, I also suggest to add a short paragraph in the conclusions describing the future planned improvements of your research.
The reviewed manuscript will be briefly checked again by me, before proceeding with the subsequent steps.
Best regards,
Stefano Luigi Gariano
NHESS Guest Editor

**Authors:** Thank you for summarizing our contribution and providing encouragement to our work. In agreement with you and reviewer, we have now also added a paragraph in the conclusion section describing the future planned improvements of our research work (pgs. 31-32). Finally, we have now also improved the quality of all the figures in the revised manuscript.

**After several stages, the work results improved and, in my view, worth to be published after minor revisions. The topic is highly interesting also if landslide dynamics are not deeply included and explained in the text. Basically, it represents a first stage of a work that should be deepened in next years but it could be a good example for researchers involved in gamification and participative processes.**

**Authors:** Thank you for summarizing our contribution and providing encouragement to our work. As per your kind suggestions, we have now discussed that this paper is a preliminary work and that we plan to deepen it in the near future (pg. 31-32). Also, that this paper forms a good example for researchers involved in gamification and participative processes (pg. 31).

**Furthermore, minor revisions should be introduced:**
**• Generally, the average quality of the pictures should be improved.**

**Authors:** Thank you for your comment. In the revised manuscript, we have improved the quality of all pictures.

**• L41-43: please enrich literature also considering works included in the same special issue.**

**Authors:** Thank you for your comment. As per your suggestion, we have added literature from the current special issue of NHESS (pg. 2).

**• L161: please report also in dollars or Euro.**

**Authors:** As per your kind suggestion, we have now reported the average daily income in dollars or Euro in the revised manuscript (pg. 6).

**• L171: please verify that you cite Google maps source as required by the service; if possible improve the quality of the image**

**Authors:** Thank you for your comment. In the revised manuscript, we have properly cited the Google image as required by the service and also improved the quality of the image. We have also drawn a perimeter around the study area using a red line in the image (pg. 6).

**• L248/L279-287: in this case, you could probably consider it landslide susceptibility also if it is considered by official documents as hazard.**

**Authors:** Thank you for your comment. We have now explicitly mentioned that the landslide hazard zonation (LHZ) map considers the landslide susceptibility of the area (pg. 11).

**• L312: please check for "her"**

**Authors:** We have corrected this typo in the revised manuscript (pg. 12).

• **L323: please avoid "benign" for a landslide**

**Authors:** We have replaced the term "benign" with the term "harmless" everywhere in the revised manuscript (pgs. 13, 16).

• **L330: please consider that, in 300 years, very different socio-economic conditions could arise.**

**Authors:** In agreement with you, we have now acknowledged that very different socio-economic conditions could arise over this long period. However, we do not use this period; rather, we vary this probability in the experiment (pg. 13).

• **L339: I'm not English mother tongue but probably "to allow" require –ing forms.**

**Authors:** Thank you for the comment. We have corrected this error and improved the grammar in the revised manuscript (pg. 13).

• **L495-505: I still have doubts about the representativeness of the sample for the area (age and level of education) although, in the text, some lines about it are reported (f.e. the percentage of phD)**

**Authors:** We have now discussed that there is a possibility that the participant demographics may not be representative of the area (pg. 31). Thus, as part of future research, we plan to control the sample in different ways and test the effects that demographics produces on people's investments (pg. 31).

• **L624-625: in my view, ILS is primarily devoted to Adult audiences given the required choices.**

**Authors:** Thank you for the comment. We have now discussed this part in the conclusion section (pg. 32).

• **L647: probably, the release of a version for policymakers should be evaluated (f.e. assessing where to perform the protection work)**

[revised manuscript text omitted]